# Learning Curves of Stochastic Gradient Descent in Kernel Regression

**Haihan Zhang** [* 1]  **Weicheng Lin** [* 1]  **Yuanshi Liu** [1]  **Cong Fang** [1 2]

## Abstract

This paper considers a canonical problem in kernel regression: how good are the model performances when it is trained by the popular online first-order algorithms, compared to the offline ones, such as ridge and ridgeless regression? In this paper, we analyze the foundational single-pass Stochastic Gradient Descent (SGD) in kernel regression under source condition where the optimal predictor can even not belong to the RKHS, i.e. the model is misspecified. Specifically, we focus on the inner product kernel over the sphere and characterize the exact orders of the excess risk curves under different scales of sample sizes $n$ concerning the input dimension $d$. Surprisingly, we show that SGD achieves min-max optimal rates up to constants among all the scales, *without* suffering the saturation, a prevalent phenomenon observed in (ridge) regression, except when the model is highly misspecified and the learning is in a final stage where $n \gg d^\gamma$ with any constant $\gamma > 0$. The main reason for SGD to overcome the curse of saturation is the exponentially decaying step size schedule, a common practice in deep neural network training. As a byproduct, we provide the *first* provable advantage of the scheme over the iterative averaging method in the common setting.

## 1. Introduction

Non-parametric least-squares regression within the RKHS framework represents a cornerstone of statistical learning theory. One mainstream method to solve the problem is kernel ridge regression (KRR) with optimality analysis (Caponnetto & De Vito, 2007; Smale & Zhou, 2007; Zhang et al., 2024a). Recent years have witnessed a renaissance of inter-est in kernel methods driven by the neural tangent kernel (NTK) theory (Jacot et al., 2018; Arora et al., 2019), which states that sufficiently wide neural networks, under specific initialization, can be well approximated by a deterministic kernel model derived from the network architecture. Though deep learning often operates in regimes beyond the traditional statistical mindset, recent advances demonstrate that these generalization mysteries are not peculiar to neural networks and the phenomena are also present in kernel regression, particularly in the high-dimensional regime (Ghorbani et al., 2021; Liang & Rakhlin, 2020; Zhang et al., 2024b).

Substantial studies have been made in the related regimes for kernel ridge or ridgeless methods. For instance, Liang & Rakhlin (2020) demonstrates the existence of benign overfitting for ridgeless regression, a phenomenon where the model interpolates data yet still generalizes well. In a large-dimension regime where $n \asymp d$, Ghorbani et al. (2021) provably shows the double descent phenomenon. In more general regimes, Zhang et al. (2024b) characterizes the exact orders of the excess risk curves, i.e., learning curves where the sample size scales polynomially with the input dimension $n = d^\gamma$ with $\gamma > 0$ and elaborates the interplay of the regularization parameter, $\gamma$, and the source condition.

On the other hand, in practice, dominant algorithms are implemented in an online fashion to reduce computation costs, with iterative updates relying on stochastic gradients (single-pass or minibatch) as estimates of the population counterpart. Stochastic gradient descent (SGD), one of the foundational online algorithms, admits the simplest updates and receives abundant theoretical analysis in general optimization. However, its learning dynamics over specific kernels remains underexplored. Existing theoretical results are either inapplicable or leave room for improvement. Specifically, the pioneering work of (Dieuleveut & Bach, 2016) considers the optimality of single-pass SGD with iterate averaging, yet their optimality results do not demonstrate a clear improvement over ridge regression. Another thread of work analyses single-pass SGD in the linear regression framework (Jain et al., 2018; Zou et al., 2021b; Wu et al., 2022). Although insightful, their decisive assumption on the concentration effect deviates from that in kernel regression, resulting in entirely different learning dynamics.

In this study, we investigate a fundamental question to fill

---

[*]Equal contribution [1]National Key Lab of General Artificial Intelligence, School of Intelligence Science and Technology, Peking University [2]Institute for Artificial Intelligence, Peking University. Correspondence to: Cong Fang <fangcong@pku.edu.cn>.

*Proceedings of the 42$^{nd}$ International Conference on Machine Learning*, Vancouver, Canada. PMLR 267, 2025. Copyright 2025 by the author(s).

the abovementioned gap: how does model performance, when trained using popular online first-order algorithms, compare to that of offline ridge and ridgeless regression? We analyze the single-pass SGD that incorporates the exponentially decaying step size schedule, a common practice in deep neural network training (Bengio, 2009; Ge et al., 2019). The analysis is conducted within the framework of kernel regression, considering the source condition that allows for potential model misspecification. We precisely characterize the order of learning curves in the high-dimensional setting, where $n \asymp d^\gamma$ with $\gamma > 0$ treated as a constant. Our focus is on the inner product kernel over the sphere, which encompasses NTK with ReLu activation as a special case (Bietti & Bach, 2021; Bietti & Mairal, 2019). For functions satisfying the source condition with $s > 0$, we establish excess risk convergence rates given by $\Omega\left(d^{-\min\{\gamma - p, s(p+1)\}}\right)$, where $p = \left\lfloor \frac{\gamma}{s+1} \right\rfloor$. These derived rates surprisingly match the minimax lower bounds up to constants, thereby establishing minimax optimality across all sampling scales in high-dimensional settings and throughout the spectrum of source condition smoothness (Zhang et al., 2024b). Notably, the result circumvents the saturation effect inherent to the KRR - the fundamental limitation where it fails to attain optimality for the smooth problems regardless of the ridge regularization parameter (Neubauer, 1997; Li et al., 2023; Lu et al., 2024b), which persists under the current online analysis.

With the above optimality analysis, two follow-up questions naturally arise. (i) Can the SGD algorithm consistently be optimal across all scalings of $n$, particularly when liberated from the $n \asymp d^\gamma$ constraints? (ii) Why can SGD overcome the curse of saturation?

For question (i), the claim does not hold in the asymptotic regime where $n \gg d^\gamma$: we demonstrate that SGD loses its optimality when the model is highly misspecified. Specifically, our analysis shows that SGD can only reach optimality for $s > \frac{1}{d+1}$.

For question (ii), we demonstrate that the exponentially decaying step size schedule serves as the critical mechanism for eliminating the saturation effect. To illustrate our arguments, we study a class of SGD with averaged iterates. Our result establishes a lower bound on this scheme, and as a byproduct, firstly demonstrates the provable advantage of decaying step sizes over iterate averaging.

To summarize our contribution:

- We establish the optimality condition of SGD in both high-dimensional and asymptotic settings under the source condition.

- We demonstrate the advantages of SGD with exponentially decaying step size over iterative averaging methods, highlighting the former's efficiency in well-specified problems without suffering from the saturation effect.

## 2. Related Work

**Kernel Regression.** In the asymptotic setting, when $n \gg d$, the eigenvalues of the RKHS kernel often exhibit polynomial decay $\lambda_i \asymp i^{-\alpha}$ (Bietti & Mairal, 2019; Bietti & Bach, 2021; Li et al., 2024), which is an extensively studied setting in the RKHS framework. Under a source condition of smoothness $s$ (which will be specified later in Section 3.1), Caponnetto & De Vito (2007) establishs a minimax lower bound of $n^{-\frac{s\alpha}{s\alpha+1}}$. For offline algorithms, KRR has been shown to achieve optimality for $0 < s \leq 2$ (Caponnetto & De Vito, 2007; Smale & Zhou, 2007; Steinwart et al., 2009; Dicker et al., 2017; Blanchard & Mücke, 2018; Fischer & Steinwart, 2020; Lin et al., 2020; Li et al., 2022; Liu & Shi, 2024; Zhang et al., 2024a), but suffers from the saturation effect when $s > 2$ (Neubauer, 1997; Bauer et al., 2007; Dicker et al., 2017; Cui et al., 2021; Li et al., 2023). Gradient descent (GD) with early stopping has been shown to achieve the minimax optimal rate for all $s > 0$ (Yao et al., 2007; Raskutti et al., 2014; Lin et al., 2020). Several studies have also revealed connections between the optimal stopping time in early-stopped GD and the regularization in KRR (Ali et al., 2019; Sonthalia et al., 2024). Velikanov et al. (2024) further demonstrates that spectral algorithms with finite qualification $\tau$ experience saturation and fail to achieve optimality for $s > 2\tau$, with only GF with early stopping able to achieve optimality (Raskutti et al., 2014). For online algorithms, Dieuleveut & Bach (2016) shows that single-pass SGD with constant step size and averaged iterates can achieve optimality for $\frac{\alpha-1}{\alpha} < s \leq 2$.

In the high-dimensional setting, where $n \asymp d^\gamma$, for $\gamma > 0$, the eigenvalues of the RKHS kernel decay at a rate that is not polynomial depending on $d$, leading to the ineffectiveness of asymptotic analysis. Recently, several works have investigated the excess risk of offline algorithms in this setting. It has been proven that KRR and kernel gradient flow are capable of achieving convergent excess risk for certain regression functions (Advani et al., 2020; Bordelon et al., 2020; Ghorbani et al., 2021; Donhauser et al., 2021; Xiao et al., 2022; Misiakiewicz, 2022; Tsigler & Bartlett, 2023). Kernel interpolation is shown to exhibit the benign overfitting phenomenon, which allows generalization under certain values of $\gamma$ and $s$ (Liang & Rakhlin, 2020; Liang et al., 2020; Bartlett et al., 2020; Barzilai & Shamir, 2024; Zhang et al., 2025). Both the excess risk and minimax rate have been shown to exhibit periodic plateau behavior and multiple descent behaviors in the high-dimensional setting (Bordelon et al., 2020; Ghorbani et al., 2021; Xiao et al., 2022; Lu et al., 2023; Zhang et al., 2024b). For the minimax optimality analysis in high-dimensional settings,

KRR has been demonstrated to roughly achieve optimality for $s \leq 1$ and $\gamma > \frac{3s}{2(s+1)}$, while experiencing saturation effects in the remaining range (Zhang et al., 2024b). Lu et al. (2024b) further demonstrates that offline algorithms with finite qualification $\tau$ experience saturation and fail to achieve optimality for $s > \tau$, with only gradient flow with early stopping able to achieve optimality for all $s > 0$ and $\gamma > 0$. However, in high-dimensional settings, the excess risk curve and optimality analysis for online algorithms remain an open area of research.

**SGD over Linear Model.** SGD in linear regression has been extensively studied. When the RKHS is finite-dimensional, it has been shown that single-pass SGD, with either an exponentially decaying step size or a constant step size with averaged iterates, achieves the minimax optimal rate $\mathcal{O}\left(\frac{d}{n}\right)$ (Bach & Moulines, 2013; Jain et al., 2018; Ge et al., 2019). In overparameterized linear regression, single-pass SGD exhibits a benign overfitting behavior (Zou et al., 2021a;b; Wu et al., 2022), with Zou et al. (2021b) and Wu et al. (2022) establishing excess risk bounds for SGD under both constant step sizes with iterate averaging and exponentially decaying step size schedules. These results crucially rely on a fourth-moment condition on the covariates to ensure sufficient concentration, in addition to depending on the spectral properties of the covariates and the target parameters. Under the source condition with $s$ and polynomial eigenvalue decay of the covariance matrix $\lambda_i \asymp i^{-\alpha}$, Zhang et al. (2024c) showed that SGD with an exponentially decaying step size can attain the minimax optimal rate for all $s \geq \frac{\alpha-1}{\alpha}$. For general kernel regression problems, under the source condition with $s$ and the spectrum of the kernel operator that decays polynomially $\lambda_i \asymp i^{-\alpha}$, Dieuleveut & Bach (2016); Dieuleveut et al. (2017) demonstrated that SGD achieves optimal performance when $\frac{\alpha-1}{\alpha} \leq s \leq 2$, while failing to reach optimality and encountering saturation for $s > 2$. In the low-regularity regime $s < \frac{\alpha-1}{\alpha}$, single-pass SGD is limited by its optimization capacity and cannot reach optimality; in such cases, momentum acceleration (Zhang et al., 2024c; Dieuleveut et al., 2017) or multiple passes over the data (Lin & Rosasco, 2017; Pillaud-Vivien et al., 2018) are required.

## 3. Preliminaries

**Notations.** In this paper, we consider the input space $\mathcal{X} \subseteq \mathbb{R}^{d+1}$ and the output space $\mathcal{Y} \subseteq \mathbb{R}$, with a probability distribution $\rho$ on $\mathcal{X} \times \mathcal{Y}$. Let $\rho_{\mathcal{X}}$ denote the marginal probability distribution on $\mathcal{X}$, $\rho_{y|\mathbf{x}}$ the conditional distribution of $y$ given $\mathbf{x}$. $\|\cdot\|_{L^2}$ denotes the norm $\|f\|_{L^2}^2 = \int_{\mathcal{X}} |f(\mathbf{x})|^2 \, d\rho_{\mathcal{X}}(\mathbf{x})$ where $L^2$ represents the $L^2$-space $L^2(\mathcal{X}, \rho_{\mathcal{X}})$. Denote $\mathbb{S}^d$ as the unit sphere in $\mathbb{R}^{d+1}$. In a Hilbert space $\mathcal{H}$, when $\mathbf{A}$ and $\mathbf{B}$ are self-adjoint operators, the symbol $\mathbf{A} \preceq \mathbf{B}$ denotes a positive semi-definite

relationship, that is: for any $f \in \mathcal{H}$, $\langle f, \mathbf{A}f \rangle_{\mathcal{H}} \leq \langle f, \mathbf{B}f \rangle_{\mathcal{H}}$. For any $f, g \in \mathcal{H}$, define the operator $f \otimes g : \mathcal{H} \to \mathcal{H}$ by $f \otimes g(h) = \langle f, h \rangle_{\mathcal{H}} g$.

### 3.1. Regression in RKHS

We consider the regression problem of minimizing the prediction error for a function $f \in L^2$, defined as

$$\mathcal{E}_\rho(f) = \mathbb{E}_\rho\left[(f(\mathbf{x}) - y)^2\right],$$

where the optimal predictor $f_\rho^*(\mathbf{x}) = \mathbb{E}_\rho[y|\mathbf{x}]$ achieves the minimal error. The prediction error of $f$ can be equivalently measured by the excess risk $\|f - f_\rho^*\|_{L^2}^2$.

In this paper, we investigate regression problems within the framework of the Reproducing Kernel Hilbert Space (RKHS) (Berlinet & Thomas-Agnan, 2011). Let $\mathcal{H}$ denote the RKHS associated with a continuous kernel $K$ defined on the input space $\mathcal{X}$. The covariance operator $T : L^2 \to \mathcal{H}$ associated with the kernel $K$ and the marginal distribution $\rho_{\mathcal{X}}$ is defined as:

$$T(f)(\mathbf{z}) = \int_{\mathcal{X}} f(\mathbf{x}) K(\mathbf{x}, \mathbf{z}) \, d\rho_{\mathcal{X}}(\mathbf{x}).$$

By Mercer's theorem (Aronszajn, 1950; Steinwart & Scovel, 2012),

$$T = \sum_{i=1}^{\infty} \lambda_i \langle \cdot, \phi_i \rangle_{L^2} \phi_i,$$

$$K(\mathbf{x}, \mathbf{y}) = \sum_{i=1}^{\infty} \lambda_i \phi_i(\mathbf{x}) \phi_i(\mathbf{y}),$$

where $\{\lambda_i\}_{i=1}^{\infty}$ are the eigenvalues of $T$ in non-increasing order, and $\{\phi_i\}_{i=1}^{\infty}$ are the corresponding eigenfunctions of $T$ which form an orthonormal basis of $L^2$. Let $K_{\mathbf{x}}$ denote the short hand of the function $K_{\mathbf{x}} = K(\mathbf{x}, \cdot)$. If the embedding operator $T^* : \mathcal{H} \to L^2$ satisfies $\overline{\text{Ran}(T^*)} \subseteq L^2$, then $\boldsymbol{\Sigma} : \mathcal{H} \to \mathcal{H}$ defined as $\boldsymbol{\Sigma} = \mathbb{E}[K_{\mathbf{x}} \otimes K_{\mathbf{x}}]$ admits the decomposition of

$$\boldsymbol{\Sigma} = \sum_{i=1}^{\infty} \lambda_i \langle \cdot, \lambda_i^{1/2} \phi_i \rangle_{\mathcal{H}} \lambda_i^{1/2} \phi_i,$$

with $\left\{\lambda_i^{\frac{1}{2}} \phi_i\right\}_{i=1}^{\infty}$ forming an orthonormal basis of $\mathcal{H}$ (Dieuleveut & Bach, 2016).

To quantify the regularity of the optimal predictor $f_\rho^*$, we introduce the interpolation space $[\mathcal{H}]^s$. For any $s \geq 0$, the $s$-th power of $T$ is defined as

$$T^s(f) = \sum_{i=1}^{\infty} \lambda_i^s \langle f, \phi_i \rangle_{L^2} \phi_i.$$

The interpolation space $[\mathcal{H}]^s$ is defined by

$$[\mathcal{H}]^s = \left\{ \sum_{i=1}^{\infty} a_i \lambda_i^{\frac{s}{2}} \phi_i \mid \{a_i\}_{i=1}^{\infty} \in \ell^2 \right\},$$

with the inner product $\langle f, g \rangle_{[\mathcal{H}]^s} = \left\langle T^{-\frac{s}{2}} f, T^{-\frac{s}{2}} g \right\rangle_{L^2}$.

### 3.2. Dot-Product Kernels on the Sphere

A dot-product kernel $K$ is defined as $K(\mathbf{x}, \mathbf{y}) = \Phi(\langle \mathbf{x}, \mathbf{y} \rangle)$, for any $\mathbf{x}, \mathbf{y} \in \mathbb{S}^d$, where $\Phi \in \mathcal{C}^{\infty}[-1, 1]$ is a fixed function independent of $d$. The Mercer's decomposition for $K$ (Gallier, 2009) can be written as

$$K(\mathbf{x}, \mathbf{y}) = \sum_{k=0}^{N(d,k)} \mu_k \sum_{j=1}^{N(d,k)} Y_{k,j}(\mathbf{x}) Y_{k,j}(\mathbf{y}),$$

where $\{Y_{k,j}\}_{j=1}^{N(d,k)}$ are spherical harmonic polynomials of degree k and $\mu_k's$ are the eigenvalues of $K$ with multiplicity $N(d, 0) = 1$, $N(d, k) = \frac{2k+d-1}{k} \cdot \frac{(k+d-2)!}{(d-1)!(k-1)!}$, for $k \geq 1$.

The smoothness of $\Phi \in \mathcal{C}^{\infty}[-1, 1]$ and compactness of $\mathbb{S}^d$ ensure that the dot-product kernel $K$ is a bounded kernel. For any dimension $d$, the dot-product kernel $K$ exhibits the following properties:

**Property 1.** *There exists $\kappa > 0$, such that*

- *(a) The dot-product kernel $K$ is bounded, that is $\sup_{\mathbf{x} \in \mathcal{X}} K(\mathbf{x}, \mathbf{x}) \leq \kappa^2$.*

- *(b) The trace of the covariance operator $T$ is bounded, that is, $\text{tr}(T) \leq \kappa^2$.*

- *(c) The marginal probability distribution $\rho_{\mathcal{X}}$ satisfies $\mathbb{E}[K(\mathbf{x}, \mathbf{x}) K_{\mathbf{x}} \otimes K_{\mathbf{x}}] \preceq \kappa^2 \Sigma$.*

The NTK of a ReLU network with $L$ layers with inputs on $\mathbb{S}^d$ is a specific dot-product kernel (Jacot et al., 2018; Bietti & Bach, 2021), which is defined as

$$K_{\text{NTK}}(\mathbf{x}, \mathbf{y}) = \kappa_{\text{NTK}}^L(\langle \mathbf{x}, \mathbf{y} \rangle).$$

$\kappa_{\text{NTK}}^L$ is defined recursively, starting with $\kappa_{\text{NTK}}^1(t) = \kappa^1(t) = t$, and for $2 \leq \ell \leq L$,

$$\kappa^\ell(t) = \kappa_1\left(\kappa^{\ell-1}(t)\right),$$
$$\kappa_{\text{NTK}}^\ell(t) = \kappa_{\text{NTK}}^{\ell-1}(t)\kappa_0\left(\kappa^{\ell-1}(t)\right) + \kappa^\ell(t),$$

where $\kappa_0(t) = \frac{1}{\pi}(\pi - \arccos(t))$ and $\kappa_1(t) = \frac{1}{\pi}\left(t(\pi - \arccos(t)) + \sqrt{1-t^2}\right)$. Let the Mercer decomposition of $K_{\text{NTK}}$ be $K_{\text{NTK}}(\mathbf{x}, \mathbf{y}) = \sum_{i=1}^{\infty} \lambda_i \phi_i(\mathbf{x}) \phi_i(\mathbf{y})$, where $\{\lambda_i\}_{i=1}^{\infty}$ are the eigenvalues of $K_{\text{NTK}}$ in non-increasing order, and $\{\phi_i\}_{i=1}^{\infty}$ are the corresponding eigenfunctions. As demonstrated in Bietti & Bach (2021), when

$n \gg d$, the decay rate of the eigenvalues of $K_{\text{NTK}}$ is given by

$$\lambda_j \asymp j^{-\frac{d+1}{d}}, \quad (1)$$

indicating that the eigenvalues of $K_{\text{NTK}}$ decay polynomially. This decay rate is consistent with the standard capacity condition typically employed to assess the optimality of algorithms in the RKHS framework (Caponnetto & De Vito, 2007; Steinwart et al., 2009).

## 4. Setup

### 4.1. SGD for Kernel Regression

Leveraging the properties of RKHS, for any function $f \in L^2$, recall that the prediction error of $f$ is expressed as

$$\mathcal{E}_\rho(f) = \mathbb{E}_\rho[y - f(x)]^2.$$

We then employ stochastic gradient descent (SGD) to solve the kernel regression problem. Starting from an initial function $f_0$, the SGD updates are performed recursively as follows:

$$f_t = f_{t-1} - \eta_t\left(f_{t-1}(\mathbf{x}_t) - y_t\right) K_{\mathbf{x}_t}.$$

At the $t$-th iteration, we observe a fresh data $(\mathbf{x}_t, y_t)$ from $\rho$, and $\eta_t$ denotes the step size. Assuming the initialization starts from $f \equiv 0$, $f_t$ can be expressed as:

$$f_t = \sum_{j=1}^{t} a_j K_{\mathbf{x}_j},$$

where $(a_j)_{1 \leq j \leq t}$ is given by the following iteration, starting from $a_0 = 0$:

$$a_t = -\eta_t\left(\sum_{j=1}^{t-1} a_j K(\mathbf{x}_j, \mathbf{x}_t) - y_t\right).$$

When the kernel is chosen as the NTK, the dynamics of SGD for kernel regression exactly coincide with the training process of an infinitely wide neural network under gradient descent (Jacot et al., 2018). For finite-width networks, this correspondence holds approximately in the early training phase (Chizat et al., 2019). In this paper, we investigate two types of step size schedules: an exponentially decaying step size schedule (Bengio, 2009; Ge et al., 2019) and a constant step size with averaged iterates (Polyak & Juditsky, 1992; Dieuleveut & Bach, 2016).

- **Exponentially Decaying Step Size Schedule.** Given a total of $n$ iterations, the step size is piecewise constant and decays by a factor after each stage, as defined by

$$\eta_t = \frac{\eta_0}{2^{\ell-1}}, \text{ if } m(\ell-1)+1 \leq t \leq m\ell, \quad (2)$$

for $1 \leq \ell \leq \lceil \log_2 n \rceil$, where $\lceil \log_2 n \rceil$ is the total number of stages, and $m = \lceil \frac{n}{\log_2 n} \rceil$. The final iterate $f_n^{dec} = f_n$ is considered the output.

- **Constant Step Size with Averaged Iterates.** In this schedule, the step size remains constant, and the output is given by the average of the $n$ iterates: $f_n^{avg} = \frac{1}{n} \sum_{t=0}^{n-1} f_t$.

## 4.2. Assumptions

### 4.2.1. DATA NOISE ASSUMPTION

Denote $\Xi_{(\mathbf{x},y)} = \left(y - f_\rho^*(\mathbf{x})\right) K_\mathbf{x}$ as the residual of $(\mathbf{x}, y)$, where $y - f_\rho^*(\mathbf{x})$ can be regarded as the noise of data. To control the interaction between bias and variance in the RKHS, we assume the residual's covariance structure satisfies the following assumption.

**Assumption 4.1.** There exists $\sigma > 0$ such that the residual $\Xi_{(\mathbf{x},y)}$ satisfies $\mathbb{E}\left[\Xi_{(\mathbf{x},y)} \otimes \Xi_{(\mathbf{x},y)}\right] \preceq \sigma^2 \boldsymbol{\Sigma}$ and $\mathbb{E}[\Xi_{(\mathbf{x},y)}] = 0$.

*Remark* 4.2. This assumption holds when the data is bounded or $y - f_\rho^*(\mathbf{x})$ is sub-Gaussian conditioned on $\mathbf{x}$. It does not require $\mathbb{E}[\Xi|\mathbf{x}] = 0$, thus encompassing a broader class of problems. This assumption is also adopted by several works (Bach & Moulines, 2013; Dieuleveut & Bach, 2016; Dieuleveut et al., 2017).

### 4.2.2. SOURCE CONDITION

The source condition assumes that the optimal predictor $f_\rho^*$ lies within a bounded ball in $[\mathcal{H}]^s$.

**Assumption 4.3.** For a given $s > 0$, we assume that $f_\rho^*$ belongs to the unit ball in $[\mathcal{H}]^s$, that is,

$$\left\| f_\rho^* \right\|_{[\mathcal{H}]^s} \leq 1.$$

*Remark* 4.4. The source condition serves as a criterion for evaluating optimality in the RKHS framework. It characterizes the smoothness of $f_\rho^*$: a larger $s$ implies a faster decay of the Fourier coefficients of $f_\rho^*$ in $L^2$, indicating higher smoothness and an intrinsically simpler learning problem. This condition further categorizes the problem settings: $0 < s < 1$ is called the misspecified problem, and $s \geq 1$ is called the well-specified problem.

### 4.2.3. ASSUMPTION ON THE DOT-PRODUCT KERNEL

For the high-dimensional settings, where $c_1 d^\gamma < n < c_2 d^\gamma$ for some fixed $\gamma > 0$ and absolute constants $c_1, c_2$, we adopt the assumption for $\Phi$ as stated in Lu et al. (2023).

**Assumption 4.5.** There exists a non-negative sequence of absolute constants $\{a_j\}_{j=0}^\infty$, such that $\Phi(t) = \sum_{j=0}^\infty a_j t^j$, where $a_j > 0$ for any $j \leq \lfloor \gamma \rfloor + 3$.

*Remark* 4.6. This assumption is made to maintain the clarity of the main results and proofs, and it can be extended to the NTK as discussed in Lu et al. (2023).

## 4.3. Minimax Lower Bound

The minimax lower bound serves as a fundamental criterion for evaluating statistical optimality in learning algorithms. An algorithm achieves minimax optimality if its worst-case excess risk upper bound matches the corresponding minimax lower bound over a given function class. Formally, for a family of distributions $\mathcal{P}$ and estimators $\hat{f}$ trained on $n$ i.i.d. samples $\{(\mathbf{x}_i, y_i)\}_{i=1}^n$, the minimax lower bound is defined as:

$$\inf_{\hat{f}} \sup_{\rho \in \mathcal{P}} \left\| \hat{f} - f_\rho^* \right\|_2^2,$$

characterizing the irreducible statistical error in the worst-case distributional setting. The minimax lower bound for some kernel methods has been extensively studied (Yang & Barron, 1999; Caponnetto & De Vito, 2007; Lu et al., 2023; 2024a). Below, we now present two key minimax lower bounds under distinct scaling regimes. The first result, adapted from Lu et al. (2024a), addresses the high-dimensional setting under source condition with $s > 0$.

**Proposition 4.7** (Minimax Lower Bound in High-Dimensional Settings). *In high-dimensional settings, where $n$ is bounded by $c_1 d^\gamma < n < c_2 d^\gamma$ for some fixed $\gamma > 0$ and constants $c_1, c_2$, consider $\mathcal{X} = \mathbb{S}^d$. The marginal distribution $\rho_\mathcal{X}$ is assumed to be the uniform distribution on $\mathbb{S}^d$. Let $K$ denote a dot-product kernel on the sphere, which satisfies Assumption 4.5. Let $\mathcal{P}$ consist of all distributions $\rho$ on $\mathcal{X} \times \mathcal{Y}$ given by $y = f_\rho^*(\mathbf{x}) + \epsilon$, where $\epsilon \sim \mathcal{N}(0, 1)$ is independent of $\mathbf{x}$. Suppose Assumption 4.3 holds with $s > 0$. Let $p = \left\lceil \frac{\gamma}{s+1} \right\rceil - 1$, then we have:*

$$\inf_{\hat{f}} \sup_{\rho \in \mathcal{P}} \mathbb{E}_\rho \left\| \hat{f} - f_\rho^* \right\|_{L^2}^2 = \Omega \left( d^{-\min\{\gamma - p, s(p+1)\}} \right).$$

The second result pertains to the asymptotic setting $n \gg d$, leveraging the spectral properties of NTK for ReLU networks and the optimality analysis in Caponnetto & De Vito (2007) :

**Proposition 4.8** (Minimax Lower Bound in Asymptotic Settings). *In asymptotic settings, where $n \gg d$, consider $\mathcal{X} = \mathbb{S}^d$. The marginal distribution $\rho_\mathcal{X}$ is assumed to be the uniform distribution on $\mathbb{S}^d$. Let $K_{\mathrm{NTK}}$ the NTK of a ReLU network with $L$ layers with inputs on $\mathbb{S}^d$. Let $\mathcal{P}$ consist of all distributions $\rho$ on $\mathcal{X} \times \mathcal{Y}$ given by $y = f_\rho^*(\mathbf{x}) + \epsilon$, where $\epsilon \sim \mathcal{N}(0, 1)$ is independent of $\mathbf{x}$. Suppose Assumption 4.3 holds with $s > 0$. Then we have:*

$$\inf_{\hat{f}} \sup_{\rho \in \mathcal{P}} \mathbb{E}_\rho \left\| \hat{f} - f_\rho^* \right\|_{L^2}^2 = \Omega \left( n^{-\frac{s(d+1)}{s(d+1)+d}} \right).$$

## 5. Main Result

### 5.1. Convergence Rates in High-Dimensional Settings

We first present the convergence rates of the excess risk for SGD in high-dimensional settings where $n \asymp d^\gamma$. The-

orem 5.1 shows that SGD with exponentially decaying step size is efficient for well-specified problems, and Theorem 5.2 shows that SGD with averaged iterates is efficient for mis-specified problems.

**Theorem 5.1** (Convergence Rate for Well-Specified Problems). *In high-dimensional settings, where $n$ is bounded by $c_1 d^\gamma < n < c_2 d^\gamma$ for some fixed $\gamma > 0$ and constants $c_1, c_2$, consider $\mathcal{X} = \mathbb{S}^d$. The marginal distribution $\rho_{\mathcal{X}}$ is assumed to be the uniform distribution on $\mathbb{S}^d$. Let $K$ denote the dot-product kernel on $\mathbb{S}^d$, which satisfies Assumption 4.5. Given $s \geq 1$, supposing that Assumption 4.3 holds with $s$, and treating $\gamma, \sigma, \kappa, c_1, c_2$ and $s$ as constants, the excess risk of the output of SGD with exponentially decaying step size $f_n^{dec}$ satisfies:*

*(i) When $\gamma \in (ps + p, ps + p + s]$ for some $p \in \mathbb{N}$, with initial step size $\eta_0 = \Theta(d^{-\gamma+p} \log_2 n \ln d)$, there exists a constant $d_0$ such that for any $d \geq d_0$, we have:*

$$\mathbb{E}\left[\|f_n^{dec} - f_\rho^*\|_{L^2}^2\right] \lesssim d^{-\gamma+p} \log_2^2 d.$$

*(ii) When $\gamma \in (ps + p + s, (p+1)s + p + 1]$ for some $p \in \mathbb{N}$, with initial step size $\eta_0 = \Theta(d^{-\gamma+p} \log_2 n \ln d)$, there exists a constant $d_0$ such that for any $d \geq d_0$, we have:*

$$\mathbb{E}\left[\|f_n^{dec} - f_\rho^*\|_{L^2}^2\right] \lesssim d^{-(p+1)s}.$$

Compared to the minimax lower bound in Proposition 4.7, Theorem 5.1 demonstrates that SGD with exponentially decaying step size can achieve optimality for any $\gamma > 0$ and $s \geq 1$. Notably, this theorem illustrates that SGD with exponentially decaying step size does not suffer from the saturation effect encountered by KRR when $n \asymp d^\gamma$ (Neubauer, 1997; Bauer et al., 2007). Specifically, when $n \asymp d^\gamma$, Lu et al. (2024b) shows that if Assumption 4.3 holds with $s > 1$, the convergence rate of the excess risk for KRR can be bounded from below by

$$\mathbb{E}\left(\left\|\hat{f}_{\lambda_*}^{\mathrm{KRR}} - f_\rho^*\right\|_{L^2}^2\right)$$
$$= \Theta_{\mathbb{P}}\left(d^{-\min\left\{\gamma-p, \frac{\gamma-p+p\tilde{s}+1}{2}, \tilde{s}(p+1)\right\}}\right) \cdot poly\left(\ln(d)\right),$$

where $\tilde{s} = \min\{s, 2\}$, and no choice of the regularization parameter $\lambda = \lambda(d, n)$ leads to an improvement in the excess risk. This result suggests that when $n \asymp d^\gamma$ and $s > 1$, there exists a region in which KRR fails to achieve optimality. Therefore, for high-dimensional problems, SGD with an exponentially decaying step size matches or outperforms KRR in well-specified problems.

**Theorem 5.2** (Convergence Rate for Mis-Specified Problems). *In high-dimensional settings, where $n$ is bounded by $c_1 d^\gamma < n < c_2 d^\gamma$ for some fixed $\gamma > 0$ and constants $c_1, c_2$, consider $\mathcal{X} = \mathbb{S}^d$. The marginal distribution $\rho_{\mathcal{X}}$ is assumed to be the uniform distribution on $\mathbb{S}^d$. Let $K$ denote the dot-product kernel on $\mathbb{S}^d$, which satisfies Assumption 4.5. Given*

$0 < s < 1$, *supposing that Assumption 4.3 holds with $s$, and treating $\gamma, \sigma, \kappa, c_1, c_2$ and $s$ as constants, the excess risk of the output of SGD with averaged iterates $f_n^{avg}$ satisfies:*

*(i) When $\gamma \in (ps+p, ps+p+s]$ for some $p \in \mathbb{N}$, with initial step size $\eta_0 = \Theta(d^{-\gamma+p+\frac{s}{2}})$ $(p > 0)$ or $\eta_0 = \Theta(d^{-\frac{\gamma}{2}})$ $(p = 0)$, there exists a constant $d_0$ such that for any $d \geq d_0$, we have:*

$$\mathbb{E}\left[\|f_n^{avg} - f_\rho^*\|_{L^2}^2\right] \lesssim d^{-\gamma+p}.$$

*(ii) When $\gamma \in (ps + p + s, (p+1)s + p + 1]$ for some $p \in \mathbb{N}$, with initial step size $\eta_0 = \Theta(d^{-\gamma+p+\frac{s}{2}})$, there exists a constant $d_0$ such that for any $d \geq d_0$, we have:*

$$\mathbb{E}\left[\|f_n^{avg} - f_\rho^*\|_{L^2}^2\right] \lesssim d^{-(p+1)s}.$$

Compared to the minimax lower bound in Proposition 4.7, Theorem 5.2 demonstrates that SGD with averaged iterates can achieve optimality for any $\gamma > 0$ and $0 < s < 1$. In contrast, when $0 < s \leq 1$ and $n \asymp d^\gamma$, Zhang et al. (2024a) shows that KRR only achieves optimality if $\gamma > \frac{3s}{2(s+1)}$. This indicates that, for high-dimensional problems, SGD has an advantage over KRR when dealing with mis-specified problems.

### 5.2. Convergence Rates in Asymptotic Settings

In this section, we present the convergence rates of the excess risk for SGD in asymptotic settings where $n \gg d$.

**Theorem 5.3** (Convergence Rate for Well-Specified Problems). *In asymptotic settings, where $n \gg d$, consider $\mathcal{X} = \mathbb{S}^d$. The marginal distribution $\rho_{\mathcal{X}}$ is assumed to be the uniform distribution on $\mathbb{S}^d$. Let $K_{\mathrm{NTK}}$ denote the NTK of a ReLU network with $L$ layers with inputs on $\mathbb{S}^d$. For $s \geq 1$, supposing that Assumption 4.3 holds for $s$, with initial step size $\eta_0 = \Theta\left(n^{\frac{1-s(d+1)}{s(d+1)+d}}\right)$, the excess risk of the output of SGD with exponentially decaying step size $f_n^{dec}$ satisfies:*

$$\mathbb{E}\left[\|f_n^{dec} - f_\rho^*\|_{L^2}^2\right] \lesssim \log_2^s n \cdot n^{-\frac{s(d+1)}{s(d+1)+d}}.$$

Compared to the minimax lower bound in Proposition 4.8, Theorem 5.3 demonstrates that SGD with exponentially decaying step size can achieve optimality for any $s > 1$. This further indicates that in asymptotic settings, SGD with exponentially decaying step size does not encounter the saturation effect. In contrast, Cui et al. (2021) and Li et al. (2023) show that if Assumption 4.3 holds with $s > 2$, the convergence rate of the excess risk for KRR is bounded from below by:

$$\mathbb{E}\left(\left\|\hat{f}_{\lambda_*}^{\mathrm{KRR}} - f_\rho^*\right\|_{L^2}^2\right) = \Theta_{\mathbb{P}}\left(n^{-\frac{2d+2}{3d+2}}\right),$$

This result suggests that for $s > 2$, KRR becomes suboptimal. Consequently, we argue that SGD with exponentially

decaying step size is more effective for problems where $f_\rho^*$ is sufficiently smooth.

**Theorem 5.4** (Convergence Rate for Mis-Specified Problems). *In asymptotic settings, where $n \gg d$, consider $\mathcal{X} = \mathbb{S}^d$. The marginal distribution $\rho_\mathcal{X}$ is assumed to be the uniform distribution on $\mathbb{S}^d$. Let $K_{\mathrm{NTK}}$ denote the NTK of a ReLU network with $L$ layers with inputs on $\mathbb{S}^d$. For $\frac{1}{d+1} \le s \le 1$, supposing that Assumption 4.3 holds for $s$, with initial step size $\eta_0 = \Theta\left(n^{\frac{1-s(d+1)}{s(d+1)+d}}\right)$, the excess risk of the output of SGD with averaged iterates $f_n^{avg}$ satisfies:*

$$\mathbb{E}\left[\|f_n^{avg} - f_\rho^*\|_{L^2}^2\right] \lesssim n^{-\frac{s(d+1)}{s(d+1)+d}}.$$

Compared to the minimax lower bound in Proposition 4.8, Theorem 5.4 demonstrates that SGD with averaged iterates can achieve optimality for $\frac{1}{d+1} \le s \le 1$. Unfortunately, in asymptotic settings, SGD with averaged iterates cannot achieve optimality for all mis-specified problems. This is because, in the asymptotic regime, the eigenvalues of the NTK exhibit polynomial decay, making optimization more challenging for $f_\rho^*$ in mis-specified problems. Due to the limitations in SGD's optimization capabilities, optimality can only be achieved within the range $s \ge \frac{1}{d+1}$. To achieve optimality over a broader range of problems, the introduction of other techniques such as momentum (Dieuleveut et al., 2017) or multi-pass strategies (Pillaud-Vivien et al., 2018) is necessary. The differences in generalization performance of SGD in high-dimensional and asymptotic settings highlight the role of overparameterization.

We provide a graphical illustration of Theorem 5.1 and Theorem 5.2 and a summary table of the optimality regions of SGD, KRR, and GD with early stopping in Appendix F.

## 6. Discussion

### 6.1. SGD is Efficient for Well-Specified Problems

As demonstrated in Section 5.1, SGD with exponentially decaying step size achieves optimality for all well-specified problems in both high-dimensional and asymptotic settings. This indicates that SGD with exponentially decaying step size is not subject to the saturation effect encountered by offline algorithms such as KRR. The saturation effect refers to a situation where the algorithm fails to reach optimality when $f_\rho^*$ is sufficiently smooth(Neubauer, 1997; Bauer et al., 2007; Dicker et al., 2017; Cui et al., 2021; Li et al., 2023). Specifically, this occurs when source condition holds with a large value of $s$, the algorithm results in suboptimal performance. For instance, KRR is suboptimal for $s > 1$ in high-dimensional settings (Zhang et al., 2024b) and $s > 2$ in asymptotic settings Li et al. (2023).

The theoretical efficiency of SGD with exponentially decaying step size for well-specified problems arises from the implicit regularization effect induced by the exponentially decaying step size. For well-specified problems, the excess risk of SGD with exponentially decaying step size can be approximated as the sum of the bias and variance terms in the population, which are

$$\mathrm{Bias} \approx \left\| \left(\prod_{i=1}^n (\boldsymbol{I} - \eta_i \boldsymbol{\Sigma})\right) \boldsymbol{\Sigma}^{\frac{1}{2}} (f_0 - f_\rho^*) \right\|_{\mathcal{H}}^2,$$

$$\mathrm{Var} \approx \sigma^2 \sum_{i=1}^n \eta_i^2 \mathrm{tr}\left( \left(\prod_{j=i+1}^n (\boldsymbol{I} - \eta_j \boldsymbol{\Sigma})^2\right) \boldsymbol{\Sigma}^2 \right).$$

In the first phase, a constant step size induces exponential shrinkage of the coefficients along the top eigendirections with large eigenvalues. The subsequent decaying step sizes reduce the variance and ensure that the bias does not increase further. Since the coefficients of $f_\rho^*$ along the eigendirections in the well-specified setting exhibit a decaying trend with respect to the eigenvalues, SGD implicitly balances the remaining bias and the resulting variance, enabling it to achieve optimality. In contrast, the explicit regularization added in offline algorithms cannot eliminate the impact of the coefficients corresponding to the large eigenvalues on the bias. As a result, these algorithms fail to achieve optimality when $f_\rho^*$ is sufficiently smooth. Velikanov et al. (2024) and Lu et al. (2024b) further analyze the offline algorithms constructed using certain analytic filter functions, which are characterized by a qualification $\tau \ge 1$. For high-dimensional settings, the saturation effect of offline algorithms arises when $s \ge \tau$. For asymptotic settings, the saturation effect arises when $s \ge 2\tau$. KRR corresponds to the case where $\tau = 1$ and offline algorithms do not exhibit saturation effects only when $\tau = \infty$, which corresponds to the kernel gradient flow with early stopping. Consequently, when dealing with problems where $f_\rho^*$ is relatively smooth, SGD outperforms most offline algorithms.

It is worth noting that the implicit regularization effect of SGD allows the excess risk convergence analysis to depend solely on the boundedness of the kernel. This contrasts with offline algorithms, which typically require matrix concentration results specific to the kernels in use. In fact, as demonstrated in Proposition B.11 in the appendix, we establish a general convergence rate for the excess risk of SGD with exponentially decaying step sizes, which applies to any bounded kernel. It is straightforward to verify that for any RKHS associated with a bounded kernel whose eigenvalues exhibit polynomial decay, SGD with exponentially decaying step size can achieve optimal performance for well-specified problems. Given this implicit regularization effect exhibited by SGD in well-specified problems, it is reasonable to expect that SGD will also perform effectively when $f_\rho^*$ is a smooth function in a Sobolev space. As discussed in Steinwart & Scovel (2012); Fischer & Steinwart

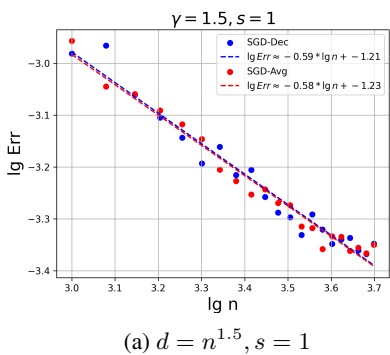
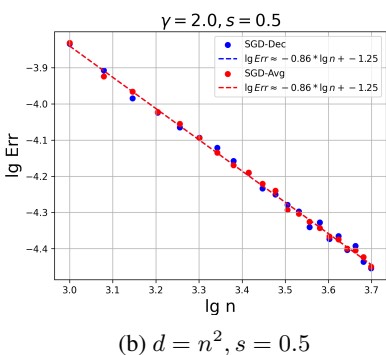
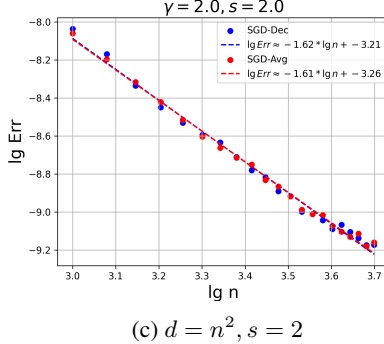

(a) $d = n^{1.5}, s = 1$     (b) $d = n^2, s = 0.5$     (c) $d = n^2, s = 2$

*Figure 1.* Log excess risk decay curves of SGD with the two schedules. The blue curves represent SGD with exponentially decaying step size schedule, and the red curves represent SGD with constant step size and averaged iterates.

(2020), consider RKHS $\mathcal{H}$ as the Sobolev space $H^r(\mathcal{X})$ of order $r > \frac{d}{2}$ where $\mathcal{X} \subseteq \mathbb{R}^d$ is a bounded domain with a smooth boundary. The integral operator of the associated kernel admits eigenvalues that polynomially decay as $\lambda_i \asymp i^{-\frac{2r}{d}}$ (Stone, 1982). Under such spectral decay, SGD is expected to attain the optimal rate $n^{-\frac{2r}{2r+d}}$. Moreover, it holds that $[\mathcal{H}]^s \cong H^{rs}(\mathcal{X})$. Given this equivalence, we expect SGD to exhibit similar regularization benefits for Sobolev smooth functions.

### 6.2. Comparison of Two Types of SGD

**Iterate averaging is effective for mis-specified problems.** We primarily use SGD with averaged iterates for mis-specified problems because we only have Property 1. (c) of the kernel. When analyzing the error of the SGD algorithm using the common bias-variance decomposition, it is essential to estimate the residuals between the actual bias $f_t^b$ and the population-level bias $\tilde{f}_t^b$, where

$$f_t^b = (\mathbf{I} - \eta_t K_{\mathbf{x}_t} \otimes K_{\mathbf{x}_t}) f_{t-1}^b, \ f_0^b = f_\rho^*,$$
$$\tilde{f}_t^b = (\mathbf{I} - \eta_t \boldsymbol{\Sigma}) \tilde{f}_{t-1}^b, \ \tilde{f}_0^b = f_\rho^*.$$

For mis-specified problems, when only Property 1 holds, the error for the residual $\left\| \tilde{f}_t^b - f_t^b \right\|_2^2$ may not converge. Fortunately, the following property from Bach & Moulines (2013) supports the convergence of the residuals when using constant step size and averaged iterates.

**Proposition 6.1.** *Let* $g_t = \tilde{f}_t^b - f_t^b$, $r_t = (K_{\mathbf{x}_t} \otimes K_{\mathbf{x}_t} - \boldsymbol{\Sigma}) g_{t-1}$, *the following inequality holds:*

$$\mathbb{E} \|g_{t-1}\|_{\mathcal{H}}^2 \leq \frac{1}{2\eta(1 - \eta\kappa^2)} \left( \mathbb{E} \|g_{t-1}\|_{L^2}^2 - \mathbb{E} \|g_t\|_{L^2}^2 \right.$$
$$\left. + 2\eta^2 \mathbb{E} \|r_t\|_{L^2}^2 \right),$$

*which implies that* $\mathbb{E} \|\bar{g}_n\|_{\mathcal{H}}^2 \leq \frac{1}{1-\eta\kappa^2} \frac{\eta}{\kappa} \mathbb{E} \left[ \sum_{i=1}^n \|r_i\|_{L^2}^2 \right].$

For the bound of $\|r_i\|_{L^2}^2$, we only require that $\left\| f_\rho^* \right\|_{L^2}^2 < \infty$, which holds for mis-specified problems. This condition enables us to derive the excess risk convergence analysis for SGD in mis-specified problems. Furthermore, when the kernel has additional properties, such as a bounded kurtosis for the projection of $K_{\mathbf{x}}$ onto any function $f$, that is, $\mathbb{E} \langle f, K_{\mathbf{x}} \rangle^4 \leq \kappa \|f\|_{\mathcal{H}}^2$, we can demonstrate that SGD with exponentially decaying step size achieves the same effectiveness in handling mis-specified problems as SGD with averaged iterates.

**Iterate averaging suffers from the saturation effect.** SGD with averaged iterates can handle mis-specified problems under Property 1. However, it faces the saturation effect when $f_\rho^*$ is relatively smooth. We prove that the excess risk of SGD with constant step size and averaged iterates can be bounded from below by

$$\frac{1}{n^2 \eta_0^s} \max_{i \in \mathbb{N}^+} \left\{ (1 - (1 - \lambda_i \eta_0)^n)^2 (\lambda_i \eta_0)^{s-2} \right\} \left\| \boldsymbol{\Sigma}^{\frac{1-s}{2}} f_\rho^* \right\|_{\mathcal{H}}^2$$
$$+ \sigma^2 \left( \frac{1}{16} \frac{k^*}{n} + \frac{1}{64} \sum_{i=k^*+1}^{+\infty} n \eta_0^2 \lambda_i^2 \right),$$

where $k^* = \max \left\{ k : \lambda_k \geq (\eta_0 n)^{-1} \right\}$ often referred to as the effective dimension. This implies that the optimization ability of SGD with averaged iterates with respect to the bias is inherently limited by $\frac{1}{n^2 \eta_0^s}$. When $s$ is large, optimality cannot be achieved. As demonstrated in the Appendix E, we show that in high-dimensional settings, when $s > 1$, there exists a region where SGD with averaged iterates cannot achieve optimality. Furthermore, in asymptotic settings, SGD with averaged iterates fails to achieve optimality for any $s > 2$. This demonstrates that SGD with exponentially decaying step size has an advantage over SGD with averaged iterates when $f_\rho^*$ is relatively smooth.

## 7. Simulations

In this section, we show our experiments with the NTK kernel $\kappa_{\text{NTK}}^1$, which shows that the convergence rate of SGD matches our theoretical result.

The data is generated as follows with a fixed $f_\rho^*$:

$$y_i = f_\rho^*(\mathbf{x}_i) + \epsilon_i, i = 1, \ldots, n,$$

where $x_i$ is i.i.d. sampled from the uniform distribution on sphere $\mathbb{S}^d$, and $\epsilon_i \overset{i.i.d}{\sim} \mathcal{N}(0, 1)$.

In experiment with $s = 1$, we generate the regression function by:

$$f_\rho^*(\mathbf{x}) = K(\mathbf{x}, \mathbf{u}_1) + K(\mathbf{x}, \mathbf{u}_2) + K(\mathbf{x}, \mathbf{u}_3),$$

where each $\mathbf{u}_i$ $(i = 1, 2, 3)$ is i.i.d. sampled from the uniform distribution on sphere $\mathbb{S}^d$.

In experiment with $s \neq 1$, we retain the settings above but replace the regression function with $f_\rho^*(\mathbf{x}) = \mu_2^{\frac{s}{2}} N(d, 2)^{-\frac{1}{2}} P_2(\mathbf{u}, \mathbf{x})$, where $P_2(x) = \frac{dx^2 - 1}{d - 1}$ is the Gegenbauer polynomial, and $\mathbf{u}$ is i.i.d. sampled from the uniform distribution on sphere $\mathbb{S}^d$.

We conduct these experiments to simulate our results under different high-dimensional settings $n \asymp d^\gamma$: (1) $\gamma = 1.5$, $s = 1$, $n$ from 1000 to 2000, with intervals 200, $d = n^{\frac{2}{3}}$; (2) $\gamma = 2$, $s = 0.5$, $n$ from 1000 to 2000, with intervals 200, $d = n^{\frac{1}{2}}$; (3) $\gamma = 1.5$, $s = 2$, $n$ from 1000 to 2000, with intervals 200, $d = n^{\frac{1}{2}}$.

We numerically approximate the excess risk by the empirical excess risk on 1000 i.i.d. sampled data from the uniform distribution on the sphere $\mathbb{S}^d$. As shown in Figure 1, the results support our theoretical findings and indicate that SGD with an exponentially decaying step size does not suffer from the saturation effect.

## Acknowledgements

This work is supported by the NSF China (No.s 92470117 and 62376008).

## Impact Statement

This paper presents work whose goal is to advance the field of Machine Learning. There are many potential societal consequences of our work, none which we feel must be specifically highlighted here.

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

# A. Basic Properties of the Dot-Product Kernel $K$

Recall our notation defined in Section 3.1 and 3.2. This section provides some basic properties of the dot-product kernel $K$.

## A.1. Proof of Property 1

*Proof.* (a) By the boundedness property of continuous functions, and given that $\Phi \in \mathcal{C}^\infty[-1, 1]$, we obtain that there exists a constant $\kappa > 0$ such that

$$\sup_{x \in [0,1]} \Phi(x) < \kappa^2.$$

Since $K(\mathbf{x}, \mathbf{x}) = \Phi(\langle \mathbf{x}, \mathbf{x} \rangle)$ and $\langle \mathbf{x}, \mathbf{x} \rangle \leq 1$ for any $\mathbf{x} \in \mathcal{X}$, we obtain that

$$\sup_{\mathbf{x} \in \mathcal{X}} K(\mathbf{x}, \mathbf{x}) \leq \kappa^2. \tag{3}$$

(b) By the definition of $T$, we have

$$\text{tr}(T) = \sum_{i=1}^\infty \langle \phi_i, T\phi_i \rangle_{L^2} = \sum_{i=1}^\infty \mathbb{E}\left( \lambda_i^{1/2} \phi_i(X) \right)^2 \overset{(i)}{=} \sum_{i=1}^\infty \mathbb{E}\left\langle K_X, \lambda_i^{1/2} \phi_i \right\rangle_{\mathcal{H}}^2 \overset{(ii)}{=} \mathbb{E}\left\langle K_X, K_X \right\rangle_{\mathcal{H}} \overset{(iii)}{=} \mathbb{E}K(X, X) \leq \kappa^2,$$

where (i) and (iii) follow the reproducing property, and (ii) follows from Parseval's identity.

(c) From (a) and $\mathbb{E}[K_\mathbf{x} \otimes K_\mathbf{x}] = \boldsymbol{\Sigma}$, we obtain that

$$\mathbb{E}[K(\mathbf{x}, \mathbf{x}) K_\mathbf{x} \otimes K_\mathbf{x}] \preceq \kappa^2 \mathbb{E}[K_\mathbf{x} \otimes K_\mathbf{x}] = \kappa^2 \boldsymbol{\Sigma}.$$

$\square$

## A.2. Covariance Operator $T$

In this section, we prove that $T|_\mathcal{H} = \boldsymbol{\Sigma}$. Based on this, we can express $\|f\|_{L^2}^2$ by $\langle f, \boldsymbol{\Sigma} f \rangle_\mathcal{H}$ for any $f \in \mathcal{H}$ in subsequent sections.

**Lemma A.1.** *The Reproducing Kernel Hilbert Space $\mathcal{H}$ is a subspace of $L^2$.*

*Proof.* From Property 1(a), and given that $\|f\|_\mathcal{H} < +\infty$ for any $f \in \mathcal{H}$, we have

$$\mathbb{E}\left[ f(\mathbf{x})^2 \right] \overset{(i)}{=} \mathbb{E}[\langle f, K_\mathbf{x} \rangle_\mathcal{H}^2] \overset{(ii)}{\leq} \mathbb{E}[K(\mathbf{x}, \mathbf{x}) \|f\|_\mathcal{H}^2] \overset{(iii)}{\leq} \kappa^2 \|f\|_\mathcal{H}^2 < +\infty,$$

where (i) follows from the reproducing property, (ii) from the Cauchy–Schwarz inequality, and (iii) from Property 1(a), along with $\|f\|_\mathcal{H} < +\infty$. This implies that $\mathcal{H}$ is a subspace of $L^2$.

$\square$

**Lemma A.2.** *The restriction of the covariance operator $T$ to $\mathcal{H}$ coincides with $\boldsymbol{\Sigma}$.*

*Proof.* Recall that $(f \otimes g)h = \langle f, h \rangle_\mathcal{H} g$.

Thus for any function $f \in \mathcal{H}$ and any $\mathbf{z} \in \mathcal{X}$, we have

$$\mathbb{E}[K(\mathbf{x}, \mathbf{z}) f(\mathbf{x})] = \mathbb{E}[\langle K_\mathbf{x}, f \rangle_\mathcal{H} K_\mathbf{x}(\mathbf{z})] = \mathbb{E}[(K_\mathbf{x} \otimes K_\mathbf{x}) f(\mathbf{z})] = (\boldsymbol{\Sigma} f)(\mathbf{z}),$$

where in the first equality, we use the reproducing property of $\mathcal{H}$.

This equation can be equivalently expressed as

$$(\boldsymbol{\Sigma} f)(\mathbf{z}) = \int_\mathcal{X} f(\mathbf{x}) K(\mathbf{x}, \mathbf{z}) \, d\rho_\mathcal{X}(\mathbf{x}),$$

whose form is the same as the definition of $T$.

That is to say, the restriction of $T$ to $\mathcal{H}$ coincides with $\boldsymbol{\Sigma}$.

$\square$

## A.3. Eigenvalue Decay Rate of the Dot-Product Kernel $K$

We borrowed the following lemma from Ghorbani et al. (2021) and Lu et al. (2023):

**Lemma A.3.** *Suppose $n \asymp d^\gamma$ Assumption 4.5 holds. Suppose $p > 0$ is any fixed integer. There exist $C, C_1, C_2, C_3, C_4 > 0$ that depend on $p$, such that for any $d > C$, we have*

$$C_1 d^{-k} \le \mu_k \le C_2 d^{-k}, \qquad k = 0, 1, \dots, p + 1;$$
$$C_3 d^k \le N(d, k) \le C_4 d^k, \quad k = 0, 1, \dots, p + 1.$$

# B. General Bound for SGD with Exponentially Decaying Step Sizes in Well-Specified Problems

## B.1. Bias-Variance Decomposition

Our goal is to minimize the prediction error

$$\mathcal{E}_\rho(f) = \mathbb{E}_\rho[(y - f(\mathbf{x}))^2] \tag{4}$$

using the stochastic gradient oracles. Minimizing the prediction error in (4) is equivalent to minimizing the excess risk, given by

$$\mathcal{E}(f) - \mathcal{E}(f_\rho^*) = \left\| f - f_\rho^* \right\|_{L^2}^2,$$

where $f_\rho^*(\mathbf{x}) = \mathbb{E}_\rho[y|\mathbf{x}]$ denotes the optimal predictor in $L^2$. For well-specified problems, the source condition 4.3 assumes that $f_\rho^* \in [\mathcal{H}]^s$ for $s \ge 1$, which implies $f_\rho^* \in \mathcal{H}$. By Lemma A.2, the excess risk can be equivalently expressed as

$$\mathbb{E}\left[ \left\| f - f_\rho^* \right\|_{L^2}^2 \right] = \mathbb{E}\left[ \left\langle f - f_\rho^*, \boldsymbol{\Sigma}(f - f_\rho^*) \right\rangle_{\mathcal{H}} \right].$$

The iterative update of SGD is given by

$$f_t = f_{t-1} + \eta_t(y_t - f_{t-1}(\mathbf{x}_t))K_{\mathbf{x}_t}, \quad f_0 = 0,$$

where $\eta_t \in \mathbb{R}$ is the step size at iteration $t$, and $(\mathbf{x}_t, y_t)$ is a fresh data from $\rho$.

We write the update rule of $f_t - f_\rho^*$ in the recursive form:

$$f_t - f_\rho^* = (\boldsymbol{I} - \eta_t K_{\mathbf{x}_t} \otimes K_{\mathbf{x}_t})(f_{t-1} - f_\rho^*) + \eta_t \Xi_t,$$

where $\Xi_t = (y_t - f_\rho^*(\mathbf{x}_t))K_{\mathbf{x}_t}$ represents the residual $(\mathbf{x}_t, y)$.

We first apply the commonly used bias-variance decomposition, expressing $f_t - f_\rho^*$ as $f_t - f_\rho^* = f_t^b + f_t^v$. $f_t^b$ stands for the bias term, which demonstrates the decline of the initial error, and $f_t^v$ stands for the variance term, capturing the influence of the noise. Their recursive updates are given by:

$$f_t^b = (\boldsymbol{I} - \eta_t K_{\mathbf{x}_t} \otimes K_{\mathbf{x}_t})f_{t-1}^b, \quad f_0^b = f_0 - f_\rho^*. \tag{5}$$

$$f_t^v = (\boldsymbol{I} - \eta_t K_{\mathbf{x}_t} \otimes K_{\mathbf{x}_t})f_{t-1}^v + \eta_t \Xi_t, \quad f_0^v = 0. \tag{6}$$

Consequently, the excess risk admits a decomposition as stated in Lemma B.1.

**Lemma B.1** (Bias-Variance Decomposition). *For any iteration $t$, we have*

$$\mathbb{E}\left[ \left\langle f_t - f_\rho^*, \boldsymbol{\Sigma}(f_t - f_\rho^*) \right\rangle_{\mathcal{H}} \right] \le 2 \left( \mathbb{E}\left[ \left\langle f_t^b, \boldsymbol{\Sigma} f_t^b \right\rangle_{\mathcal{H}} \right] + \mathbb{E}\left[ \left\langle f_t^v, \boldsymbol{\Sigma} f_t^v \right\rangle_{\mathcal{H}} \right] \right).$$

*Proof.* By Lemma A.2, we have

$$\left\langle f_t^b + f_t^v, \boldsymbol{\Sigma}\left( f_t^b + f_t^v \right) \right\rangle_{\mathcal{H}} = \left\| f_t^b + f_t^v \right\|_{L^2}^2, \quad \left\langle f_t^b, \boldsymbol{\Sigma} f_t^b \right\rangle_{\mathcal{H}} = \left\| f_t^b \right\|_{L^2}^2, \quad \left\langle f_t^v, \boldsymbol{\Sigma} f_t^v \right\rangle_{\mathcal{H}} = \left\| f_t^v \right\|_{L^2}^2.$$

By Minkowski's Inequality, we have

$$\left\| f_t^b + f_t^v \right\|_{L^2}^2 \leq \left( \left( \left\| f_t^b \right\|_{L^2}^2 \right)^{\frac{1}{2}} + \left( \left\| f_t^v \right\|_{L^2}^2 \right)^{\frac{1}{2}} \right)^2 \leq 2 \left( \left\| f_t^b \right\|_{L^2}^2 + \left\| f_t^v \right\|_{L^2}^2 \right).$$

Consequently, by the decomposition $f_t - f_\rho^*$, it follows that

$$\mathbb{E}\left[ \langle f_t - f_\rho^*, \boldsymbol{\Sigma}(f_t - f_\rho^*) \rangle \right]_{\mathcal{H}} = \mathbb{E}\left[ \langle f_t^v + f_t^b, \boldsymbol{\Sigma}(f_t^b + f_t^v) \rangle_{\mathcal{H}} \right]$$
$$\leq 2 \left( \mathbb{E}\left[ \langle f_t^b, \boldsymbol{\Sigma} f_t^b \rangle_{\mathcal{H}} \right] + \mathbb{E}\left[ \langle f_t^v, \boldsymbol{\Sigma} f_t^v \rangle_{\mathcal{H}} \right] \right).$$

$\square$

Then we can bound the excess risk by bounding $\mathbb{E}\left[ \langle f_t^b, \boldsymbol{\Sigma} f_t^b \rangle_{\mathcal{H}} \right]$ and $\mathbb{E}[\langle f_t^v, \boldsymbol{\Sigma} f_t^v \rangle_{\mathcal{H}}]$, respectively.

Our second effort to isolate the stochastic effect is analyzing the concentration effect of the data-dependent operator $K_{\mathbf{x}_t} \otimes K_{\mathbf{x}_t}$ to its population counterpart $\boldsymbol{\Sigma}$.

We decompose the bias $f_t^b$ into two terms: $\tilde{f}_t^b + f_t^{b(1)}$ where $\tilde{f}_t^b$ describes the expected behavior and is recursively defined as

$$\tilde{f}_t^b = (\boldsymbol{I} - \eta_t \boldsymbol{\Sigma}) \tilde{f}_{t-1}^b, \quad \tilde{f}_0^b = f_0 - f_\rho^*;$$

$f_t^{b(1)}$ quantifies the deviation of empirical fluctuations from $\tilde{f}_t^b$. Its recursive form follows as

$$f_t^{b(1)} = f_t^b - \tilde{f}_t^b$$
$$= (\boldsymbol{I} - \eta_t K_{\mathbf{x}_t} \otimes K_{\mathbf{x}_t}) f_{t-1}^b - (\boldsymbol{I} - \eta_t \boldsymbol{\Sigma}) \tilde{f}_{t-1}^b$$
$$= (\boldsymbol{I} - \eta_t K_{\mathbf{x}_t} \otimes K_{\mathbf{x}_t}) f_{t-1}^b - (\boldsymbol{I} - \eta_t \boldsymbol{\Sigma}) \left( f_{t-1}^b - f_{t-1}^{b(1)} \right)$$
$$= (\boldsymbol{I} - \eta_t \boldsymbol{\Sigma}) f_{t-1}^{b(1)} + \eta_t (\boldsymbol{\Sigma} - K_{\mathbf{x}_t} \otimes K_{\mathbf{x}_t}) f_{t-1}^b.$$

Summarizing, we have

$$f_t^{b(1)} = (\boldsymbol{I} - \eta_t \boldsymbol{\Sigma}) f_{t-1}^{b(1)} + \eta_t (\boldsymbol{\Sigma} - K_{\mathbf{x}_t} \otimes K_{\mathbf{x}_t}) f_{t-1}^b, \quad f_0^{b(1)} = 0. \tag{7}$$

Similarly, we split the variance $f_t^v$ into two components: $f_t^v = \tilde{f}_t^v + f_t^{v(1)}$, where $\tilde{f}_t^v$ captures the expected variance dynamics and is recursively defined as:

$$\tilde{f}_t^v = (\boldsymbol{I} - \eta_t \boldsymbol{\Sigma}) \tilde{f}_{t-1}^v + \eta_t \Xi_t, \quad \tilde{f}_0^v = 0.$$

Meanwhile, $f_t^{v(1)}$ accounts for deviations of empirical fluctuations from $\tilde{f}_t^v$, following the recursion

$$f_t^{v(1)} = (\boldsymbol{I} - \eta_t \boldsymbol{\Sigma}) f_{t-1}^{v(1)} + \eta_t (\boldsymbol{\Sigma} - K_{\mathbf{x}_t} \otimes K_{\mathbf{x}_t}) f_{t-1}^v, \quad f_0^{v(1)} = 0. \tag{8}$$

To bound $\mathbb{E}\left[ \langle f_t^b, \boldsymbol{\Sigma} f_t^b \rangle_{\mathcal{H}} \right]$ and $\mathbb{E}[\langle f_t^v, \boldsymbol{\Sigma} f_t^v \rangle_{\mathcal{H}}]$, we have the following lemma:

**Lemma B.2.** *For any iteration $t$, we have*

$$\mathbb{E}\left[ \langle f_t^b, \boldsymbol{\Sigma} f_t^b \rangle_{\mathcal{H}} \right] \leq 2 \left( \left\langle \tilde{f}_t^b, \boldsymbol{\Sigma} \tilde{f}_t^b \right\rangle_{\mathcal{H}} + \mathbb{E}\left[ \left\langle f_t^{b(1)}, \boldsymbol{\Sigma} f_t^{b(1)} \right\rangle_{\mathcal{H}} \right] \right),$$
$$\mathbb{E}[\langle f_t^v, \boldsymbol{\Sigma} f_t^v \rangle_{\mathcal{H}}] \leq 2 \left( \mathbb{E}\left[ \left\langle \tilde{f}_t^v, \boldsymbol{\Sigma} \tilde{f}_t^v \right\rangle_{\mathcal{H}} \right] + \mathbb{E}\left[ \left\langle f_t^{v(1)}, \boldsymbol{\Sigma} f_t^{v(1)} \right\rangle_{\mathcal{H}} \right] \right).$$

We omit the proof of the lemma since it follows a similar argument to that of Lemma B.1.

Therefore, we have

$$\mathbb{E}\left[ \left\| f_t - f_\rho^* \right\|_{L^2}^2 \right]$$
$$\leq 4 \left( \left\langle \tilde{f}_t^b, \boldsymbol{\Sigma} \tilde{f}_t^b \right\rangle_{\mathcal{H}} + \mathbb{E}\left[ \left\langle f_t^{b(1)}, \boldsymbol{\Sigma} f_t^{b(1)} \right\rangle_{\mathcal{H}} \right] + \mathbb{E}\left[ \left\langle \tilde{f}_t^v, \boldsymbol{\Sigma} \tilde{f}_t^v \right\rangle_{\mathcal{H}} \right] + \mathbb{E}\left[ \left\langle f_t^{v(1)}, \boldsymbol{\Sigma} f_t^{v(1)} \right\rangle_{\mathcal{H}} \right] \right).$$

Therefore, we only need to bind these four terms individually at the final iteration $t = n$.

## B.2. Analysis of SGD with Exponentially Decaying Step Size Schedule

In this section, we demonstrate the analysis of SGD with exponentially decaying step size schedule. The exponentially decaying step size schedule takes the form

$$\eta_t = \frac{\eta_0}{2^{\ell-1}}, \quad \text{if } m(\ell-1)+1 \le t \le m\ell, \quad \text{where } m = \left\lceil \frac{n}{\log_2 n} \right\rceil \text{ and } 1 \le \ell \le \lceil \log_2 n \rceil. \tag{9}$$

### B.2.1. POPULATION UPPER BOUND

In this section, we provide the upper bounds for $\left\langle \tilde{f}_n^b, \Sigma \tilde{f}_n^b \right\rangle_{\mathcal{H}}$ and $\mathbb{E}\left[ \left\langle \tilde{f}_n^v, \Sigma \tilde{f}_n^v \right\rangle_{\mathcal{H}} \right]$.

By unrolling the recursive definitions of $\tilde{f}_n^b$ and $\tilde{f}_n^v$, we obtain the following expressions:

$$\tilde{f}_n^b = \left( \prod_{i=1}^{n} (I - \eta_i \Sigma) \right) (f_0 - f_\rho^*);$$

$$\tilde{f}_n^v = \sum_{i=1}^{n} \eta_i \left( \prod_{j=i+1}^{n} (I - \eta_j \Sigma) \right) \Xi_i.$$

The following lemma gives the upper bound of $\left\langle \tilde{f}_n^b, \Sigma \tilde{f}_n^b \right\rangle_{\mathcal{H}}$.

**Lemma B.3.** *Consider SGD with exponentially decaying step size schedule defined in* (9). *Suppose that $\eta_0 \le \frac{1}{\lambda_1}$. Then we have*

$$\left\langle \tilde{f}_n^b, \Sigma \tilde{f}_n^b \right\rangle_{\mathcal{H}} \le \left( \frac{s}{4e} \right)^s \left( \frac{\log_2 n}{n\eta_0} \right)^s \left\| \Sigma^{\frac{1-s}{2}} (f_0 - f_\rho^*) \right\|_{\mathcal{H}}^2.$$

*Proof.* By the definition of $\tilde{f}_n^b$, we have

$$\begin{aligned}
\left\langle \tilde{f}_n^b, \Sigma \tilde{f}_n^b \right\rangle_{\mathcal{H}} &= \left\langle \left( \prod_{i=1}^{n} (I - \eta_i \Sigma) \right) (f_0 - f_\rho^*), \Sigma \left( \prod_{i=1}^{n} (I - \eta_i \Sigma) \right) (f_0 - f_\rho^*) \right\rangle_{\mathcal{H}} \\
&= \left\| \Sigma^{\frac{1}{2}} \left( \prod_{i=1}^{n} (I - \eta_i \Sigma) \right) (f_0 - f_\rho^*) \right\|_{\mathcal{H}}^2.
\end{aligned} \tag{10}$$

Since $\eta_0 \le \frac{1}{\lambda_1}$ and $\eta_i \le \eta_0$, it follows that $0 \preceq I - \eta_i \Sigma \preceq I$. Moreover, for any $i$ and $j$, the operators $I - \eta_i \Sigma$, $I - \eta_j \Sigma$ and $\Sigma$ are positive semidefinite, self-adjoint operators and mutually commuting operators. As a result, the following operator inequality holds:

$$\left( \prod_{i=1}^{n} (I - \eta_i \Sigma) \right) \Sigma \left( \prod_{i=1}^{n} (I - \eta_i \Sigma) \right) \preceq \left( \prod_{i=1}^{m} (I - \eta_0 \Sigma) \right) \Sigma \left( \prod_{i=1}^{m} (I - \eta_0 \Sigma) \right) \preceq (I - \eta_0 \Sigma)^m \Sigma (I - \eta_0 \Sigma)^m. \tag{11}$$

Substituting (11) into (10), we have

$$\begin{aligned}
\left\langle \tilde{f}_n^b, \Sigma \tilde{f}_n^b \right\rangle_{\mathcal{H}} &\le \left\| \Sigma^{\frac{1}{2}} (I - \eta_0 \Sigma)^m (f_0 - f_\rho^*) \right\|_{\mathcal{H}}^2 \\
&\le \frac{1}{\eta_0^s} \left\| (I - \eta_0 \Sigma)^m (\eta_0 \Sigma)^{\frac{s}{2}} \right\|^2 \left\| \Sigma^{\frac{1-s}{2}} (f_0 - f_\rho^*) \right\|_{\mathcal{H}}^2,
\end{aligned} \tag{12}$$

where the second inequality uses the operator norm bound $\|T f\|_{\mathcal{H}} \le \|T\| \|f\|_{\mathcal{H}}$.

To bound the operator norm $\left\| (I - \eta_0 \Sigma)^m (\eta_0 \Sigma)^{\frac{s}{2}} \right\|$, we use the inequality

$$\sup_{0 \le x \le 1} (1 - x)^m x^{\frac{s}{2}} \le \sup_{0 \le x \le 1} \exp(-mx) x^{\frac{s}{2}} \le \left( \frac{s}{2em} \right)^{\frac{s}{2}},$$

which leads to the bound

$$\left\langle \tilde{f}_n^b, \boldsymbol{\Sigma} \tilde{f}_n^b \right\rangle_{\mathcal{H}} \leq \left( \frac{s}{2e} \right)^s \left( \frac{1}{m\eta_0} \right)^s \left\| \boldsymbol{\Sigma}^{\frac{1-s}{2}} (f_0 - f_\rho^*) \right\|_{\mathcal{H}}^2 \leq \left( \frac{s}{4e} \right)^s \left( \frac{\log_2 n}{n\eta_0} \right)^s \left\| \boldsymbol{\Sigma}^{\frac{1-s}{2}} (f_0 - f_\rho^*) \right\|_{\mathcal{H}}^2.$$

$\square$

The following lemma gives the upper bound of $\mathbb{E}\left[ \left\langle \tilde{f}_n^v, \boldsymbol{\Sigma} \tilde{f}_n^v \right\rangle_{\mathcal{H}} \right]$.

**Lemma B.4.** *Consider SGD with exponentially decaying step size schedule defined in* (9)*. Suppose that $\eta_0 \leq \frac{1}{\lambda_1}$. Then, for any integer $k^* \geq 1$, the following bound holds:*

$$\mathbb{E}\left[ \left\langle \tilde{f}_n^v, \boldsymbol{\Sigma} \tilde{f}_n^v \right\rangle_{\mathcal{H}} \right] \leq \sigma^2 \left( \left( \frac{16 \log_2^2 n}{e^2} + \frac{\eta_0^2}{16 \log_2 n} \right) \frac{k^*}{n} + \sum_{k=k^*+1}^{+\infty} n\eta_0^2 \lambda_k^2 \right).$$

*Proof.* Notice that for any $1 \leq i \neq j \leq n$, $\Xi_i$ and $\Xi_j$ are independent. Moreover, under Assumption 4.1, we have $E[\Xi_i] = 0$. Therefore

$$\mathbb{E}[\Xi_i \otimes \Xi_j] = \mathbb{E}[\Xi_i] \otimes \mathbb{E}[\Xi_j] = \mathbf{0}.$$

Consequently, we obtain the following expression for $\mathbb{E}\left[ \left\langle \tilde{f}_n^v, \boldsymbol{\Sigma} \tilde{f}_n^v \right\rangle_{\mathcal{H}} \right]$:

$$\begin{aligned}
\mathbb{E}\left[ \left\langle \tilde{f}_n^v, \boldsymbol{\Sigma} \tilde{f}_n^v \right\rangle_{\mathcal{H}} \right] &= \mathbb{E}\left[ \left\langle \sum_{i=1}^n \eta_i \left( \prod_{j=i+1}^n (\boldsymbol{I} - \eta_j \boldsymbol{\Sigma}) \right) \Xi_i, \boldsymbol{\Sigma} \sum_{i=1}^n \eta_i \left( \prod_{j=i+1}^n (\boldsymbol{I} - \eta_j \boldsymbol{\Sigma}) \right) \Xi_i \right\rangle_{\mathcal{H}} \right] \\
&= \sum_{i=0}^n \eta_i^2 \mathbb{E}\left[ \mathrm{tr}\left( \left( \prod_{j=i+1}^n (\boldsymbol{I} - \eta_j \boldsymbol{\Sigma})^2 \right) \boldsymbol{\Sigma} \Xi_i \otimes \Xi_i \right) \right] \quad (13) \\
&= \sum_{i=0}^n \eta_i^2 \mathrm{tr}\left( \left( \prod_{j=i+1}^n (\boldsymbol{I} - \eta_j \boldsymbol{\Sigma})^2 \right) \boldsymbol{\Sigma} \mathbb{E}[\Xi_i \otimes \Xi_i] \right),
\end{aligned}$$

where the second equality follows from the mutual independence and orthogonality of each $\Xi_i$, and from the commutativity of the self-adjoint operators $\boldsymbol{I} - \eta_j \boldsymbol{\Sigma}$.

Under Assumption 4.1 we have $\mathbb{E}[\Xi_i \otimes \Xi_i] \preceq \sigma^2 \boldsymbol{\Sigma}$, for any $1 \leq i \leq n$. Therefore, we obtain

$$\mathbb{E}\left[ \left\langle \tilde{f}_n^v, \boldsymbol{\Sigma} \tilde{f}_n^v \right\rangle_{\mathcal{H}} \right] \leq \sigma^2 \sum_{i=1}^n \eta_i^2 \mathrm{tr}\left( \left( \prod_{j=i+1}^n (\boldsymbol{I} - \eta_j \boldsymbol{\Sigma})^2 \right) \boldsymbol{\Sigma}^2 \right) = \sum_{k=1}^{+\infty} \sigma^2 \sum_{i=1}^n \eta_i^2 \left( \prod_{j=i+1}^n (1 - \eta_j \lambda_k)^2 \right) \lambda_k^2.$$

We split the summation on the RHS into two parts: $k \leq k^*$ and $k > k^*$. We then bound each term individually by analyzing the contribution of each eigendirection.

For direction $k \leq k^*$, we use the inequality

$$\left( \prod_{j=i+1}^n (1 - \eta_j \lambda_k)^2 \right) \lambda_k^2 \preceq \begin{cases} \left( \prod_{j=i+1}^{i+m} (\boldsymbol{I} - \frac{\eta_i}{2} \lambda_k)^2 \right) \lambda_k^2 \preceq \left( (\boldsymbol{I} - \frac{\eta_i}{2} \lambda_k)^{2m} \right) \lambda_k^2, & \forall 1 \leq i \leq n - m; \\ \lambda_k^2, & \forall n - m < i \leq n. \end{cases}$$

This leads to the bound:

$$\sigma^2 \sum_{i=1}^n \eta_i^2 \left( \prod_{j=i+1}^n (1 - \eta_j \lambda_k)^2 \right) \lambda_k^2 \leq \sigma^2 \sum_{i=1}^{n-m} \left( 1 - \frac{\eta_i}{2} \lambda_k \right)^{2m} (\eta_i \lambda_k)^2 + \sigma^2 \sum_{i=n-m+1}^{n-1} \eta_i^2 \lambda_k^2.$$

For direction $k > k^*$, we simply use the bound:

$$\sigma^2 \sum_{i=1}^{n} \eta_i^2 \left( \prod_{j=i+1}^{n} (1 - \eta_j \lambda_k)^2 \right) \lambda_k^2 \leq \sigma^2 \sum_{k=k^*+1}^{+\infty} n\eta_0^2 \lambda_k^2.$$

Combining the two bounds, we obtain

$$\mathbb{E}\left[ \left\langle \tilde{f}_n^v, \boldsymbol{\Sigma} \tilde{f}_n^v \right\rangle_{\mathcal{H}} \right] \leq \sigma^2 \sum_{k=1}^{k^*} \left( \sum_{j=1}^{n-m} \left( 1 - \frac{\eta_j}{2} \lambda_k \right)^{2m} (\eta_j \lambda_k)^2 + \sum_{j=n-m+1}^{n} \eta_j^2 \lambda_k^2 \right) + \sigma^2 \sum_{k=k^*+1}^{+\infty} n\eta_j^2 \lambda_k^2. \tag{14}$$

For $j > n - m$, we note that

$$\eta_j \leq \frac{\eta_0}{2^{\left\lfloor \frac{n-m}{m} \right\rfloor}} \leq \frac{\eta_0}{2^{\frac{n}{m}-2}} \leq \frac{\eta_0}{2^{\log_2 n - 2}} \leq \frac{\eta_0}{4n}. \tag{15}$$

For $j \leq n - m$, we use the following inequality to bound $\left( 1 - \frac{\eta_j}{2} \lambda_k \right)^{2m} (\eta_j \lambda_k)^2$:

$$\sup_{0 \leq x \leq 1} \left( 1 - \frac{x}{2} \right)^{2m} x^2 \leq \sup_{0 \leq x \leq 1} \exp(-mx) x^2 = \left( \frac{2}{em} \right)^2. \tag{16}$$

Finally, substituting (15) and (16) into (14), we have

$$\begin{aligned}
\mathbb{E}\left[ \left\langle \tilde{f}_n^v, \boldsymbol{\Sigma} \tilde{f}_n^v \right\rangle_{\mathcal{H}} \right] &\leq \sigma^2 \sum_{k=1}^{k^*} \left( (n-m) \left( \frac{2}{em} \right)^2 + m \left( \frac{\eta_0}{4n} \right)^2 \right) + \sigma^2 \sum_{k=k^*+1}^{+\infty} n\eta_0^2 \lambda_k^2 \\
&\leq \sigma^2 k^* \left( n \left( \frac{4 \log_2 n}{en} \right)^2 + \frac{n}{\log_2 n} \left( \frac{\eta_0}{4n} \right)^2 \right) + \sigma^2 \sum_{k=k^*+1}^{+\infty} n\eta_0^2 \lambda_k^2 \\
&= \sigma^2 \left( \left( \frac{16 \log_2^2 n}{e^2} + \frac{\eta_0^2}{16 \log_2 n} \right) \frac{k^*}{n} + \sum_{k=k^*+1}^{+\infty} n\eta_0^2 \lambda_k^2 \right).
\end{aligned}$$

$\square$

### B.2.2. Effect of Replacing $K_{\mathbf{x}} \otimes K_{\mathbf{x}}$ with $\boldsymbol{\Sigma}$

In this section, we incorporate the operator $K_{\mathbf{x}} \otimes K_{\mathbf{x}}$ into the analysis, and provide upper bounds for $\mathbb{E}\left[ \left\langle f_n^{b(1)}, \boldsymbol{\Sigma} f_n^{b(1)} \right\rangle_{\mathcal{H}} \right]$ and $\mathbb{E}\left[ \left\langle f_n^{v(1)}, \boldsymbol{\Sigma} f_n^{v(1)} \right\rangle_{\mathcal{H}} \right]$. These results will be used to provide the upper bounds of the bias term $\mathbb{E}\left[ \left\langle f_n^b, \boldsymbol{\Sigma} f_n^b \right\rangle_{\mathcal{H}} \right]$ and the variance term $\mathbb{E}\left[ \left\langle f_n^v, \boldsymbol{\Sigma} f_n^v \right\rangle_{\mathcal{H}} \right]$.

Recalling the recursive definitions in (7) and (8), and solving the resulting expressions of $f_t^{b(1)}$ and $f_t^{v(1)}$, we obtain

$$f_n^{b(1)} = \sum_{i=1}^{n} \eta_i \left( \prod_{j=i+1}^{n} (\boldsymbol{I} - \eta_j \boldsymbol{\Sigma}) \right) \Xi_i^{b(1)}, \quad f_n^{v(1)} = \sum_{i=1}^{n} \eta_i \left( \prod_{j=i+1}^{n} (\boldsymbol{I} - \eta_j \boldsymbol{\Sigma}) \right) \Xi_i^{v(1)},$$

where

$$\Xi_i^{b(1)} = \left( \boldsymbol{\Sigma} - K_{x_i} \otimes K_{x_i} \right) f_{i-1}^b, \quad \Xi_i^{v(1)} = \left( \boldsymbol{\Sigma} - K_{x_i} \otimes K_{x_i} \right) f_{i-1}^v.$$

To bound the quantities $\mathbb{E}\left[ \left\langle f_n^{b(1)}, \boldsymbol{\Sigma} f_n^{b(1)} \right\rangle_{\mathcal{H}} \right]$ and $\mathbb{E}\left[ \left\langle f_n^{v(1)}, \boldsymbol{\Sigma} f_n^{v(1)} \right\rangle_{\mathcal{H}} \right]$. We derive upper bounds for the operators $\mathbb{E}\left[ \Xi_i^{b(1)} \otimes \Xi_i^{b(1)} \right]$ and $\mathbb{E}\left[ \Xi_i^{v(1)} \otimes \Xi_i^{v(1)} \right]$, as stated in Lemma B.6 and Lemma B.8.

The following lemma provides a key auxiliary result used in proving Lemma B.6. It shows that $\mathbb{E}\left[ \|f_t^b\|_{\mathcal{H}}^2 \right]$ can be uniformly bounded by the initial error $\|f_0 - f_\rho^*\|_{\mathcal{H}}^2$ throughout the iteration.

**Lemma B.5.** *Consider SGD with exponentially decaying step size schedule defined in* (9). *Suppose that* $\eta_0 \leq \frac{2}{\kappa^2}$. *Then for any iteration* $t$, *we have*

$$\mathbb{E}\left[\|f_t^b\|_{\mathcal{H}}^2\right] \leq \|f_0 - f_\rho^*\|_{\mathcal{H}}^2.$$

*Proof.* Firstly, it is obvious that $\mathbb{E}\left[\|f_0^b\|_{\mathcal{H}}^2\right] \leq \|f_0 - f_\rho^*\|_{\mathcal{H}}^2$.

Notice that for any $1 \leq t \leq n$, $K_{\mathbf{x}_t}$ and $f_{t-1}^b$ are independent, then for any $1 \leq t \leq n$, we have

$$
\begin{aligned}
\mathbb{E}\left[\|f_t^b\|_{\mathcal{H}}^2\right] &= \mathbb{E}\left[\|(\boldsymbol{I} - \eta_t K_{\mathbf{x}_t} \otimes K_{\mathbf{x}_t})f_{t-1}^b\|_{\mathcal{H}}^2\right] \\
&= \mathbb{E}\left[\|f_{t-1}^b\|_{\mathcal{H}}^2\right] - 2\eta_t \mathbb{E}\left[\langle f_{t-1}^b, (K_{\mathbf{x}_t} \otimes K_{\mathbf{x}_t})f_{t-1}^b\rangle_{\mathcal{H}}\right] + \eta_t^2 \mathbb{E}\left[\langle (K_{\mathbf{x}_t} \otimes K_{\mathbf{x}_t})f_{t-1}^b, (K_{\mathbf{x}_t} \otimes K_{\mathbf{x}_t})f_{t-1}^b\rangle_{\mathcal{H}}\right] \\
&= \mathbb{E}\left[\|f_{t-1}^b\|_{\mathcal{H}}^2\right] - 2\eta_t \mathbb{E}\left[\langle f_{t-1}^b, \mathbb{E}\left[(K_{\mathbf{x}_t} \otimes K_{\mathbf{x}_t})\right] f_{t-1}^b\rangle_{\mathcal{H}}\right] + \eta_t^2 \mathbb{E}\left[\langle f_{t-1}^b, \mathbb{E}\left[K(\mathbf{x}_t, \mathbf{x}_t)K_{\mathbf{x}_t} \otimes K_{\mathbf{x}_t}\right] f_{t-1}^b\rangle_{\mathcal{H}}\right] \\
&\leq \mathbb{E}\left[\|f_{t-1}^b\|_{\mathcal{H}}^2\right] + \left(-2\eta_t + \eta_t^2 \kappa^2\right)\mathbb{E}\left[\langle f_{t-1}^b, \boldsymbol{\Sigma} f_{t-1}^b\rangle_{\mathcal{H}}\right],
\end{aligned}
$$

where in the inequality, we use $\mathbb{E}\left[K_{\mathbf{x}_t} \otimes K_{\mathbf{x}_t}\right] = \boldsymbol{\Sigma}$ and $\mathbb{E}\left[K(\mathbf{x}_t, \mathbf{x}_t)K_{\mathbf{x}_t} \otimes K_{\mathbf{x}_t}\right] \preceq \kappa^2 \boldsymbol{\Sigma}$.

Since the function $g(x) = -2x + x^2\kappa^2$ is non-positive for $0 \leq x \leq \frac{2}{\kappa^2}$, we have $-2\eta_t + \eta_t^2 \kappa^2 \leq 0$.

Therefore, for any $1 \leq t \leq n$, we have

$$\mathbb{E}\left[\|f_t^b\|_{\mathcal{H}}^2\right] \leq \mathbb{E}[\|f_{t-1}^b\|_{\mathcal{H}}^2].$$

By recursion, for any $0 \leq t \leq n$, we have

$$\mathbb{E}\left[\|f_t^b\|_{\mathcal{H}}^2\right] \leq \|f_0 - f_\rho^*\|_{\mathcal{H}}^2.$$

$\square$

The following lemma gives the upper bound of $\mathbb{E}\left[\Xi_i^{b(1)} \otimes \Xi_i^{b(1)}\right]$.

**Lemma B.6.** *Consider SGD with exponentially decaying step size schedule defined in* (9). *Suppose that* $\eta_0 \leq \frac{2}{\kappa^2}$. *Then for any iteration* $t$, *we have*

$$\mathbb{E}\left[\Xi_t^{b(1)} \otimes \Xi_t^{b(1)}\right] \preceq \|f_0 - f_\rho^*\|_{\mathcal{H}}^2 \kappa^2 \boldsymbol{\Sigma}.$$

*Proof.* Notice that for any $1 \leq t \leq n$, $K_{\mathbf{x}_t}$ and $f_{t-1}^b$ are independent, then we have

$$
\begin{aligned}
\mathbb{E}\left[\Xi_t^{b(1)} \otimes \Xi_t^{b(1)}\right] &= \mathbb{E}[(\boldsymbol{\Sigma} - K_{\mathbf{x}_t} \otimes K_{\mathbf{x}_t})(f_{t-1}^b \otimes f_{t-1}^b)(\boldsymbol{\Sigma} - K_{\mathbf{x}_t} \otimes K_{\mathbf{x}_t})] \\
&= \mathbb{E}\left[(\boldsymbol{\Sigma} - K_{\mathbf{x}_t} \otimes K_{\mathbf{x}_t})\mathbb{E}\left[(f_{t-1}^b \otimes f_{t-1}^b)\right](\boldsymbol{\Sigma} - K_{\mathbf{x}_t} \otimes K_{\mathbf{x}_t})\right] \\
&\preceq \mathbb{E}\left[\|f_{t-1}^b \otimes f_{t-1}^b\|\right]\mathbb{E}\left[\boldsymbol{\Sigma}^2 - \boldsymbol{\Sigma}K_{\mathbf{x}_t} \otimes K_{\mathbf{x}_t} - K_{\mathbf{x}_t} \otimes K_{\mathbf{x}_t}\boldsymbol{\Sigma} + K(\mathbf{x}_t, \mathbf{x}_t)K_{\mathbf{x}_t} \otimes K_{\mathbf{x}_t}\right] \\
&= \mathbb{E}\left[\|f_{t-1}^b\|_{\mathcal{H}}^2\right]\left(\mathbb{E}\left[K(\mathbf{x}_t, \mathbf{x}_t)K_{\mathbf{x}_t} \otimes K_{\mathbf{x}_t}\right] - \boldsymbol{\Sigma}^2\right) \\
&\preceq \|f_0 - f_\rho^*\|_{\mathcal{H}}^2(\kappa^2 \boldsymbol{\Sigma} - \boldsymbol{\Sigma}^2) \\
&\preceq \|f_0 - f_\rho^*\|_{\mathcal{H}}^2 \kappa^2 \boldsymbol{\Sigma}.
\end{aligned}
$$

$\square$

The following lemma provides a key auxiliary result used in proving Lemma B.8. It shows that the operator $\mathbb{E}[f_t^v \otimes f_t^v]$ can be uniformly bounded by $\alpha \boldsymbol{I}$ with a constant $\alpha$ throughout the iteration.

**Lemma B.7.** *Consider SGD with exponentially decaying step size schedule defined in* (9). *Suppose that* $\eta_0 < \frac{2}{\kappa^2}$. *Then for any iteration* $t$, *we have*

$$\mathbb{E}[f_t^v \otimes f_t^v] \preceq \alpha \boldsymbol{I}, \text{ where } \alpha = \frac{\eta_0 \sigma^2}{2 - \eta_0 \kappa^2}.$$

*Proof.* We prove this by induction.

For the base case, $\mathbb{E}[f_0^v \otimes f_0^v] = 0 \otimes 0 \preceq \alpha \boldsymbol{I}$ naturally holds.

Suppose that $\mathbb{E}[f_{t-1}^v \otimes f_{t-1}^v] \preceq \alpha \boldsymbol{I}$. We aim to show that the same inequality holds for $t$.

Notice that for any $1 \leq t \leq n$, $K_{\mathbf{x}_t}$ and $f_{t-1}^v$ are independent, then we have

$$
\begin{aligned}
\mathbb{E}\left[f_t^v \otimes f_t^v\right] &= \mathbb{E}\left[(\boldsymbol{I} - \eta_t K_{\mathbf{x}_t} \otimes K_{\mathbf{x}_t})(f_{t-1}^v \otimes f_{t-1}^v)(\boldsymbol{I} - \eta_t K_{\mathbf{x}_t} \otimes K_{\mathbf{x}_t}) + \eta_t^2 \Xi_t \otimes \Xi_t\right] \\
&= \mathbb{E}\left[(\boldsymbol{I} - \eta_t K_{\mathbf{x}_t} \otimes K_{\mathbf{x}_t})\mathbb{E}\left[(f_{t-1}^v \otimes f_{t-1}^v)\right](\boldsymbol{I} - \eta_t K_{\mathbf{x}_t} \otimes K_{\mathbf{x}_t}) + \eta_t^2 \Xi_t \otimes \Xi_t\right] \\
&\preceq \mathbb{E}\left[(\boldsymbol{I} - \eta_t K_{\mathbf{x}_t} \otimes K_{\mathbf{x}_t})\alpha \boldsymbol{I}(\boldsymbol{I} - \eta_t K_{\mathbf{x}_t} \otimes K_{\mathbf{x}_t}) + \eta_t^2 \Xi_t \otimes \Xi_t\right] \\
&= \alpha \mathbb{E}\left[\boldsymbol{I} - 2\eta_t K_{\mathbf{x}_t} \otimes K_{\mathbf{x}_t} + \eta_t^2 K(\mathbf{x}_t, \mathbf{x}_t) K_{\mathbf{x}_t} \otimes K_{\mathbf{x}_t}\right] + \eta_t^2 \mathbb{E}\left[\Xi_t \otimes \Xi_t\right].
\end{aligned}
$$

Using $\mathbb{E}\left[K_{\mathbf{x}_t} \otimes K_{\mathbf{x}_t}\right] = \boldsymbol{\Sigma}$, $\mathbb{E}\left[\Xi_t \otimes \Xi_t\right] \preceq \sigma^2 \boldsymbol{\Sigma}$ and $\mathbb{E}[K(\mathbf{x}_t, \mathbf{x}_t) K_{\mathbf{x}_t} \otimes K_{\mathbf{x}_t}] \preceq \kappa^2 \boldsymbol{\Sigma}$, we have

$$
\mathbb{E}\left[f_t^v \otimes f_t^v\right] \preceq \alpha(\boldsymbol{I} - 2\eta_t \boldsymbol{\Sigma} + \eta_t^2 \kappa^2 \boldsymbol{\Sigma}) + \eta_t^2 \sigma^2 \boldsymbol{\Sigma}
$$

$$
\preceq \alpha I + \eta_t^2 (\sigma^2 - \alpha \frac{2 - \eta_t \kappa^2}{\eta_t}) \boldsymbol{\Sigma}.
$$

Since the function $g(x) = \frac{2 - x\kappa^2}{x}$ is monotonically decreasing, we have

$$
\sigma^2 - \alpha \frac{2 - \eta_t \kappa^2}{\eta_t} \leq \sigma^2 - \alpha \frac{2 - \eta_0 \kappa^2}{\eta_0} = \sigma^2 - \alpha \frac{\sigma^2}{\alpha} = 0.
$$

Therefore, we conclude that

$$
\mathbb{E}\left[f_t^v \otimes f_t^v\right] \preceq \alpha I + \eta_t^2 (\sigma^2 - \alpha \frac{2 - \eta_0 \kappa^2}{\eta_0}) \boldsymbol{\Sigma} \preceq \alpha \boldsymbol{I},
$$

completing the inductive step. This proves the lemma.

$\square$

The following lemma gives the upper bound of $\mathbb{E}\left[\Xi_t^{v(1)} \otimes \Xi_t^{v(1)}\right]$.

**Lemma B.8.** *Consider SGD with exponentially decaying step size schedule defined in* (9)*. Suppose that $\eta_0 < \frac{2}{\kappa^2}$. Then for any iteration $t$, we have*

$$
\mathbb{E}\left[\Xi_t^{v(1)} \otimes \Xi_t^{v(1)}\right] \preceq \alpha \kappa^2 \boldsymbol{\Sigma}, \text{ where } \alpha = \frac{\eta_0 \sigma^2}{2 - \eta_0 \kappa^2}.
$$

*Proof.* Notice that for any $1 \leq t \leq n$, $K_{\mathbf{x}_t}$ and $f_{t-1}^v$ are independent, then we have

$$
\begin{aligned}
\mathbb{E}\left[\Xi_t^{v(1)} \otimes \Xi_t^{v(1)}\right] &= \mathbb{E}[(\boldsymbol{\Sigma} - K_{\mathbf{x}_t} \otimes K_{\mathbf{x}_t})(f_{t-1}^v \otimes f_{t-1}^v)(\boldsymbol{\Sigma} - K_{\mathbf{x}_t} \otimes K_{\mathbf{x}_t})] \\
&= \mathbb{E}[(\boldsymbol{\Sigma} - K_{\mathbf{x}_t} \otimes K_{\mathbf{x}_t})\mathbb{E}\left[(f_{t-1}^v \otimes f_{t-1}^v)\right](\boldsymbol{\Sigma} - K_{\mathbf{x}_t} \otimes K_{\mathbf{x}_t})] \\
&\preceq \mathbb{E}[(\boldsymbol{\Sigma} - K_{\mathbf{x}_t} \otimes K_{\mathbf{x}_t})\alpha \boldsymbol{I}(\boldsymbol{\Sigma} - K_{\mathbf{x}_t} \otimes K_{\mathbf{x}_t})] \\
&= \alpha \mathbb{E}[\boldsymbol{\Sigma}^2 - \boldsymbol{\Sigma} K_{\mathbf{x}_t} \otimes K_{\mathbf{x}_t} - K_{\mathbf{x}_t} \otimes K_{\mathbf{x}_t} \boldsymbol{\Sigma} + K(\mathbf{x}_t, \mathbf{x}_t) K_{\mathbf{x}_t} \otimes K_{\mathbf{x}_t}] \\
&\preceq \alpha(\kappa^2 \boldsymbol{\Sigma} - \boldsymbol{\Sigma}^2) \\
&\preceq \alpha \kappa^2 \boldsymbol{\Sigma}.
\end{aligned}
$$

$\square$

Then we can bound $\mathbb{E}\left[\left\langle f_n^{b(1)}, \boldsymbol{\Sigma} f_n^{b(1)}\right\rangle_{\mathcal{H}}\right]$ and $\mathbb{E}\left[\left\langle f_n^{v(1)}, \boldsymbol{\Sigma} f_n^{v(1)}\right\rangle_{\mathcal{H}}\right]$ in a manner similar to Lemma B.4.

We have the following results:

**Lemma B.9** (Residual Bias Upper Bound)**.** *Consider SGD with exponentially decaying step size schedule defined in* (9)*. Suppose that $\eta_0 \leq \min\left\{\frac{2}{\kappa^2}, \frac{1}{\lambda_1}\right\}$. Then, for any integer $k^* \geq 1$, we have*

$$
\mathbb{E}\left[\left\langle f_n^{b(1)}, \boldsymbol{\Sigma} f_n^{b(1)}\right\rangle_{\mathcal{H}}\right] \leq \|f_0 - f_\rho^*\|_{\mathcal{H}}^2 \kappa^2 \left(\left(\frac{16 \log_2^2 n}{e^2} + \frac{\eta_0^2}{16 \log_2 n}\right) \frac{k^*}{n} + \sum_{k=k^*+1}^{+\infty} n\eta_0^2 \lambda_k^2\right).
$$

*Proof.* Notice that for any $1 \leq t \leq n$, $K_{\mathbf{x}_t}$ and $f_{t-1}^b$ are independent. Under Assumption 4.1, we have $E[\Xi_i] = 0$. Thus when $i < j$, we have

$$
\begin{aligned}
\mathbb{E}\left[\Xi_i^{b(1)} \otimes \Xi_j^{b(1)}\right] &= \mathbb{E}\left[(\boldsymbol{\Sigma} - K_{\mathbf{x}_i} \otimes K_{\mathbf{x}_i})\left(f_{i-1}^b \otimes f_{j-1}^b\right)(\boldsymbol{\Sigma} - K_{\mathbf{x}_j} \otimes K_{\mathbf{x}_j})\right] \\
&= \mathbb{E}\left[(\boldsymbol{\Sigma} - K_{\mathbf{x}_i} \otimes K_{\mathbf{x}_i})\left(f_{i-1}^b \otimes f_{j-1}^b\right)\right]\mathbb{E}\left[(\boldsymbol{\Sigma} - K_{\mathbf{x}_j} \otimes K_{\mathbf{x}_j})\right] \\
&= \mathbf{0}.
\end{aligned}
$$

Therefore, according to the proof of Lemma B.4, we have

$$
\mathbb{E}\left[\left\langle f_n^{v(1)}, \boldsymbol{\Sigma} f_n^{v(1)}\right\rangle_{\mathcal{H}}\right] \leq \|f_0 - f_\rho^*\|_{\mathcal{H}}^2 \kappa^2 \left(\left(\frac{16 \log_2^2 n}{e^2} + \frac{\eta_0^2}{16 \log_2 n}\right)\frac{k^*}{n} + \sum_{k=k^*+1}^{+\infty} n\eta_0^2 \lambda_k^2\right).
$$

$\square$

**Lemma B.10** (Residual Variance Upper Bound). *Consider SGD with exponentially decaying step size schedule defined in* (9). *Suppose that* $\eta_0 \leq \min\left\{\frac{2}{\kappa^2}, \frac{1}{\lambda_1}\right\}$. *Then, for any integer* $k^* \geq 1$, *we have*

$$
\mathbb{E}\left[\left\langle f_n^{v(1)}, \boldsymbol{\Sigma} f_n^{v(1)}\right\rangle_{\mathcal{H}}\right] \leq \frac{\eta_0 \sigma^2}{2 - \eta_0 \kappa^2}\kappa^2 \left(\left(\frac{16 \log_2^2 n}{e^2} + \frac{\eta_0^2}{16 \log_2 n}\right)\frac{k^*}{n} + \sum_{k=k^*+1}^{+\infty} n\eta_0^2 \lambda_k^2\right).
$$

*Proof.* Notice that for any $1 \leq t \leq n$, $K_{\mathbf{x}_t}$ and $f_{t-1}^b$ are independent. Under Assumption 4.1, we have $E[\Xi_i] = 0$. Thus, when $i < j$, we have

$$
\begin{aligned}
\mathbb{E}\left[\Xi_i^{v(1)} \otimes \Xi_j^{v(1)}\right] &= \mathbb{E}\left[(\boldsymbol{\Sigma} - K_{\mathbf{x}_i} \otimes K_{\mathbf{x}_i})\left(f_{i-1}^v \otimes f_{j-1}^v\right)(\boldsymbol{\Sigma} - K_{\mathbf{x}_j} \otimes K_{\mathbf{x}_j})\right] \\
&= \mathbb{E}\left[(\boldsymbol{\Sigma} - K_{\mathbf{x}_i} \otimes K_{\mathbf{x}_i})\left(f_{i-1}^v \otimes f_{j-1}^v\right)\right]\mathbb{E}\left[(\boldsymbol{\Sigma} - K_{\mathbf{x}_j} \otimes K_{\mathbf{x}_j})\right] \\
&= \mathbf{0}.
\end{aligned}
$$

Therefore, according to the proof of Lemma B.4, we have

$$
\mathbb{E}\left[\left\langle f_n^{v(1)}, \boldsymbol{\Sigma} f_n^{v(1)}\right\rangle_{\mathcal{H}}\right] \leq \frac{\eta_0 \sigma^2}{2 - \eta_0 \kappa^2}\kappa^2 \left(\left(\frac{16 \log_2^2 n}{e^2} + \frac{\eta_0^2}{16 \log_2 n}\right)\frac{k^*}{n} + \sum_{k=k^*+1}^{+\infty} n\eta_0^2 \lambda_k^2\right).
$$

$\square$

Combining Lemma B.1, B.2, B.3, B.4, B.9 and B.10, we have the following proposition:

**Proposition B.11** (Upper Bound for SGD with Exponentially Decaying Step Size Schedule). *Consider SGD with exponentially decaying step size schedule defined in* (9). *Suppose that* $\eta_0 \leq \min\left\{\frac{2}{\kappa^2}, \frac{1}{\lambda_1}\right\}$. *Then, for any integer* $k^* \geq 1$, *we have*

$$
\begin{aligned}
\mathbb{E}\left[\|f_n - f_\rho^*\|_{L^2}^2\right] \leq{}& 4\left(\frac{s}{4e}\right)^s \left(\frac{\log_2 n}{n\eta_0}\right)^s \left\|\boldsymbol{\Sigma}^{\frac{1-s}{2}}(f_0 - f_\rho^*)\right\|_{\mathcal{H}}^2 + 4\sigma^2 \left(\left(\frac{16 \log_2^2 n}{e^2} + \frac{\eta_0^2}{16 \log_2 n}\right)\frac{k^*}{n} + \sum_{k=k^*+1}^{+\infty} n\eta_0^2 \lambda_k^2\right) \\
&+ 4\left(\|f_0 - f_\rho^*\|_{\mathcal{H}}^2 \kappa^2 + \frac{\eta_0 \sigma^2 \kappa^2}{2 - \eta_0 \kappa^2}\right)\left(\left(\frac{16 \log_2^2 n}{e^2} + \frac{\eta_0^2}{16 \log_2 n}\right)\frac{k^*}{n} + \sum_{k=k^*+1}^{+\infty} n\eta_0^2 \lambda_k^2\right).
\end{aligned}
$$

## C. General Bound for SGD with Averaged Iterates

In this section, we demonstrate the analysis of SGD with averaged iterates, which is:

$$\forall \, 0 \le t < n, \quad \eta_t = \eta_0, \quad \text{with } \overline{f_n} = \frac{1}{n} \sum_{t=0}^{n-1} f_t.$$

In this section, we need to address the misspecified setting, where $f_\rho^*$ may not lie in $\mathcal{H}$. Specifically, we consider the case where the source condition 4.3 holds with some $0 < s \le 1$. To accommodate this broader setting, we extend the operator framework developed in Appendix B.1 to act on functions outside of $\mathcal{H}$. Following Dieuleveut & Bach (2016), define $\widetilde{K_{\mathbf{x}} \otimes K_{\mathbf{x}}} : L^2 \to \mathcal{H}$ by $\widetilde{K_{\mathbf{x}} \otimes K_{\mathbf{x}}} \circ f = f(\mathbf{x}) K_{\mathbf{x}}$. This extension satisfies that $\widetilde{K_{\mathbf{x}} \otimes K_{\mathbf{x}}}|_{\mathcal{H}} = K_{\mathbf{x}} \otimes K_{\mathbf{x}}$ and $\mathbb{E}\left[\widetilde{K_{\mathbf{x}} \otimes K_{\mathbf{x}}}\right] = T$. For notational simplicity, we will continue to write $K_{\mathbf{x}} \otimes K_{\mathbf{x}}$ instead of $\widetilde{K_{\mathbf{x}} \otimes K_{\mathbf{x}}}$ and denote its expectation by $\boldsymbol{\Sigma}$. Additionally, we denote $\langle f, \boldsymbol{\Sigma} f \rangle_{\mathcal{H}}$ by $\left\| \boldsymbol{\Sigma}^{\frac{1}{2}} f \right\|_{\mathcal{H}}^2$ in the following. With these conventions, the notation used in this section remains fully consistent with that of Appendix B.1.

For convenience, we defined

$$\overline{f_n^b} = \frac{1}{n} \sum_{t=0}^{n-1} f_t^b, \;\; \overline{f_n^v} = \frac{1}{n} \sum_{t=0}^{n-1} f_t^v; \;\; \overline{\tilde{f}_n^b} = \frac{1}{n} \sum_{t=0}^{n-1} \tilde{f}_t^b, \;\; \overline{\tilde{f}_n^v} = \frac{1}{n} \sum_{t=0}^{n-1} \tilde{f}_t^v; \;\; \overline{f_n^{b(1)}} = \frac{1}{n} \sum_{t=0}^{n-1} f_t^{b(1)}, \;\; \overline{f_n^{v(1)}} = \frac{1}{n} \sum_{t=0}^{n-1} f_t^{v(1)}.$$

By Lemma B.1, B.2 and Minkovski inequality, we have

$$\mathbb{E}\left[ \left\langle \overline{f_n} - f_\rho^*, \boldsymbol{\Sigma} \left( \overline{f_n} - f_\rho^* \right) \right\rangle_{\mathcal{H}} \right]$$
$$\le 4 \left( \left\langle \overline{\tilde{f}_n^b}, \boldsymbol{\Sigma} \overline{\tilde{f}_n^b} \right\rangle + \mathbb{E}\left[ \left\langle \overline{\tilde{f}_n^v}, \boldsymbol{\Sigma} \overline{\tilde{f}_n^v} \right\rangle \right] + \mathbb{E}\left[ \left\langle \overline{f_n^{b(1)}}, \boldsymbol{\Sigma} \overline{f_n^{b(1)}} \right\rangle \right] + \mathbb{E}\left[ \left\langle \overline{f_n^{v(1)}}, \boldsymbol{\Sigma} \overline{f_n^{v(1)}} \right\rangle \right] \right),$$

which means we only need to bound these four terms individually.

The following analysis is based on Dieuleveut & Bach (2016), which provides the following results:

**Lemma C.1** (Population Bias Upper Bound). *Consider SGD with averaged iterates defined in* (17). *Suppose that $\eta_0 \le \frac{1}{\lambda_1}$. Then we have*

$$\left\langle \overline{\tilde{f}_n^b}, \boldsymbol{\Sigma} \overline{\tilde{f}_n^b} \right\rangle_{\mathcal{H}} = \frac{1}{n^2} \left\| \sum_{i=0}^{n-1} (\boldsymbol{I} - \eta_0 \boldsymbol{\Sigma})^i \boldsymbol{\Sigma}^{\frac{1}{2}} (f_0 - f_\rho^*) \right\|_{\mathcal{H}}^2 \le \frac{1}{n^{\min\{s,2\}} \eta_0^s} \left\| \boldsymbol{\Sigma}^{\frac{1-s}{2}} (f_0 - f_\rho^*) \right\|_{\mathcal{H}}^2. \tag{17}$$

**Lemma C.2** (Residual Bias Upper Bound). *Consider SGD with averaged iterates defined in* (17). *Suppose that $\eta_0 \le \min\left\{ \frac{1}{\kappa^2}, \frac{1}{\lambda_1} \right\}$. Then, for any integer $k^* \ge 1$, we have*

$$\mathbb{E}\left[ \left\langle \overline{f_n^{b(1)}}, \boldsymbol{\Sigma} \overline{f_n^{b(1)}} \right\rangle_{\mathcal{H}} \right] \le \frac{1}{1 - \eta_0 \kappa^2} \frac{\eta_0 \kappa^2}{n} \frac{1}{\eta_0^s} \left\| \left( \sum_{i=0}^{n-1} (\boldsymbol{I} - \eta_0 \boldsymbol{\Sigma})^{2i} (\eta_0 \boldsymbol{\Sigma})^s \right)^{\frac{1}{2}} \right\|^2 \left\| \boldsymbol{\Sigma}^{\frac{1-s}{2}} (f_0 - f_\rho^*) \right\|_{\mathcal{H}}^2$$
$$\le \frac{2\kappa^2}{1 - \eta_0 \kappa^2} \frac{1}{n^{\min\{s,1\}} \eta_0^{s-1}} \left\| \boldsymbol{\Sigma}^{\frac{1-s}{2}} (f_0 - f_\rho^*) \right\|_{\mathcal{H}}^2. \tag{18}$$

Meanwhile, Dieuleveut & Bach (2016) provides the following expression $\mathbb{E}\left[ \left\langle \overline{\tilde{f}_n^v}, \boldsymbol{\Sigma} \overline{\tilde{f}_n^v} \right\rangle_{\mathcal{H}} \right]$, which is

$$\mathbb{E}\left[ \left\langle \overline{\tilde{f}_n^v}, \boldsymbol{\Sigma} \overline{\tilde{f}_n^v} \right\rangle_{\mathcal{H}} \right] = \frac{1}{n^2} \mathbb{E}\left[ \left\| \sum_{j=1}^{n-1} \eta_0^2 \sum_{i=j}^{n-1} (\boldsymbol{I} - \eta_0 \boldsymbol{\Sigma})^{i-j} \boldsymbol{\Sigma}^{\frac{1}{2}} \Xi_j \right\|_{\mathcal{H}}^2 \right]. \tag{19}$$

**Lemma C.3** (Population Variance Upper Bound). *Consider SGD with averaged iterates defined in* (17). *Suppose that $\eta_0 \le \frac{1}{\lambda_1}$. Then, for any integer $k^* \ge 1$, we have*

$$\mathbb{E}\left[ \left\langle \overline{\tilde{f}_n^v}, \boldsymbol{\Sigma} \overline{\tilde{f}_n^v} \right\rangle_{\mathcal{H}} \right] \le \sigma^2 \left( \frac{k^*}{n} + \frac{1}{3} \sum_{i=k^*+1}^{+\infty} n \eta_0^2 \lambda_i^2 \right).$$

*Proof.* According to the form in (19), we have

$$
\mathbb{E}\left[\left\langle \overline{\widetilde{f}_n^v}, \mathbf{\Sigma}\overline{\widetilde{f}_n^v}\right\rangle_{\mathcal{H}}\right] = \frac{1}{n^2}\mathbb{E}\left[\left\|\sum_{j=1}^{n-1}\eta_0^2\sum_{i=j}^{n-1}(\mathbf{I}-\eta_0\mathbf{\Sigma})^{i-j}\mathbf{\Sigma}^{\frac{1}{2}}\Xi_j\right\|_{\mathcal{H}}^2\right]
$$

$$
= \frac{1}{n^2}\mathbb{E}\left[\text{tr}\left(\sum_{j=1}^{n-1}\eta_0^2\left(\sum_{i=j}^{n-1}(\mathbf{I}-\eta_0\mathbf{\Sigma})^{i-j}\right)^2\mathbf{\Sigma}\Xi_j\otimes\Xi_j\right)\right]
$$

$$
= \frac{1}{n^2}\text{tr}\left(\sum_{j=1}^{n-1}\eta_0^2\left(\sum_{i=j}^{n-1}(\mathbf{I}-\eta_0\mathbf{\Sigma})^{i-j}\right)^2\mathbf{\Sigma}\mathbb{E}\left[\Xi_i\otimes\Xi_i\right]\right),
$$

where in the second equality, we use the commutativity of each $\mathbf{I}-\eta_j\mathbf{\Sigma}$, and from the fact that for any $1\le i\ne j\le n$, $\Xi_i$ and $\Xi_j$ are independent, and under Assumption 4.1 $\mathbb{E}\left[\Xi_i\right]=0$.

Using the property that for any $1\le i\le n$, $\mathbb{E}\left[\Xi_i\otimes\Xi_i\right]\preceq\sigma^2\mathbf{\Sigma}$, we have

$$
\mathbb{E}\left[\left\langle \overline{\widetilde{f}_n^v}, \mathbf{\Sigma}\overline{\widetilde{f}_n^v}\right\rangle_{\mathcal{H}}\right] \le \frac{\sigma^2}{n^2}\text{tr}\left(\sum_{j=1}^{n-1}\eta_0^2\left(\sum_{i=j}^{n-1}(\mathbf{I}-\eta_0\mathbf{\Sigma})^{i-j}\right)^2\mathbf{\Sigma}^2\right).
$$

Similar to the proof of Lemma B.4, we split the summation on the RHS into two parts: $k\le k^*$ and $k>k^*$. The resulting upper bound is summarized in the following:

$$
\mathbb{E}\left[\left\langle \overline{\widetilde{f}_n^v}, \mathbf{\Sigma}\overline{\widetilde{f}_n^v}\right\rangle_{\mathcal{H}}\right] \le \frac{\sigma^2}{n^2}\sum_{k=1}^{+\infty}\left(\sum_{j=1}^{n-1}\eta_0^2\left(\sum_{i=j}^{n-1}(1-\eta_0\lambda_k)^{i-j}\right)^2\lambda_k^2\right)
$$

$$
= \sigma^2\left(\sum_{k=1}^{k^*}\frac{1}{n^2}\sum_{j=1}^{n-1}\left(1-(1-\eta_0\lambda_k)^j\right)^2 + \sum_{k=k^*+1}^{+\infty}\frac{1}{n^2}\sum_{j=1}^{n-1}\left(1-(1-\eta_0\lambda_k)^j\right)^2\right)
$$

$$
\le \sigma^2\left(\sum_{k=1}^{k^*}\frac{1}{n^2}\sum_{j=1}^{n-1}1 + \sum_{k=k^*+1}^{+\infty}\frac{1}{n^2}\sum_{j=1}^{n-1}(j\eta_0\lambda_k)^2\right)
$$

$$
\le \sigma^2\left(\frac{k^*}{n} + \frac{1}{3}\sum_{k=k^*+1}^{+\infty}n\eta_0^2\lambda_k^2\right),
$$

where in the second inequality, we use $1-nx\le(1-x)^n\le1$ for any $0<x<1$ and $n\in\mathbb{N}$. $\qquad\square$

Besides, we have the following lemma:

**Lemma C.4** (Residual Variance Upper Bound)**.** *Consider SGD with averaged iterates defined in* (17)*. Suppose that* $\eta_0\le\min\left\{\frac{2}{\kappa^2},\frac{1}{\lambda_1}\right\}$*. Then, for any integer* $k^*\ge1$*, we have*

$$
\mathbb{E}\left[\left\langle \overline{f_n^{v(1)}}, \mathbf{\Sigma}\overline{f_n^{v(1)}}\right\rangle_{\mathcal{H}}\right] \le \frac{\eta_0\sigma^2}{2-\eta_0\kappa^2}\kappa^2\left(\frac{k^*}{n} + \frac{1}{3}\sum_{k=k^*+1}^{+\infty}n\eta_0^2\lambda_k^2\right).
$$

We omit the proof since it follows a similar argument to that of Lemma B.10,

Combining Lemma C.1, C.2, C.3, and C.4, we have the following corollary.

**Corollary C.5** (Upper Bound for SGD with Averaged Iterates). *Consider SGD with averaged iterates defined in* (17). *Suppose that* $\eta_0 \leq \min\left\{\frac{1}{\kappa^2}, \frac{1}{\lambda_1}\right\}$. *Then, for any integer* $k^* \geq 1$, *we have*

$$\mathbb{E}\left[\|f_n - f_\rho^*\|_{L^2}^2\right] \leq 4\frac{1}{n^{\min\{s,2\}}\eta_0^s}\left\|\boldsymbol{\Sigma}^{\frac{1-s}{2}}(f_0 - f_\rho^*)\right\|_{\mathcal{H}}^2 + 4\sigma^2\left(\frac{k^*}{n} + \frac{1}{3}\sum_{k=k^*+1}^{+\infty} n\eta_0^2\lambda_k^2\right).$$

$$+ 4\frac{2\kappa^2}{1-\eta_0\kappa^2}\frac{1}{n^{\min\{s,1\}}\eta_0^{s-1}}\left\|\boldsymbol{\Sigma}^{\frac{1-s}{2}}(f_0 - f_\rho^*)\right\|_{\mathcal{H}}^2 + 4\frac{\eta_0\sigma^2}{2-\eta_0\kappa^2}\kappa^2\left(\frac{k^*}{n} + \frac{1}{3}\sum_{k=k^*+1}^{+\infty} n\eta_0^2\lambda_k^2\right).$$

# D. Proof of Optimal Rate

In this section, we discuss the optimality of SGD for kernel regression with the dot-product kernel $K$ under high-dimensional and asymptotic settings, respectively.

## D.1. High-Dimensional Setting

Under this setting, we consider sufficient large $n$ which satisfies $c_1 d^\gamma < n < c_2 d^\gamma$ for some fixed $\gamma > 0$ and absolute constants $c_1, c_2$.

To demonstrate the optimal rate, here we borrowed a minimax lower bound of $\|f - f_\rho^*\|_{L^2}^2$ from Lu et al. (2024a).

**Theorem D.1** (Minimax Lower Bound, from Lu et al. (2024a)). *In high-dimensional settings, where* $n$ *is bounded by* $c_1 d^\gamma < n < c_2 d^\gamma$ *for some fixed* $\gamma > 0$ *and constants* $c_1, c_2$, *consider* $\mathcal{X} = \mathbb{S}^d$. *The marginal distribution* $\rho_{\mathcal{X}}$ *is assumed to be the uniform distribution on* $\mathbb{S}^d$. *Let* $K$ *denote a dot-product kernel on the sphere, which satisfies Assumption 4.5. Let* $\mathcal{P}$ *consist of all distributions* $\rho$ *on* $\mathcal{X} \times \mathcal{Y}$ *given by* $y = f_\rho^*(\mathbf{x}) + \epsilon$, *where* $f_\rho^* \in [\mathcal{H}]^s$ *and* $\epsilon \sim \mathcal{N}(0, 1)$ *is independent of* $\mathbf{x}$. *Let* $p = \left\lceil\frac{\gamma}{s+1}\right\rceil - 1$, *then we have:*

$$\inf_{\hat{f}} \sup_{\rho \in \mathcal{P}} \mathbb{E}_\rho\left\|\hat{f} - f_\rho^*\right\|_{L^2}^2 = \Omega\left(d^{-\min\{\gamma-p, s(p+1)\}}\right).$$

The well-specified case corresponds to $s \geq 1$, and the mis-specified case corresponds to $0 < s < 1$.

In this section, we use different schedules to reach the minimax lower bound in the well-specified and mis-specified cases.

### D.1.1. WELL-SPECIFIED CASE

In the well-specified case, we use SGD with exponentially step decay schedule to reach the minimax lower bound.

**Theorem D.2** (Theorem 5.1). *In high-dimensional settings, where* $n$ *is bounded by* $c_1 d^\gamma < n < c_2 d^\gamma$ *for some fixed* $\gamma > 0$ *and constants* $c_1, c_2$, *consider* $\mathcal{X} = \mathbb{S}^d$. *The marginal distribution* $\rho_{\mathcal{X}}$ *is assumed to be the uniform distribution on* $\mathbb{S}^d$. *Let* $K$ *denote the dot-product kernel on* $\mathbb{S}^d$, *which satisfies Assumption 4.5. Given* $s \geq 1$, *supposing that Assumption 4.3 holds with* $s$, *and treating* $\gamma, \sigma, \kappa, c_1, c_2$ *and* $s$ *as constants, the excess risk of the output of SGD with exponentially decaying step size* $f_n^{dec}$ *satisfies:*

*(i) When* $\gamma \in (ps+p, ps+p+s]$ *for some* $p \in \mathbb{N}$, *with initial step size* $\eta_0 = \Theta(d^{-\gamma+p}\log_2 n \ln d) \leq \min\left\{\frac{1}{\kappa^2}, \frac{1}{\lambda_1}\right\}$, *and* $k^* = \Theta(d^p)$, *there exists a constant* $d_0$ *such that for any* $d \geq d_0$, *we have:*

$$\mathbb{E}\left[\|f_n^{dec} - f_\rho^*\|_{L^2}^2\right] \lesssim d^{-\gamma+p}\log_2^2 d.$$

*(ii) When* $\gamma \in (ps+p+s, (p+1)s+p+1]$ *for some* $p \in \mathbb{N}$, *with initial step size* $\eta_0 = \Theta(d^{-\gamma+p}\log_2 n \ln d) \leq \min\left\{\frac{1}{\kappa^2}, \frac{1}{\lambda_1}\right\}$, *and* $k^* = \Theta(d^p)$, *there exists a constant* $d_0$ *such that for any* $d \geq d_0$, *we have:*

$$\mathbb{E}\left[\|f_n^{dec} - f_\rho^*\|_{L^2}^2\right] \lesssim d^{-(p+1)s}.$$

*Proof.* According to (12) in Lemma B.3, we have

$$
\left\langle \tilde{f}_n^b, \boldsymbol{\Sigma} \tilde{f}_n^b \right\rangle_{\mathcal{H}} \leq \eta_0^{-s} \left\| (\boldsymbol{I} - \eta_0 \boldsymbol{\Sigma})^m (\eta_0 \boldsymbol{\Sigma})^{\frac{s}{2}} \right\|^2 \left\| \boldsymbol{\Sigma}^{\frac{1-s}{2}} (f_0 - f_\rho^*) \right\|_{\mathcal{H}}^2
$$
$$
\leq \eta_0^{-s} \left( \max_{k \geq 1} (1 - \eta_0 \lambda_k)^{2m} (\eta_0 \lambda_k)^s \right) \left\| \boldsymbol{\Sigma}^{\frac{1-s}{2}} (f_0 - f_\rho^*) \right\|_{\mathcal{H}}^2 .
$$

Select $k^*$ such that $\lambda_{k^*} = \mu_p$ and $\lambda_{k^*+1} = \mu_{p+1}$, so that $k^* = \Theta(d^p), p \in \mathbb{N}$, $\lambda_{k^*} = \Theta(d^{-p})$, and $\lambda_{k^*+1} = \Theta(d^{-p-1})$ by Lemma A.3.

We take $\eta_0 = \Theta(d^{-\gamma+p} \log_2 n \ln d)$. Then for $k \leq k^*$, we have $\log_2^2 d \cdot d^{-\gamma} \lesssim \eta_0 \lambda_k \lesssim \eta_0$. Recall $n = \Theta(d^\gamma)$, let $\eta_0 \lambda_k \geq C \log_2^2 n \cdot n^{-1}$ for some constant $C > 0$, we obtain:

$$
\eta_0^{-s} (1 - \eta_0 \lambda_k)^{2m} (\eta_0 \lambda_k)^s = (1 - \eta_0 \lambda_k)^{2m} \lambda_k^s
$$
$$
\leq \exp(-2m \eta_0 \lambda_k)
$$
$$
\lesssim \exp\left( -2 \frac{n}{\log_2 n} C \log_2^2 n \cdot n^{-1} \right)
$$
$$
= \exp(-2C \log_2 n).
$$

When $2C > \frac{(p+1)s}{\gamma}$, we have $\eta_0^{-s} (1 - \eta_0 \lambda_k)^{2m} (\eta_0 \lambda_k)^s \lesssim d^{-(p+1)s}$. Therefore,

$$
\eta_0^{-s} \left( \max_{k \leq k^*} (1 - \eta_0 \lambda_k)^{2m} (\eta_0 \lambda_k)^s \right) \lesssim d^{-(p+1)s}. \tag{20}
$$

For $k \geq k^* + 1$, we have $\lambda_k \lesssim d^{-(p+1)}$. This implies that

$$
\eta_0^{-s} \left( \max_{k \geq k^*+1} (1 - \eta_0 \lambda_k)^{2m} (\eta_0 \lambda_k)^s \right) \lesssim \max_{k \geq k^*+1} \lambda_k^s \lesssim d^{-(p+1)s}. \tag{21}
$$

Combining (20) and (21), we have

$$
\left\langle \tilde{f}_n^b, \boldsymbol{\Sigma} \tilde{f}_n^b \right\rangle_{\mathcal{H}} \lesssim d^{-(p+1)s} \left\| \boldsymbol{\Sigma}^{\frac{1-s}{2}} (f_0 - f_\rho^*) \right\|_{\mathcal{H}}^2 . \tag{22}
$$

Combining (22) with Lemma B.4, B.9 and B.10, we derive the total excess risk bound:

$$
\mathbb{E} \left[ \left\| f_n^{dec} - f_\rho^* \right\|_{L^2}^2 \right] \lesssim d^{-(p+1)s} \left\| \boldsymbol{\Sigma}^{\frac{1-s}{2}} (f_0 - f_\rho^*) \right\|_{\mathcal{H}}^2
$$
$$
+ \left( \sigma^2 + \frac{\eta_0 \sigma^2 \kappa^2}{2 - \eta_0 \kappa^2} + \|f_0 - f_\rho^*\|_{\mathcal{H}}^2 \kappa^2 \right) \left( \frac{k^* \log_2^2 n}{n} + \sum_{i=k^*+1}^{+\infty} n \eta_0^2 \lambda_i^2 \right).
$$

Since $\eta_0 \leq \min \left\{ \frac{1}{\kappa^2}, \frac{1}{\lambda_1} \right\}$, we have $\frac{\eta_0 \sigma^2 \kappa^2}{2 - \eta_0 \kappa^2} \leq \eta_0 \sigma^2 \kappa^2 \leq \sigma^2 = 1$. Additionally, by the source condition with $s \geq 1$, we have $\|f_0 - f_\rho^*\|_{\mathcal{H}}^2 \leq \left\| \boldsymbol{\Sigma}^{\frac{1-s}{2}} (f_0 - f_\rho^*) \right\|_{\mathcal{H}}^2 \lesssim 1$. Hence, the bound simplifies to:

$$
\mathbb{E} \left[ \left\| f_n^{dec} - f_\rho^* \right\|_{L^2}^2 \right] \lesssim d^{-(p+1)s} + \frac{k^* \log_2^2 n}{n} + \sum_{k=k^*+1}^{+\infty} n \eta_0^2 \lambda_k^2 .
$$

Since $\sum_{k=k^*+1}^{+\infty} \lambda_k^2 \leq \lambda_{k^*+1} \sum_{k=k^*+1}^{+\infty} \lambda_k \leq \kappa^2 \lambda_{k^*+1}$, we further obtain:

$$
\mathbb{E} \left[ \left\| f_n^{dec} - f_\rho^* \right\|_{L^2}^2 \right] \lesssim d^{-(p+1)s} + \frac{k^* \log_2^2 n}{n} + n \eta_0^2 \lambda_{k^*+1} .
$$

We then set $p = \left\lceil \frac{\gamma}{s+1} \right\rceil - 1$, which ensures $p + ps < \gamma \le (p+1) + (p+1)s$. Given $n = \Theta(d^\gamma)$ and $\eta_0 = \Theta(d^{-\gamma+p} \log_2 n \ln d)$,

$$
\begin{aligned}
\mathbb{E}\left[ \left\| f_n^{dec} - f_\rho^* \right\|_{L^2}^2 \right] &\lesssim d^{-(p+1)s} + \frac{k^* \log_2^2 n}{n} + n\eta_0^2 \lambda_{k^*+1} \\
&\lesssim d^{-(p+1)s} + d^{p-\gamma} \log_2^2 d + d^{p-\gamma-1} \log_2^2 d \ln^2 d \\
&\lesssim d^{-(p+1)s} + d^{p-\gamma} \log_2^2 d \\
&= \begin{cases} \mathcal{O}(d^{-(p+1)s}) & \gamma \in (p + ps + s, (p+1) + (p+1)s] \\ \mathcal{O}(d^{p-\gamma} \log_2^2 d) & \gamma \in (p + ps, p + ps + s] \end{cases}.
\end{aligned}
$$

$\square$

### D.1.2. MIS-SPECIFIED CASE

In the mis-specified case, we use SGD with averaged iterates to reach the minimax lower bound.

**Theorem D.3** (Theorem 5.2). *In high-dimensional settings, where $n$ is bounded by $c_1 d^\gamma < n < c_2 d^\gamma$ for some fixed $\gamma > 0$ and constants $c_1, c_2$, consider $\mathcal{X} = \mathbb{S}^d$. The marginal distribution $\rho_\mathcal{X}$ is assumed to be the uniform distribution on $\mathbb{S}^d$. Let $K$ denote the dot-product kernel on $\mathbb{S}^d$, which satisfies Assumption 4.5. Given $0 < s < 1$, supposing that Assumption 4.3 holds with $s$, and treating $\gamma, \sigma, \kappa, c_1, c_2$ and $s$ as constants, the excess risk of the output of SGD with averaged iterates $f_n^{avg}$ satisfies:*

*(i) When $\gamma \in (ps + p, ps + p + s]$ for some $p \in \mathbb{N}$, with initial step size $\eta_0 = \Theta\left(d^{-\gamma+p+\frac{s}{2}}\right) \le \min\left\{\frac{1}{\kappa^2}, \frac{1}{\lambda_1}\right\}$ for $p \ge 1$ and $\eta_0 = \Theta\left(d^{-\frac{\gamma}{2}}\right) \le \min\left\{\frac{1}{\kappa^2}, \frac{1}{\lambda_1}\right\}$ for $p = 0$, there exists a constant $d_0$ such that for any $d \ge d_0$, we have:*

$$
\mathbb{E}\left[ \left\| f_n^{avg} - f_\rho^* \right\|_{L^2}^2 \right] \lesssim d^{-\gamma+p}.
$$

*(ii) When $\gamma \in (ps + p + s, (p+1)s + p + 1]$ for some $p \in \mathbb{N}$, with initial step size $\eta_0 = \Theta\left(d^{-\gamma+p+\frac{s}{2}}\right) \le \min\left\{\frac{1}{\kappa^2}, \frac{1}{\lambda_1}\right\}$ for $p \ge 1$ and $\eta_0 = \Theta\left(d^{-\frac{\gamma}{2}}\right) \le \min\left\{\frac{1}{\kappa^2}, \frac{1}{\lambda_1}\right\}$ for $p = 0$, there exists a constant $d_0$ such that for any $d \ge d_0$, we have:*

$$
\mathbb{E}\left[ \left\| f_n^{avg} - f_\rho^* \right\|_{L^2}^2 \right] \lesssim d^{-(p+1)s}.
$$

*Proof.* According to (17) in Lemma C.1, we have

$$
\begin{aligned}
\left\langle \overline{\tilde{f}_n^b}, \Sigma \overline{\tilde{f}_n^b} \right\rangle_\mathcal{H} &\le n^{-2}\eta_0^{-s} \left\| \sum_{i=0}^{n-1} (\boldsymbol{I} - \eta_0 \boldsymbol{\Sigma})^i (\eta_0 \boldsymbol{\Sigma})^{\frac{s}{2}} \right\|^2 \left\| \boldsymbol{\Sigma}^{\frac{1-s}{2}} (f_0 - f_\rho^*) \right\|_\mathcal{H}^2 \\
&= n^{-2}\eta_0^{-s} \left( \max_{k\ge 1} \left( \sum_{i=0}^{n-1} (1 - \eta_0 \lambda_k)^i \right)^2 (\eta_0 \lambda_k)^s \right) \left\| \boldsymbol{\Sigma}^{\frac{1-s}{2}} (f_0 - f_\rho^*) \right\|_\mathcal{H}^2.
\end{aligned}
$$

Select $k^*$ such that $\lambda_{k^*} = \mu_p$ and $\lambda_{k^*+1} = \mu_{p+1}$, so that $k^* = \Theta(d^p), p \in \mathbb{N}$, $\lambda_{k^*} = \Theta(d^{-p})$, and $\lambda_{k^*+1} = \Theta(d^{-p-1})$ by Lemma A.3.

For $k \le k^*$, recall $n = \Theta(d^\gamma)$.

In the case $p \ge 1$, we take $\eta_0 = \Theta\left(d^{-\gamma+p+\frac{s}{2}}\right) \lesssim \min\left\{\frac{1}{\kappa^2}, \frac{1}{\lambda_1}\right\}$, then $\eta_0 \lambda_k^{1-\frac{s}{2}} \ge C \cdot d^{\frac{(p+1)s}{2}} n^{-1}$ for some constant $C > 0$, so we obtain:

$$
\begin{aligned}
n^{-2}\eta_0^{-s} \left( \left( \sum_{i=0}^{n-1} (1 - \eta_0 \lambda_k)^i \right)^2 (\eta_0 \lambda_k)^s \right) &= n^{-2} \left( 1 - (1 - \eta_0 \lambda_k)^n \right)^2 \eta_0^{-2} \lambda_k^{s-2} \\
&\le n^{-2} \eta_0^{-2} \lambda_k^{s-2} \\
&\le \frac{1}{C^2} d^{-(p+1)s}.
\end{aligned}
$$

In the case $p = 0$, we take $\eta_0 = \Theta\left(d^{-\frac{\gamma}{2}}\right) \lesssim \min\left\{\frac{1}{\kappa^2}, \frac{1}{\lambda_1}\right\}$, so we obtain:

$$n^{-2}\eta_0^{-s}\left(\left(\sum_{i=0}^{n-1}(1 - \eta_0\lambda_k)^i\right)^2 (\eta_0\lambda_k)^s\right) \leq n^{-2}\eta_0^{-2}\lambda_k^{s-2}$$

$$\lesssim d^{-\gamma}$$

$$= d^{p-\gamma}.$$

Therefore, we have

$$n^{-2}\eta_0^{-s}\left(\max_{k \leq k^*}\left(\sum_{i=0}^{n-1}(1 - \eta_0\lambda_k)^i\right)^2 (\eta_0\lambda_k)^s\right) \lesssim d^{-(p+1)s} + d^{p-\gamma}. \tag{23}$$

For $k \geq k^* + 1$, we have

$$n^{-2}\eta_0^{-s}\left(\left(\sum_{i=0}^{n-1}(1 - \eta_0\lambda_k)^i\right)^2 (\eta_0\lambda_k)^s\right) = n^{-2}\left(1 - (1 - \eta_0\lambda_k)^n\right)^2 \eta_0^{-2}\lambda_k^{s-2}$$

$$\overset{(i)}{\leq} n^{-2}(n\eta_0\lambda_k)^2\eta_0^{-2}\lambda_k^{s-2}$$

$$= \lambda_k^s,$$

where inequality (i) uses the fact that if $0 \leq nx \leq 1$, we have $1 - (1 - x)^n \leq nx$. This condition is satisfied since $n\eta_0\lambda_k \lesssim d^{\frac{s}{2}-1} \leq 1$ ($s \leq 1$). Therefore, we have

$$n^{-2}\eta_0^{-s}\left(\max_{k \geq k^*+1}\left(\sum_{i=0}^{n-1}(1 - \eta_0\lambda_k)^i\right)^2 (\eta_0\lambda_k)^s\right) \lesssim \max_{k \geq k^*+1}\lambda_k^s \lesssim d^{-(p+1)s}. \tag{24}$$

Combining (23) and (24), we have

$$\left\langle\overline{\widetilde{f}_n^b}, \mathbf{\Sigma}\overline{\widetilde{f}_n^b}\right\rangle_{\mathcal{H}} \lesssim (d^{-(p+1)s} + d^{p-\gamma})\left\|\mathbf{\Sigma}^{\frac{1-s}{2}}(f_0 - f_\rho^*)\right\|_{\mathcal{H}}^2. \tag{25}$$

According to in (18) Lemma C.2, we have

$$\mathbb{E}\left[\left\langle\overline{f_n^{b(1)}}, \mathbf{\Sigma}\overline{f_n^{b(1)}}\right\rangle_{\mathcal{H}}\right] \leq \frac{1}{1 - \eta_0\kappa^2}\frac{\eta_0\kappa^2}{n}\frac{1}{\eta_0^s}\left\|\left(\sum_{i=0}^{n-1}(\mathbf{I} - \eta_0\mathbf{\Sigma})^{2i}(\eta_0\mathbf{\Sigma})^s\right)^{\frac{1}{2}}\right\|^2\left\|\mathbf{\Sigma}^{\frac{1-s}{2}}(f_0 - f_\rho^*)\right\|_{\mathcal{H}}^2$$

$$= \frac{1}{1 - \eta_0\kappa^2}\frac{\eta_0\kappa^2}{n}\frac{1}{\eta_0^s}\left(\max_{k \geq 1}\sum_{i=0}^{n-1}(1 - \eta_0\lambda_k)^{2i}(\eta_0\lambda_k)^s\right)\left\|\mathbf{\Sigma}^{\frac{1-s}{2}}(f_0 - f_\rho^*)\right\|_{\mathcal{H}}^2$$

$$\lesssim n^{-1}\left(\max_{k \geq 1}\left(1 - (1 - \eta_0\lambda_k)^{2n}\right)\lambda_k^{s-1}\right)\left\|\mathbf{\Sigma}^{\frac{1-s}{2}}(f_0 - f_\rho^*)\right\|_{\mathcal{H}}^2.$$

For $k \leq k^*$, we have $n^{-1}\left(1 - (1 - \eta_0\lambda_k)^{2n}\right)\lambda_k^{s-1} \leq \lambda_k^{s-1}n^{-1}$ and $s \leq 1$. This implies that

$$n^{-1}\left(\max_{k \leq k^*}\left(1 - (1 - \eta_0\lambda_k)^{2n}\right)\lambda_k^{s-1}\right) \leq \max_{k \leq k^*}\lambda_k^{s-1}n^{-1} \lesssim d^{-ps+p-\gamma} \lesssim d^{p-\gamma}. \tag{26}$$

For $k \geq k^* + 1$, we have

$$n^{-1}\left(\left(1 - (1 - \eta_0\lambda_k)^{2n}\right)\lambda_k^{s-1}\right) \overset{(i)}{\leq} n^{-1}(2n\eta_0\lambda_k)\lambda_k^{s-1}$$

$$\leq \lambda_k^s,$$

where inequality (i) uses the fact that if $0 \leq nx \leq 1$, we have $1 - (1-x)^n \leq nx$. This condition is satisfied since $n\eta_0\lambda_k \lesssim d^{\frac{s}{2}-1} \leq 1$ ($s \leq 1$). Therefore, we have

$$n^{-1}\left(\max_{k \geq k^*+1}\left(1 - (1-\eta_0\lambda_k)^{2n}\right)\lambda_k^{s-1}\right) \lesssim \max_{k \geq k^*+1}\lambda_k^s \lesssim d^{-(p+1)s}. \tag{27}$$

Combining (26) and (27), we have

$$\mathbb{E}\left[\left\langle \overline{f_n^{b(1)}}, \boldsymbol{\Sigma}\overline{f_n^{b(1)}}\right\rangle_{\mathcal{H}}\right] \lesssim \left(d^{-(p+1)s} + d^{p-\gamma}\right)\left\|\boldsymbol{\Sigma}^{\frac{1-s}{2}}(f_0 - f_\rho^*)\right\|_{\mathcal{H}}^2. \tag{28}$$

Combining (25) and (28) with Lemma C.2, C.3, and C.4, we derive the total excess risk bound:

$$\mathbb{E}\left[\left\|f_n^{avg} - f_\rho^*\right\|_{L^2}^2\right] \lesssim \left(d^{-(p+1)s} + d^{p-\gamma}\right)\left\|\boldsymbol{\Sigma}^{\frac{1-s}{2}}(f_0 - f_\rho^*)\right\|_{\mathcal{H}}^2$$
$$+ \left(\sigma^2 + \frac{\eta_0\sigma^2\kappa^2}{2-\eta_0\kappa^2}\right)\left(\frac{k^*}{n} + \sum_{k=k^*+1}^{+\infty} n\eta_0^2\lambda_k^2\right).$$

Since $\eta_0 \leq \min\left\{\frac{1}{\kappa^2}, \frac{1}{\lambda_1}\right\}$, we have $\frac{\eta_0\sigma^2\kappa^2}{2-\eta_0\kappa^2} \leq \eta_0\sigma^2\kappa^2 \leq \sigma^2 = 1$. Additionally, by the source condition with $s \geq 1$, we have $\|f_0 - f_\rho^*\|_{\mathcal{H}}^2 \leq \left\|\boldsymbol{\Sigma}^{\frac{1-s}{2}}(f_0 - f_\rho^*)\right\|_{\mathcal{H}}^2 \lesssim 1$. Hence, the bound simplifies to:

$$\mathbb{E}\left[\left\|f_n^{avg} - f_\rho^*\right\|_{L^2}^2\right] \lesssim d^{-(p+1)s} + d^{p-\gamma} + \frac{k^*}{n} + \sum_{k=k^*+1}^{+\infty} n\eta_0^2\lambda_k^2.$$

Since $\sum_{k=k^*+1}^{+\infty}\lambda_k^2 \leq \lambda_{k^*+1}\sum_{k=k^*+1}^{+\infty}\lambda_k \leq \kappa^2\lambda_{k^*+1}$, we further obtain:

$$\mathbb{E}\left[\left\|f_n^{avg} - f_\rho^*\right\|_{L^2}^2\right] \lesssim d^{-(p+1)s} + d^{p-\gamma} + \frac{k^*}{n} + n\eta_0^2\lambda_{k^*+1}.$$

We then set $p = \left\lceil\frac{\gamma}{s+1}\right\rceil - 1$, which ensures $p + ps < \gamma \leq (p+1) + (p+1)s$. Given $n = \Theta(d^\gamma)$ and $\eta_0 = \begin{cases} \Theta(d^{-\gamma+p+\frac{s}{2}}) & , p \geq 1 \\ \Theta(d^{-\frac{\gamma}{2}}) & , p = 0 \end{cases}$,

$$\mathbb{E}\left[\left\|f_n^{avg} - f_\rho^*\right\|_{L^2}^2\right] \lesssim d^{-(p+1)s} + d^{p-\gamma} + \frac{k^*}{n} + n\eta_0^2\lambda_{k^*+1}$$
$$\lesssim d^{-(p+1)s} + d^{p-\gamma} + d^{p-\gamma-1}$$
$$\lesssim d^{-(p+1)s} + d^{p-\gamma}$$
$$= \begin{cases} \mathcal{O}(d^{-(p+1)s}) & \gamma \in (p+ps+s, (p+1)+(p+1)s] \\ \mathcal{O}(d^{p-\gamma}) & \gamma \in (p+ps, p+ps+s] \end{cases}.$$

$\square$

## D.2. Asymptotic Setting

In the asymptotic setting, $n \gg d$, the decay rate of the kernel is $\lambda_k = \Theta(k^{-1-\frac{1}{d}})$. We obtain the rate $\mathcal{O}\left(n^{-\frac{s(d+1)}{s(d+1)+d}}\right)$ when $\frac{1}{d+1} < s \leq 2$. This implies SGD with exponentially decaying step size schedule achieves optimally for $s \geq 1$, and SGD with averaged iterates achieves optimality for $\frac{1}{d+1} < s \leq 2$.

Similarly, we also borrowed a minimax lower bound for asymptotic setting from Caponnetto & De Vito (2007).

**Theorem D.4** (Minimax Lower Bound, from Caponnetto & De Vito (2007)). *In asymptotic settings, where $n \gg d$, consider $\mathcal{X} = \mathbb{S}^d$. The marginal distribution $\rho_{\mathcal{X}}$ is assumed to be the uniform distribution on $\mathbb{S}^d$. Let $K_{\mathrm{NTK}}$ the NTK of a ReLU network with $L$ layers with inputs on $\mathbb{S}^d$. Let $\mathcal{P}$ consist of all distributions $\rho$ on $\mathcal{X} \times \mathcal{Y}$ given by $y = f_\rho^*(\mathbf{x}) + \epsilon$, where $f_\rho^* \in [\mathcal{H}]^s$ and $\epsilon \sim \mathcal{N}(0,1)$ is independent of $\mathbf{x}$, then we have:*

$$\inf_{\hat{f}} \sup_{\rho \in \mathcal{P}} \mathbb{E}_\rho \left\| \hat{f} - f_\rho^* \right\|_{L^2}^2 = \Omega\left( n^{-\frac{s(d+1)}{s(d+1)+d}} \right).$$

In the well-specified case, we use SGD with exponentially step decay schedule to reach the minimax lower bound.

**Theorem D.5** (Theorem 5.3). *In asymptotic settings, where $n \gg d$, consider $\mathcal{X} = \mathbb{S}^d$. The marginal distribution $\rho_{\mathcal{X}}$ is assumed to be the uniform distribution on $\mathbb{S}^d$. Let $K_{\mathrm{NTK}}$ denote the NTK of a ReLU network with $L$ layers with inputs on $\mathbb{S}^d$. For $s \geq 1$, supposing that Assumption 4.3 holds for $s$, with initial step size $\eta_0 = \Theta\left( n^{\frac{1-s(d+1)}{s(d+1)+d}} \right) \leq \min\left\{ \frac{1}{\kappa^2}, \frac{1}{\lambda_1} \right\}$, the excess risk of the output of SGD with exponentially decaying step size $f_n^{dec}$ satisfies:*

$$\mathbb{E}\left[ \|f_n^{dec} - f_\rho^*\|_{L^2}^2 \right] \lesssim \log_2^s n \cdot n^{-\frac{s(d+1)}{s(d+1)+d}}.$$

In the mis-specified case, we use SGD with averaged iterates to reach the minimax lower bound.

*Proof.* Combining Lemma B.3, B.4, B.9 and B.10, we obtain the following bound:

$$\mathbb{E}\left[ \left\| f_n^{dec} - f_\rho^* \right\|_{L^2}^2 \right] \lesssim \left( \left( \frac{s \log_2 n}{n\eta_0} \right)^s \left\| \mathbf{\Sigma}^{\frac{1-s}{2}}(f_0 - f_\rho^*) \right\|_{\mathcal{H}}^2 \right)$$
$$+ \left( \left( \sigma^2 + \frac{\eta_0 \sigma^2 \kappa^2}{2 - \eta_0 \kappa^2} + \|f_0 - f_\rho^*\|_{\mathcal{H}}^2 \kappa^2 \right) \left( \frac{k^* \log_2^2 n}{n} + \sum_{i=k^*+1}^{+\infty} n\eta_0^2 \lambda_i^2 \right) \right).$$

Since $\eta_0 \leq \min\left\{ \frac{1}{\kappa^2}, \frac{1}{\lambda_1} \right\}$, we have $\frac{\eta_0 \sigma^2 \kappa^2}{2 - \eta_0 \kappa^2} \leq \eta_0 \sigma^2 \kappa^2 \leq \sigma^2 = 1$. Additionally, by the source condition with $s \geq 1$, we have $\|f_0 - f_\rho^*\|_{\mathcal{H}}^2 \leq \left\| \mathbf{\Sigma}^{\frac{1-s}{2}}(f_0 - f_\rho^*) \right\|_{\mathcal{H}}^2 \lesssim 1$. Hence, the bound simplifies to:

$$\mathbb{E}\left[ \left\| f_n^{dec} - f_\rho^* \right\|_{L^2}^2 \right] \lesssim \left( \frac{\log_2 n}{n\eta_0} \right)^s + \frac{k^* \log_2^2 n}{n} + \sum_{k=k^*+1}^{+\infty} n\eta_0^2 \lambda_k^2.$$

Since $\sum_{k=k^*+1}^{+\infty} \lambda_k^2 \lesssim \sum_{k=k^*+1}^{+\infty} k^{-2-\frac{2}{d}} \lesssim k^{*-1-\frac{2}{d}}$, we further obtain:

$$\mathbb{E}\left[ \left\| f_n^{avg} - f_\rho^* \right\|_{L^2}^2 \right] \lesssim \left( \frac{\log_2 n}{n\eta_0} \right)^s + \frac{k^* \log_2^2 n}{n} + n\eta_0^2 k^{*-1-\frac{2}{d}}.$$

Select $k^*$ and $\eta_0$ such that $k^* = \Theta\left( n^{\frac{d}{s(d+1)+d}} \right)$ and $\eta_0 = \Theta\left( n^{\frac{1-s(d+1)}{s(d+1)+d}} \right) \lesssim 1$, we have:

$$\mathbb{E}\left[ \left\| f_n^{avg} - f_\rho^* \right\|_{L^2}^2 \right] \lesssim \log_2^s n \cdot n^{-\frac{s(d+1)}{s(d+1)+d}}.$$

$\square$

**Theorem D.6** (Theorem 5.4). *In asymptotic settings, where $n \gg d$, consider $\mathcal{X} = \mathbb{S}^d$. The marginal distribution $\rho_{\mathcal{X}}$ is assumed to be the uniform distribution on $\mathbb{S}^d$. Let $K_{\mathrm{NTK}}$ denote the NTK of a ReLU network with $L$ layers with inputs on $\mathbb{S}^d$. For $\frac{1}{d+1} \leq s \leq 1$, supposing that Assumption 4.3 holds for $s$, with initial step size $\eta_0 = \Theta\left( n^{\frac{1-s(d+1)}{s(d+1)+d}} \right) \leq \min\left\{ \frac{1}{\kappa^2}, \frac{1}{\lambda_1} \right\}$, the excess risk of the output of SGD with averaged iterates $f_n^{avg}$ satisfies:*

$$\mathbb{E}\left[ \|f_n^{avg} - f_\rho^*\|_{L^2}^2 \right] \lesssim n^{-\frac{s(d+1)}{s(d+1)+d}}.$$

*Proof.* Combining Lemma C.1, C.3, C.2 and C.4, we obtain the following bound:

$$\mathbb{E}\left[\left\|f_n^{avg} - f_\rho^*\right\|_{L^2}^2\right] \lesssim \left(\left(\frac{1}{n\eta_0}\right)^s + \frac{2\kappa^2}{1 - \eta_0\kappa^2}\eta_0^{1-s}n^{-s}\right)\left\|\Sigma^{\frac{1-s}{2}}(f_0 - f_\rho^*)\right\|_{\mathcal{H}}^2$$
$$+ \left(\sigma^2 + \frac{\eta_0\sigma^2\kappa^2}{2 - \eta_0\kappa^2}\right)\left(\frac{k^*}{n} + \sum_{i=k^*+1}^{+\infty} n\eta_0^2\lambda_i^2\right).$$

Since $\eta_0 \leq \min\left\{\frac{1}{\kappa^2}, \frac{1}{\lambda_1}\right\}$, we have $\frac{\eta_0\sigma^2\kappa^2}{2-\eta_0\kappa^2} \leq \eta_0\sigma^2\kappa^2 \leq \sigma^2 = 1$ and $\frac{\kappa^2}{1-\eta_0\kappa^2}\eta_0^{1-s}n^{-s} \leq (n\eta_0)^{-s}$. Additionally, by the source condition with $s \geq 1$, we have $\|f_0 - f_\rho^*\|_{\mathcal{H}}^2 \leq \left\|\Sigma^{\frac{1-s}{2}}(f_0 - f_\rho^*)\right\|_{\mathcal{H}}^2 \lesssim 1$. Hence, the bound simplifies to:

$$\mathbb{E}\left[\left\|f_n^{dec} - f_\rho^*\right\|_{L^2}^2\right] \lesssim \left(\frac{1}{n\eta_0}\right)^s + \frac{k^*}{n} + \sum_{k=k^*+1}^{+\infty} n\eta_0^2\lambda_k^2.$$

Since $\sum_{k=k^*+1}^{+\infty}\lambda_k^2 \lesssim \sum_{k=k^*+1}^{+\infty} k^{-2-\frac{2}{d}} \lesssim k^{*-1-\frac{2}{d}}$, we further obtain:

$$\mathbb{E}\left[\left\|f_n^{avg} - f_\rho^*\right\|_{L^2}^2\right] \lesssim \left(\frac{1}{n\eta_0}\right)^s + \frac{k^*}{n} + n\eta_0^2 k^{*-1-\frac{2}{d}}.$$

Noticing that $s \geq \frac{1}{d+1}$, select $k^*$ and $\eta_0$ such that $k^* = \Theta\left(n^{\frac{d}{s(d+1)+d}}\right)$ and $\eta_0 = \Theta\left(n^{\frac{1-s(d+1)}{s(d+1)+d}}\right) \lesssim 1$, we have:

$$\mathbb{E}\left[\left\|f_n^{avg} - f_\rho^*\right\|_{L^2}^2\right] \lesssim n^{-\frac{s(d+1)}{s(d+1)+d}}.$$

$\square$

# E. Lower Bound of SGD with Averaged Iterates

In this section, we give a lower bound of SGD with averaged iterates, which helps to explain why it is not optimal when $s > 2$.

In this section, let $\rho$ be a distribution satisfying that

$$y = f_\rho^*(\mathbf{x}) + \epsilon, \text{ where } \epsilon \sim N(0, \sigma^2) \text{ is a independent variable for any } \mathbf{x},$$

which implies

$$\mathbb{E}_{(\mathbf{x},y)\sim\rho}[\Xi] = 0, \mathbb{E}_{(\mathbf{x},y)\sim\rho}[\Xi_{(\mathbf{x},y)} \otimes \Xi_{(\mathbf{x},y)}] = \sigma^2\Sigma.$$

To get a lower bound, we have the following lemmas:

**Lemma E.1.** *Consider SGD with averaged iterates defined in* (17)*. Then for any $0 \leq i \leq j \leq n$, we have*

$$\mathbb{E}\left[(f_i - f_\rho^*) \otimes (f_j - f_\rho^*)\right] = \mathbb{E}\left[(f_i - f_\rho^*) \otimes (\mathbf{I} - \eta_0\Sigma)^{j-i}(f_i - f_\rho^*)\right];$$
$$\mathbb{E}\left[(f_j - f_\rho^*) \otimes (f_i - f_\rho^*)\right] = \mathbb{E}\left[(\mathbf{I} - \eta_0\Sigma)^{j-i}(f_i - f_\rho^*) \otimes (f_i - f_\rho^*)\right].$$

*Proof.* We prove a more generalized version: For any fixed operator $\mathbf{A} : \mathcal{H} \to \mathcal{H}$,

$$\mathbb{E}\left[(f_i - f_\rho^*) \otimes \mathbf{A}(f_j - f_\rho^*)\right] = \mathbb{E}\left[(f_i - f_\rho^*) \otimes \mathbf{A}(\mathbf{I} - \eta_0\Sigma)^{j-i}(f_i - f_\rho^*)\right];$$
$$\mathbb{E}\left[\mathbf{A}(f_j - f_\rho^*) \otimes (f_i - f_\rho^*)\right] = \mathbb{E}\left[(\mathbf{I} - \eta_0\Sigma)^{j-i}\mathbf{A}(f_i - f_\rho^*) \otimes (f_i - f_\rho^*)\right].$$

For the base case $i = j$, it naturally holds.

Now we assume the lemma holds for $j = i + k \geq 1$, then we prove it for $j = i + k + 1$.

Notice that $K_{\mathbf{x}_t}$ and $\Xi_t$ are independent of $f_i - f_\rho^*$ for any $0 \le i < t$, we have

$$
\begin{aligned}
\mathbb{E}\left[(f_i - f_\rho^*) \otimes \boldsymbol{A}\left(f_{i+k+1} - f_\rho^*\right)\right] &= \mathbb{E}\left[(f_i - f_\rho^*) \otimes \boldsymbol{A}\left(\left(\boldsymbol{I} - \eta_0 K_{\mathbf{x}_{i+k+1}} \otimes K_{\mathbf{x}_{i+k+1}}\right) + \eta_0 \Xi_{i+k+1}\right)\left(f_{i+k} - f_\rho^*\right)\right] \\
&= \mathbb{E}\left[(f_i - f_\rho^*) \otimes \boldsymbol{A}\mathbb{E}\left[\left(\left(\boldsymbol{I} - \eta_0 K_{\mathbf{x}_{i+k+1}} \otimes K_{\mathbf{x}_{i+k+1}}\right) + \eta_0 \Xi_{i+k+1}\right)\left(f_{i+k} - f_\rho^*\right)\right]\right] \\
&= \mathbb{E}\left[(f_i - f_\rho^*) \otimes \left(\boldsymbol{A}(\boldsymbol{I} - \eta_0 \boldsymbol{\Sigma})\right)\left(f_{i+k} - f_\rho^*\right)\right] \\
&= \mathbb{E}\left[(f_i - f_\rho^*) \otimes \boldsymbol{A}(\boldsymbol{I} - \eta_0 \boldsymbol{\Sigma})^{k+1}\left(f_i - f_\rho^*\right)\right].
\end{aligned}
$$

Similarly, we have

$$
\mathbb{E}\left[\left(f_{i+k+1} - f_\rho^*\right) \otimes \left(f_i - f_\rho^*\right)\right] = \mathbb{E}\left[\boldsymbol{A}(\boldsymbol{I} - \eta_0 \boldsymbol{\Sigma})^{k+1}\left(f_i - f_\rho^*\right) \otimes \left(f_i - f_\rho^*\right)\right],
$$

from which the lemma is proved.

$\square$

**Lemma E.2.** *Consider SGD with averaged iterates defined in* (17). *Suppose $\rho$ satisfy* (29). *Then for any $0 \le t \le n$, we have*

$$
\mathbb{E}\left[f_t^b \otimes f_t^v\right] = \mathbf{0}.
$$

*Proof.* Solving the recursion of $f_t^b$ and $f_t^v$ defined in (5) and (6), we have

$$
f_t^b = \left(\prod_{i=1}^t (\boldsymbol{I} - \eta_0 K_{\mathbf{x}_i} \otimes K_{\mathbf{x}_i})\right) f_0, \quad f_t^v = \eta_0 \sum_{i=1}^t \left(\prod_{j=i+1}^t (\boldsymbol{I} - \eta_0 K_{\mathbf{x}_j} \otimes K_{\mathbf{x}_j})\right) \Xi_i.
$$

By the special property of $\rho$, we have $y_t - \left\langle f_\rho^*, K_{\mathbf{x}_t}\right\rangle_{\mathcal{H}}$ is independent to $K_{\mathbf{x}_i}$ for any $0 \le i \le n$, thus we have

$$
\begin{aligned}
\mathbb{E}\left[f_t^b \otimes f_t^v\right] &= \mathbb{E}\left[\left(\prod_{k=1}^t (\boldsymbol{I} - \eta_0 K_{\mathbf{x}_k} \otimes K_{\mathbf{x}_k})\right) f_0 \otimes \eta_0 \sum_{i=1}^t \left(\prod_{j=i+1}^t (\boldsymbol{I} - \eta_0 K_{\mathbf{x}_j} \otimes K_{\mathbf{x}_j})\right) \Xi_i\right] \\
&= \mathbb{E}\left[\left(\prod_{k=1}^t (\boldsymbol{I} - \eta_0 K_{\mathbf{x}_k} \otimes K_{\mathbf{x}_k})\right) f_0 \otimes \eta_0 \sum_{i=1}^t \left(\prod_{j=i+1}^t (\boldsymbol{I} - \eta_0 K_{\mathbf{x}_j} \otimes K_{\mathbf{x}_j})\right)\left(y_i - \left\langle f_\rho^*, K_{\mathbf{x}_i}\right\rangle_{\mathcal{H}}\right) K_{\mathbf{x}_i}\right] \\
&= \eta_0 \sum_{i=1}^t \mathbb{E}\left[\left(\prod_{k=1}^t (\boldsymbol{I} - \eta_0 K_{\mathbf{x}_k} \otimes K_{\mathbf{x}_k})\right) f_0 \otimes \left(\prod_{j=i+1}^t (\boldsymbol{I} - \eta_0 K_{\mathbf{x}_j} \otimes K_{\mathbf{x}_j})\right) K_{\mathbf{x}_i}\right] \mathbb{E}\left[\left(y_i - \left\langle f_\rho^*, K_{\mathbf{x}_i}\right\rangle_{\mathcal{H}}\right)\right] \\
&= \mathbf{0}.
\end{aligned}
$$

$\square$

**Lemma E.3.** *Consider SGD with averaged iterates defined in* (17). *Then for any $0 \le t \le n$, we have*

$$
\mathbb{E}\left[f_t^b \otimes f_t^b\right] \succeq \tilde{f}_t^b \otimes \tilde{f}_t^b;
$$
$$
\mathbb{E}\left[f_t^v \otimes f_t^v\right] \succeq \mathbb{E}\left[\tilde{f}_t^v \otimes \tilde{f}_t^v\right].
$$

*Proof.* We prove this by induction on $t$. The base case naturally holds. Assuming the lemma holds for $t - 1$, we prove this also holds for $t$.

Notice that $y_t - \left\langle f_\rho^*, K_{\mathbf{x}_t}\right\rangle_{\mathcal{H}}$, $K_{\mathbf{x}_t}$ and $f_i$ are independent to each other for any $0 \le i < t$, thus we have

$$
\begin{aligned}
&\mathbb{E}\left[f_t^v \otimes f_t^v\right] \\
&= \mathbb{E}\left[\left((\boldsymbol{I} - \eta_0 K_{\mathbf{x}_t} \otimes K_{\mathbf{x}_t}) f_{t-1}^v + \eta_0 \Xi_t\right) \otimes \left((\boldsymbol{I} - \eta_0 K_{\mathbf{x}_t} \otimes K_{\mathbf{x}_t}) f_{t-1}^v + \eta_0 \Xi_t\right)\right] \\
&= \mathbb{E}\left[(\boldsymbol{I} - \eta_0 \boldsymbol{\Sigma})\left(f_{t-1}^v \otimes f_{t-1}^v\right)(\boldsymbol{I} - \eta_0 \boldsymbol{\Sigma})\right] + \eta_0^2[(K_{\mathbf{x}_t} \otimes K_{\mathbf{x}_t})\left(f_{t-1}^v \otimes f_{t-1}^v\right)(K_{\mathbf{x}_t} \otimes K_{\mathbf{x}_t})] + \eta_0^2 \mathbb{E}\left[\Xi_t \otimes \Xi_t\right] \\
&\succeq \mathbb{E}\left[(\boldsymbol{I} - \eta_0 \boldsymbol{\Sigma})\left(f_{t-1}^v \otimes f_{t-1}^v\right)(\boldsymbol{I} - \eta_0 \boldsymbol{\Sigma})\right] + \eta_0^2 \mathbb{E}\left[\Xi_t \otimes \Xi_t\right] \\
&= \mathbb{E}\left[\tilde{f}_t^v \otimes \tilde{f}_t^v\right].
\end{aligned}
$$

Similarly, we have $\mathbb{E}\left[f_t^b \otimes f_t^b\right] \succeq \tilde{f}_t^b \otimes \tilde{f}_t^b$, from which the lemma is proved.

$\square$

Combining the three lemmas above, we have

$$\mathbb{E}\left[\langle\left(\overline{f_n} - f_\rho^*\right), \Sigma\left(\overline{f_n} - f_\rho^*\right)\rangle_{\mathcal{H}}\right]$$

$$= \mathbb{E}\left[\operatorname{tr}\left(\Sigma\frac{1}{n}\sum_{i=0}^{n-1}\left(f_i - f_\rho^*\right) \otimes \frac{1}{n}\sum_{j=0}^{n-1}\left(f_j - f_\rho^*\right)\right)\right]$$

$$= \operatorname{tr}\left(\Sigma\frac{1}{n^2}\sum_{i=0}^{n-1}\sum_{j=0}^{n-1}\mathbb{E}\left[\left(f_i - f_\rho^*\right) \otimes \left(f_j - f_\rho^*\right)\right]\right)$$

$$= \operatorname{tr}\left(\Sigma\frac{1}{n^2}\sum_{i=0}^{n-1}\left(\sum_{j=0}^{i}\mathbb{E}\left[\left(I - \eta_0\Sigma\right)^{i-j}\left(f_j^b + f_j^v\right) \otimes \left(f_j^b + f_j^v\right)\right] + \sum_{j=i+1}^{n-1}\mathbb{E}\left[\left(f_i^b + f_i^v\right) \otimes \left(I - \eta_0\Sigma\right)^{i-j}\left(f_i^b + f_i^v\right)\right]\right)\right)$$

$$= \operatorname{tr}\left(\Sigma\frac{1}{n^2}\sum_{i=0}^{n-1}\sum_{j=0}^{i}\mathbb{E}\left[\left(I - \eta_0\Sigma\right)^{i-j}f_j^b \otimes f_j^b + \left(I - \eta_0\Sigma\right)^{i-j}f_j^v \otimes f_j^v\right]\right)$$

$$+ \operatorname{tr}\left(\Sigma\frac{1}{n^2}\sum_{i=0}^{n-1}\sum_{j=i+1}^{n-1}\mathbb{E}\left[f_i^b \otimes \left(I - \eta_0\Sigma\right)^{i-j}f_i^b + f_i^v \otimes \left(I - \eta_0\Sigma\right)^{i-j}f_i^v\right]\right)$$

$$\geq \operatorname{tr}\left(\Sigma\frac{1}{n^2}\sum_{i=0}^{n-1}\sum_{j=0}^{i}\mathbb{E}\left[\left(I - \eta_0\Sigma\right)^{i-j}\tilde{f}_j^b \otimes \tilde{f}_j^b + \left(I - \eta_0\Sigma\right)^{i-j}\tilde{f}_j^v \otimes \tilde{f}_j^v\right]\right)$$

$$+ \operatorname{tr}\left(\Sigma\frac{1}{n^2}\sum_{i=0}^{n-1}\sum_{j=i+1}^{n-1}\mathbb{E}\left[\tilde{f}_i^b \otimes \left(I - \eta_0\Sigma\right)^{i-j}\tilde{f}_i^b + \tilde{f}_i^v \otimes \left(I - \eta_0\Sigma\right)^{i-j}\tilde{f}_i^v\right]\right)$$

$$= \operatorname{tr}\left(\Sigma\frac{1}{n^2}\sum_{i=0}^{n-1}\sum_{j=0}^{n-1}\left(\tilde{f}_i^b \otimes \tilde{f}_j^b + \mathbb{E}\left[\tilde{f}_i^v \otimes \tilde{f}_j^v\right]\right)\right)$$

$$= \left\langle\overline{\tilde{f}_n^b}, \Sigma\overline{\tilde{f}_n^b}\right\rangle_{\mathcal{H}} + \mathbb{E}\left[\left\langle\overline{\tilde{f}_n^v}, \Sigma\overline{\tilde{f}_n^v}\right\rangle_{\mathcal{H}}\right].$$

Thus, we have the following lower bound:

**Lemma E.4.** *2 Consider SGD with averaged iterates defined in* (17)*. Suppose that* $\eta_0 \leq \frac{1}{\lambda_1}$*. Then we have*

$$\max_\rho \mathbb{E}_{(\mathbf{x},y)\sim\rho}\|\overline{f_n} - f_\rho^*\|_{L^2}^2$$

$$\geq \frac{1}{n^2\eta_0^s}\max_{i\in\mathbb{N}_+}\left\{(1 - (1 - \lambda_i\eta_0)^n)^2(\lambda_i\eta_0)^{s-2}\right\}\left\|\Sigma^{\frac{1-s}{2}}(f_0 - f_\rho^*)\right\|_{\mathcal{H}}^2 + \sigma^2\left(\frac{1}{16}\frac{k^*}{n} + \frac{1}{64}\sum_{i=k^*+1}^{+\infty}n\eta_0^2\lambda_i^2\right),$$

*where* $k^* = \max_{i\in\mathbb{N}_+}\{k : \eta_0\lambda_k \geq \frac{1}{n}\}$*.*

*Proof.* Recall the proof of Lemma C.1, we have

$$\left\langle\overline{\tilde{f}_n^b}, \Sigma\overline{\tilde{f}_n^b}\right\rangle_{\mathcal{H}} \geq \frac{1}{n^2\eta_0^s}\left\|\sum_{i=0}^{n-1}(I - \eta_0\Sigma)^i(\eta_0\Sigma)^{\frac{s}{2}}\right\|^2\left\|\Sigma^{\frac{1-s}{2}}(f_0 - f_\rho^*)\right\|_{\mathcal{H}}^2,$$

when we choose $f_\rho^*$ that makes

$$\left\|\sum_{i=0}^{n-1}(I - \eta_0\Sigma)^i(\eta_0\Sigma)^{\frac{s}{2}}\Sigma^{\frac{1-s}{2}}(f_0 - f_\rho^*)\right\|_{\mathcal{H}} = \left\|\sum_{i=0}^{n-1}(I - \eta_0\Sigma)^i(\eta_0\Sigma)^{\frac{s}{2}}\right\|\left\|\Sigma^{\frac{1-s}{2}}(f_0 - f_\rho^*)\right\|_{\mathcal{H}}.$$

That is, choose $f_\rho^* = f_0 - \left\| \boldsymbol{\Sigma}^{\frac{1-s}{2}}(f_0 - f_\rho^*) \right\|_{\mathcal{H}} \lambda_q^{\frac{s}{2}} e_q$, where $q = \arg\max_{i \in \mathbb{N}_+} \left\{ (1 - (1 - \lambda_i \eta_0)^n)^2 (\lambda_i \eta_0)^{s-2} \right\}$.

Recalling the proof of Lemma C.3, we have

$$\mathbb{E}\left[\left\langle \overline{\tilde{f}_n^v}, \boldsymbol{\Sigma}\overline{\tilde{f}_n^v} \right\rangle_{\mathcal{H}}\right] = \frac{1}{n^2} \mathrm{tr}\left( \sum_{j=1}^{n-1} \eta_0^2 \left( \sum_{i=j}^{n-1} (\boldsymbol{I} - \eta_0 \boldsymbol{\Sigma})^{i-j} \right)^2 \boldsymbol{\Sigma}\mathbb{E}\left[\Xi_i \otimes \Xi_i\right] \right)$$

$$= \frac{\sigma^2}{n^2} \mathrm{tr}\left( \sum_{j=1}^{n-1} \eta_0^2 \left( \sum_{i=j}^{n-1} (\boldsymbol{I} - \eta_0 \boldsymbol{\Sigma})^{i-j} \right)^2 \boldsymbol{\Sigma}^2 \right)$$

$$= \sigma^2 \left( \sum_{i=1}^{k^*} \frac{1}{n^2} \sum_{j=1}^{n-1} \left(1 - (1 - \eta_0 \lambda_i)^j\right)^2 + \sum_{i=k^*+1}^{+\infty} \frac{1}{n^2} \sum_{j=1}^{n-1} \left(1 - (1 - \eta_0 \lambda_i)^j\right)^2 \right)$$

$$\geq \sigma^2 \left( \sum_{i=1}^{k^*} \frac{1}{n^2} \sum_{j=\lceil\frac{n}{2}\rceil}^{n-1} \left(1 - \exp\left(-\eta_0 \lambda_{k^*} \frac{n}{2}\right)\right)^2 + \sum_{i=k^*+1}^{+\infty} \frac{1}{n^2} \sum_{j=\lceil\frac{n}{2}\rceil}^{n-1} \left(1 - (1 - \eta_0 \lambda_i)^{\frac{n}{2}}\right)^2 \right)$$

$$\geq \sigma^2 \left( \sum_{i=1}^{k^*} \frac{1}{n^2} \frac{n}{4} \left(1 - \exp\left(-\frac{1}{n}\frac{n}{2}\right)\right)^2 + \sum_{i=k^*+1}^{+\infty} \frac{1}{n^2} \frac{n}{4} \left(\frac{n}{4}\eta_0 \lambda_i\right)^2 \right)$$

$$\geq \sigma^2 \left( \frac{1}{16}\frac{k^*}{n} + \frac{1}{64} \sum_{i=k^*+1}^{+\infty} n\eta_0^2 \lambda_i^2 \right),$$

when we choose $k^* = \arg\max_{i \in \mathbb{N}_+} \{k : \eta_0 \lambda_k \geq \frac{1}{n}\}$. Thus, we have proved there exists a $\rho$ satisfying the inequality in the lemma, from which the lemma is proved.

$\square$

In the following, we show that in high-dimensional settings, when $s > 1$, there exists a region where SGD with averaged iterates cannot achieve optimality.

**Lemma E.5.** *Let $c_1 d^\gamma < n < c_2 d^\gamma$ for some fixed $\gamma > 0$ and absolute constants $c_1, c_2$. Consider $\mathcal{X} = \mathbb{S}^d$ and the marginal distribution $\mu$ to be the uniform distribution. Let $K$ be the inner product kernel on the sphere. Consider $\gamma, \sigma, \kappa, c_1, c_2$ and $s > 1$ as constants, the excess risk of the output of SGD averaged iterates $f_n^{avg}$ cannot achieve optimality.*

*Proof.* According to Lemma E.4, we have

$$\max_\rho \mathbb{E}_{(\mathbf{x},y)\sim\rho} \|f_n^{avg} - f_\rho^*\|_{L^2}^2$$

$$= \Omega\left( \frac{1}{n^2\eta_0^s} \max_{i\in\mathbb{N}_+} \left\{ (1 - (1 - \lambda_i\eta_0)^n)^2 (\lambda_i\eta_0)^{s-2} \right\} \left\| \boldsymbol{\Sigma}^{\frac{1-s}{2}}(f_0 - f_\rho^*) \right\|_{\mathcal{H}}^2 + \sigma^2\left( \frac{k^*}{n} + \sum_{i=k^*+1}^{+\infty} n\eta_0^2\lambda_i^2 \right) \right)$$

where $k^* = \arg\max_{i\in\mathbb{N}_+}\{k : \eta_0\lambda_k \geq \frac{1}{n}\}$.

Under kernel $K$, $k^*$ defined above satisfies $\lambda_{k^*+1} = \Theta\left(\lambda_{k^*}d^{-1}\right)$, so we assume $k^* = \Theta\left(d^p\right)$, $p \in \mathbb{N}$, then $\eta_0 = \Theta(d^{p+r}n^{-1})$, where $0 \leq r < \min\{1, \gamma - p\}$ is a arbitrary constant.

In the case $0 < s \leq 2$, using the inequality $1 - (1-x)^n \geq \frac{1}{2}\min\{1, nx\}$, according to the proof of Lemma 5.2, we have

$$\max_{i\in\mathbb{N}_+} \left\{ (1 - (1 - \lambda_i\eta_0)^n)^2 (\lambda_i\eta_0)^{s-2} \right\} = \Omega\left( \inf_{x_0 \leq x < dx_0} \max\left\{ x^{s-2}, n\left(\frac{x}{d}\right)^s \right\} \right) = \Omega\left( d^{(\frac{s}{2}-1)s}n^{2-s} \right),$$

where $x_0 = \arg\max_{0\leq x\leq 1} \min\{1, nx\}x^{\frac{s}{2}-1} = \frac{1}{n}$.

We know

$$\max_{i \in \mathbb{N}_+} \left\{ (1 - (1 - \lambda_i \eta_0)^n)^2 (\lambda_i \eta_0)^{s-2} \right\} = \Theta \left( d^{(\frac{s}{2} - 1)s} n^{2-s} \right)$$

if and only if $\eta_0 = \Theta(d^{p + \frac{s}{2}} n^{-1})$, $p \in \mathbb{N}$ and $p < \gamma - \frac{s}{2}$.

When $\eta_0 = \Theta(d^{p+r} n^{-1})$ with $0 < r < \min\{\frac{s}{2}, \gamma - p\}$, we know

$$\max_{i \in \mathbb{N}_+} \left\{ (1 - (1 - \lambda_i \eta_0)^n)^2 (\lambda_i \eta_0)^{s-2} \right\} = \Theta \left( d^{r(s-2)} n^{2-s} \right)$$

When $\eta_0 = \Theta(d^{p+r} n^{-1})$ with $\frac{s}{2} < r < \min\{1, \gamma - p\}$, we know

$$\max_{i \in \mathbb{N}_+} \left\{ (1 - (1 - \lambda_i \eta_0)^n)^2 (\lambda_i \eta_0)^{s-2} \right\} = \Theta \left( d^{(r-1)s} n^{2-s} \right)$$

By contrast, in the case $s > 2$, we have

$$\max_{i \in \mathbb{N}_+} \left\{ (1 - (1 - \lambda_i \eta_0)^n)^2 (\lambda_i \eta_0)^{s-2} \right\} = \Omega \left( 1 \right).$$

Using $\sigma^2 = \Omega(1)$, $\left\| \Sigma^{\frac{1-s}{2}} (f_0 - f_\rho^*) \right\|_{\mathcal{H}}^2 = \Omega(1)$, $\sum_{i=k^*+1}^{+\infty} \lambda_i^2 = \Theta(\lambda_{k^*+1})$, and $n = \Theta(d^\gamma)$, we have

$$\max_{\rho} \mathbb{E}_{(\mathbf{x}, y) \sim \rho} \| f_n^{avg} - f_\rho^* \|_{L^2}^2$$

$$= \begin{cases}
\Omega \left( d^{p-\gamma} \right), & 0 < s \le 1, \gamma \in [p(s+1), p(s+1)+s) \\
\Omega \left( d^{-(p+1)s} \right), & 0 < s \le 1, \gamma \in [p(s+1)+s, p(s+1)+s+1) \\
\Omega \left( d^{p-\gamma} \right), & 1 < s < 2, \gamma \in [p(s+1), p(s+1)+1) \\
\Omega \left( d^{\frac{-(p+1)s+(p-\gamma)+s-1}{2}} \right), & 1 < s < 2, \gamma \in [p(s+1)+1, p(s+1)+2s-1) \\
\Omega \left( d^{-(p+1)s} \right), & 1 < s < 2, \gamma \in [p(s+1)+2s-1, p(s+1)+s+1) \\
\Omega \left( d^{p-\gamma} \right), & s \ge 2, \gamma \in [\frac{p(s+1)}{s-1}, \frac{p(s+1)+\frac{s}{2}}{s-1}) \\
\Omega \left( d^{p + \frac{s-4}{s+2} \gamma - \frac{2ps+2p+s}{s+2}} \right), & s \ge 2, \gamma \in [\frac{p(s+1)+\frac{s}{2}}{s-1}, \frac{p(s+1)+s+1}{s-1})
\end{cases} \tag{29}$$

$\square$

Similarly, we can prove that when $n \gg d$, SGD with averaged iterates is suboptimal for any $s > 2$.

# F. Graphical illustration and Summary of Optimal Region

## F.1. Graphical illustration of Theorem 5.1 and Theorem 5.2

In Figure 2, we provide a visual illustration of the convergence rate curves of the excess risk for SGD in the high-dimensional setting $n \asymp d^\gamma$, under source conditions with $s = 0.5, 1, 5$. For comparison, we also include KRR convergence curves and the minimax optimal rate curves. These references highlight the saturation phenomenon of KRR for large $s$, and the optimality of SGD across $s > 0$.

## F.2. Summary of Optimal Region

In Table 1, we summarize the optimality regions (with respect to the source condition parameter $s$) of various algorithms, including SGD, KRR, and GD with early stopping under both high-dimensional and asymptotic settings.

**In the high-dimensional setting:**

- SGD is optimal for $s > 0$ (Theorem 5.1 and Theorem 5.2).

- KRR is optimal for $0 < s \le 1$ (Zhang et al., 2024b), but for $s > 1$, there exists a range of $\gamma$ where it fails to be optimal (Zhang et al., 2024b).

- GD with early stopping is optimal for all $s > 0$ (Lu et al., 2024b).

**In the asymptotic setting:**

- SGD is optimal for $s \ge \frac{1}{d+1}$ (Theorem 5.3 and Theorem 5.4).

- KRR is optimal for $0 < s \le 2$ (Caponnetto & De Vito, 2007; Smale & Zhou, 2007; Blanchard & Mücke, 2018; Li et al., 2022; Zhang et al., 2024a), but not for $s > 2$ (Cui et al., 2021; Li et al., 2023).

- GD with early stopping is optimal for all $s > 0$ (Raskutti et al., 2014).

| | **High-Dimensional Setting**$(n \asymp d^\gamma)$ | **Asymptotic Setting**$(n \gg d)$ |
|---|---|---|
| SGD | $s > 0$ | $s \ge \frac{1}{d+1}$ |
| KRR | $0 < s \le 1$ | $0 < s \le 2$ |
| GD with Early Stopping | $s > 0$ | $s > 0$ |

*Table 1.* Optimal Region for Algorithms mentioned.

Learning Curves of Stochastic Gradient Descent in Kernel Regression

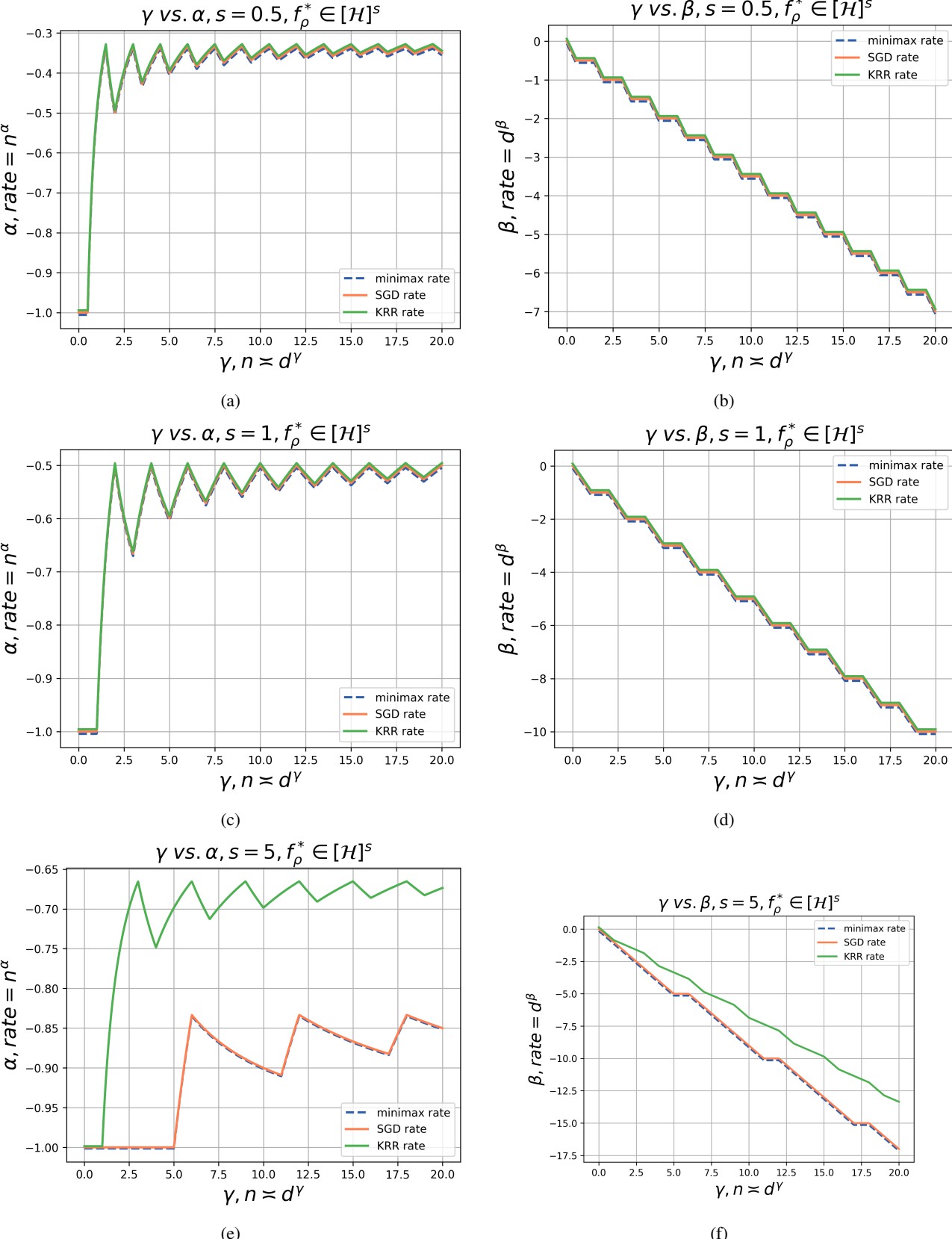

*Figure 2.* Convergence Rate of SGD and KRR with respect to $n$ and $d$, we present 6 graphs corresponding to source conditions with $s = 0.5, 1, 5$. The x-axis represents the asymptotic scaling $\gamma : n \asymp d^\gamma$; the y-axis represents the convergence rate of excess risk: excess risk $\asymp n^\alpha$ or excess risk $d^\beta$.

