# OpenReview forum: "Learning Curves of Stochastic Gradient Descent in Kernel Regression"
_ICML.cc/2025/Conference — ICML 2025 poster_

### Official Review · Reviewer_2WFu · 2025-02-27

**Overall Recommendation:** 4

**Summary:**

This paper analyzes the excess risk and the minimax lower bound of single-pass stochastic gradient descent (SGD) in kernel regression under the combinations of the following settings:

1) the model is well-specified or misspecified ( the source condition constant $s<1$ or $s\geq 1$ );

2) the number of data is in proportional/large-dimension regime ($n\sim d^\gamma$ or $n\gg d^\gamma$ for some constant $\gamma>0$);

3) the online (single-batch) SGD is run on exponentially decaying step size schedule or constant step size with averaged iterates.

There are several implications from the above results, for example, SGD achieves min-max optimal rate in almost all the cases, in contrast to the saturation effect on the kernel ridge regression (KRR), and it is also the first theoretical proof showing exponential decaying step size schedule is better than iterative averaging method in the context of kernel setting.

**Claims And Evidence:**

Yes, the theoretical claims are proved in the paper and validation experiments are shown.

**Essential References Not Discussed:**

Most relevant related works are cited.

**Experimental Designs Or Analyses:**

Yes, they are sound and valid.

**Methods And Evaluation Criteria:**

Yes.

**Other Comments Or Suggestions:**

Maybe a summarising table about the results on each combination of settings could help the presentation. This would offer a more macroscopic view on the contributions/findings of the paper.

**Other Strengths And Weaknesses:**

The paper is written in a precise and concise way, the concepts and intuitions are well-elaborated, so the paper is easy to follow.

**Questions For Authors:**

I am interested in potential extensions of the current results presented in this paper:

1. What would be the technical difficulties to extend the result to non-dot-product kernels?

2. Could one extend the analysis of the online SGD training to other spectral algorithms mentioned in [1]?


---
[1] Lu, Weihao, Yicheng Li, and Qian Lin. "On the Saturation Effects of Spectral Algorithms in Large Dimensions." Advances in Neural Information Processing Systems 37 (2025): 7011-7059.

**Relation To Broader Scientific Literature:**

The contribution is significant in kernel method, online learning and also to the broader scope of neural network training under the neural tangent kernel (NTK) framework.

As pointed out in the paper, saturation effect and the learning curve for KRR / kernel gradient flow is known. But such results on online SGD setting is novel and brings important insights to the field.

**Theoretical Claims:**

I check the proof from the main text as well as the proof in the appendix up to section B. I might have time to proof-read the rest, but the proof seems correct so far. The technique is a standard bias-variance decomposition with a careful modification for the settings in interest.

---

> ### Author Rebuttal · Authors · 2025-03-31
>
> We would like to extend our sincere appreciation to you for the thorough review and valuable feedback. We are grateful that you not only accurately summarized our contributions but also expressed strong appreciation. We would like to address your suggestion and the question you raised regarding possible extensions of our research.
> ***
> # Author's Response to Suggestion:
>
> >*1. "Maybe a summarising table about the results on each combination of settings could help the presentation."*
>
> This is a meaningful suggestion. We will include tables both summarizing our results and comparing them with prior work to facilitate a better understanding of this paper. Due to space constraints, this content will be added to the Appendix.
> ***
> # Author's Response to Questions
> ## Author's Response to Question 1:
>
> >*"1. What would be the technical difficulties to extend the result to non-dot-product kernels?"*
>
> Extending our proof to general kernels only requires verifying two conditions: (1) capacity condition or the eigenvalue decay rate (EDR) of the target kernel, and (2) a distributional condition controlling the residual noise $f _t^{b(1)}$ and $f _t^{v(1)}$, such as the kernel is bounded, or $\mathbb{E}[K(\mathbf{x},\mathbf{x} )K _{\mathbf{x}}\otimes K _{\mathbf{x}}]\preceq \kappa \mathbb{E}[K _{\mathbf{x}}\otimes K _{\mathbf{x}}]$, or $\mathbb{E} \left \langle f,K _{\mathbf{x} } \right \rangle ^4\le \kappa \left \Vert f \right \Vert _{\mathcal{H} } ^2$.
>  Once the EDR of the target kernel is known and the kernel satisfies the required distributional condition, our analysis can be applied by analyzing the iterative behavior of the residual noise under these corresponding conditions.
>
> ## Author's Response to Question 2:
>
> >*2. "Could one extend the analysis of the online SGD training to other spectral algorithms mentioned in [1]?"*
>
> This is a deep and interesting question that points to a promising direction for future work. Spectral algorithms construct the estimator $\hat{f} _{\lambda}=\phi _{\lambda}(T _X)g _Z$ using the sample basis function $g _Z=\frac{1}{n}\sum _{i=1}^{n}y _iK(x _i,\cdot)$, the sample covariance operator $T _X=\frac{1}{n}\sum _{i=1}^{n}K _{x _i}K _{x _i}^{\ast}$, and a filter function $\phi _{\lambda}$. To obtain a stochastic version of the corresponding spectral algorithm, one may first reformulate the spectral estimator as the solution to a variational problem on a functional manifold. Then, by selecting an appropriate Bregman divergence, the stochastic approximation to the spectral method can be obtained by stochastic mirror descent. This perspective enables the analysis of the implicit regularization induced by stochastic mirror descent on the functional manifold. Furthermore, by examining how the data structure constrains the optimization trajectory, one can derive generalization bounds.
>
> **References:**
>
> [1] Lu et al. "On the Saturation Effects of Spectral Algorithms in Large Dimensions." Advances in Neural Information Processing Systems 37 (2025): 7011-7059.

---

> > ### Comment · Reviewer_2WFu · 2025-04-05
> >
> > Thank you for your answer.
> >
> > I would maintain my current score and wish the authors the best on the remaining review period.

---

### Official Review · Reviewer_6fC3 · 2025-03-01

**Overall Recommendation:** 3

**Summary:**

This paper studies the generalization performance of kernel regression trained by online / single-pass SGD and compares it with offline methods such as ridge regression. Specifically, the analysis is conducted under a standard source condition assumption on the target (including the misspecified case) and focuses on dot-product kernels with uniform data on the sphere.

Under these assumptions, the authors derive excess risk decay rates in the input dimension $d$ under two scalings of sample size $n$: polynomial $n=\Theta(d^{\gamma})$ and assymptotic $n\gg d$. The main results are:

- Optimality of SGD in well-specified regimes: They show that using exponentially decaying step sizes allows SGD to achieve minimax-optimal rates for smooth targets $s\geq 1$, avoiding the so-called saturation effect that plagues certain offline algorithms.

- Efficiency in misspecified problems. For problems where the target is less smooth $s\in(0,1)$, a constant step size with iterates averaged is proven to match the minimax lower bounds in high dimensions.

They also provide a comparison with offline KRR under both scalings, showing that SGD can outperform KRR, especially when the target is sufficiently smooth.

**Claims And Evidence:**

The claims in this paper are rigorous mathematical results. Numerical simulations are also provided as an illustration of the mathematical results.

**Essential References Not Discussed:**

1. While the authors properly acknowledge the related literature which is closest to their work on the technical side, there are major omissions concerning related results which are directly relevant to this discussion, both when it comes to the learning curves of dot-product kernel on spherical data in the regime $n=\Theta(d^{\gamma})$ (Bordelon et al. 2020) and to the learning curves of KRR under source and capacity conditions in the $n\gg d$ regime (Cui et al. 2021).

    Indeed, these contain results which are discussed here, but that precede other references used by the authors. For example, Cui et al. (2021) provided a full picture of the rates of the asymptotic rates of KRR as a function of the noise level and regularization, including the suboptimal rate of KRR when $s>2$ at optimal quoted from Li et al. (2023).

2. Although you cite (Pillaud-Vivien et al., 2018) in the end of Section 5, it would be good to also mention it in the related works, when discussing SGD being suboptimal.

- [Bordelon et al. 2020] Bordelon, Blake, Abdulkadir Canatar, and Cengiz Pehlevan. "Spectrum dependent learning curves in kernel regression and wide neural networks." In International Conference on Machine Learning, pp. 1024-1034. PMLR, 2020.

- [Cui et al. 2021] Cui, H., Loureiro, B., Krzakala, F., & Zdeborová, L. (2021). Generalization error rates in kernel regression: The crossover from the noiseless to noisy regime. Advances in Neural Information Processing Systems, 34, 10131-10143.

**Experimental Designs Or Analyses:**

N/A.

**Methods And Evaluation Criteria:**

N/A.

**Other Comments Or Suggestions:**

The absence of numbers in the equations make it hard to refer to them. But here are a list of possible typos:

- Related work, L069 right column, I think there is a $-$ sign missing in $n^{\frac{s\alpha}{s\alpha+1}}$
- Section 3, L127 right column: the adjoint should be $T^{\star}:\mathcal{H}\to L^{2}$.
- 1st equation of Section 6.2: $f^{b}=f_{\star}$ and $\tilde{f}^{b}=f_{\star}$.
- 2nd equation of Section 5.2: double check the rate $\frac{2d+2}{3d+1}$, I think it might be $\frac{2d+2}{3d+2}$ instead.

**Other Strengths And Weaknesses:**

The paper is well-written and easy to follow. The results are interesting. The major weaknesses is that the discussion is specified to dot-product kernels in the sphere, while I believe that part of the discussion should generalize to more general settings. I also consider the omission of relevant related literature as an important weakness.

**Questions For Authors:**

- What are the challenges of stating the results of Section 5.2. for general source and capacity conditions of the kernel? What is exactly the role played by the spherical data here?

- I think it would be a nice addition to the manuscript to have either a plot or a schematic drawing summing the behaviour of the error curve of SGD in the different polynomial regimes $n=\Theta(d^{\gamma})$ at increasing $\gamma$.

**Relation To Broader Scientific Literature:**

- The study of excess risk rates for kernel regression is a classical topic in machine learning theory, with an extensive literature. The goal of this work is comparing the rates obtained by training a kernel method using one-pass SGD with two benchmarks: the minimax rates by and the KRR rates with a given regularisation. These have been derived in previous works both under generic source and capacity conditions e.g. (Caponnetto & De Vito, 2007; Cui et al. 2021; Lu et al. 2024), but also specifically for dot product kernels in the sphere (Bordelon et al. 2020; Bietti & Bach, 2021; Misiakiewicz, 2022)

- The proofs also leverage previous results on the analysis of SGD for kernel methods (Dieuleveut & Bach 2016).

**Theoretical Claims:**

I skimmed through the proofs of the results and did not spot any issue. The proofs are standard, and consist in bounding the bias and variance decomposition of the predictor by leveraging previous results in the literature.

---

> ### Author Rebuttal · Authors · 2025-03-31
>
> Thank you for your constructive comments and recognition of our contributions. Below, we address your concerns and questions in detail.
> ***
> # Author's Response to Essential References:
>
> >*1. "Related References Bordelon et al. (2020) and Cui et al. (2021)":*
>
> Thank you for sharing these important and relevant works in this area. They proposed the description of learning curve's exact order and have inspired subsequent efforts toward the characterization of the optimal learning curve. We have now included Bordelon et al. (2020) and Cui et al. (2021) in the revised version of the paper. We discuss the contributions of these works and their connections to our research below.
>
> Bordelon et al. (2020) studied the learning curves of dot-product kernels on spherical data in the high-dimensional regime $n\asymp d^{\gamma}$. They observed that errors in spectral modes associated with large eigenvalues decrease more rapidly as the sample size $n$ increases, while the errors in spectral modes with small eigenvalues remain nearly constant until $n$ reaches the degeneracy of the corresponding mode.
>
> Cui et al. (2021) provided a comprehensive characterization of the asymptotic rates of KRR with respect to the capacity and source conditions, the regularization parameter, and the noise level in the Gaussian design setting. Their analysis implies that KRR is suboptimal when $s>2$. Li et al. (2023) further proved this conclusion for general continuous positive-definite kernels.
>
> >*2. "Mentioning Pillaud-Vivien et al. (2018) in the Related Works":*
>
> Thank you for this suggestion. We have now mentioned it in the related works when discussing the suboptimality of SGD, noting that the introduction of multi-pass strategies can lead to a broader region of optimality.
>
> ***
> # Author's Response to Suggestions:
>
> Thank you for pointing out the typos, which definitely helps us improve the paper in a concrete way. We have made the following revisions:
> - Revised L069 right column as $n^{-\frac{s\alpha}{s\alpha+1}}$.
> - Revised L127 right column as $T^{\ast}:  \mathcal{H}\rightarrow L^2 $.
> - Revised 1st equation of Section 6.2 as $f^{b} _0=f _{\rho}^{\ast}$ ,$\tilde{f} _{0}^{b}=f _{\rho}^{\ast}$.
> - Revised 2nd equation of Section 5.2 as $n^{-\frac{2d+2}{3d+2}}$.
> ***
> # Author's Response to Questions
> ## Author's Response to Question 1:
>
> >*1. "What are the challenges of stating the results of Section 5.2. for general source and capacity conditions of the kernel?"*
>
> Thank you for raising the question of whether our asymptotic results can be extended to general source and capacity conditions of the kernel.
> First, we note that the excess risk bound derived in Appendix B relies only on the assumption that the kernel is bounded, and it already provides a general formulation.
> Once the target kernel satisfies certain capacity conditions, along with a distributional condition that controls the residual noise of
> $f_t^{b(1)}$ and $f_t^{v(1)}$, such as the boundedness of the kernel, or conditions like $\mathbb{E}[K(\mathbf{x},\mathbf{x} )K_{\mathbf{x}}\otimes K_{\mathbf{x}}]\preceq \kappa \mathbb{E}[K_{\mathbf{x}}\otimes K_{\mathbf{x}}]$, or $\mathbb{E} \left \langle f,K_{\mathbf{x} } \right \rangle ^4\le \kappa \left \Vert f \right \Vert _{\mathcal{H} } ^2$,
> our analysis can be directly applied to obtain corresponding convergence rates.
>
> >*2. "What is exactly the role played by the spherical data here?"*
>
> We did not present the general conclusion separately, but continued using spherical data in order to maintain focus on the central theme of our paper. The main goal of this work is to understand the generalization curve of SGD under the NTK. Our analysis in the asymptotic setting aims to answer the following question: Can the SGD algorithm consistently achieve optimality across all scalings of $n$, particularly when liberated from the $n\asymp d^{\gamma}$ constraints? For this reason, we adopted the assumptions in $n\asymp d^{\gamma}$ setting throughout the paper.
> We acknowledge that the assumption of a uniform input distribution on spheres is restrictive in high-dimensional scenarios. Our primary motivations for adopting this setting are that the harmonic analysis in the sphere is clearer and more concise, the analysis of Mercer’s decomposition for general kernels in high-dimensional settings is challenging, and few results are available. Due to these reasons, most existing analyses in high-dimensional setting also utilize spherical data.
> Your suggestion to extend our analysis to more general kernels is indeed a valuable and challenging direction, and we will pursue this in our future research.
>
> ## Author's Response to Question 2:
>
> >*"Addition of a plot summarizing the behavior of the error curve of SGD."*
>
> This is a valuable suggestion. We have now included in the appendix the curves of the convergence rate on the excess risk of SGD as $\gamma$ increases, under different values of $s$.

---

> > ### Comment · Reviewer_6fC3 · 2025-04-01
> >
> > I thank the authors for their rebutal. My questions were clarified, and I will keep my score.

---

### Official Review · Reviewer_sxnu · 2025-03-11

**Overall Recommendation:** 2

**Summary:**

This paper proves 4 results about the excess risks of least squares RKHS regression optimised by stochastic gradient descent. The settings of the four results are divided along two axes: firstly, the high-dimensional regime, in which the input dimension grows with the sample size, versus the fixed dimension regime, where the only the sample size grows, and secondly, the well-specified and misspecified settings. Comparing with lower bounds in existing works, optimality of their results are investigated in each case.

**Claims And Evidence:**

This is a theoretical paper, and hence the presentation of the mathematics and the correctness of the proofs are needed as evidence for the theoretical claims. I do not have big doubts that the results are correct, but the proofs are presented in a way that is rather hard to follow and I have doubts on the correctness of some parts of it. I gave up reading the proofs in the Appendix some way in, and I unfortunately can only think that there will be many more places where corrections or clarifications have to be made. See "Questions" sections for details.

**Essential References Not Discussed:**

There are some related papers that the authors have not discussed in the KRR literature, for example, [Blanchard and Mücke, 2018, Optimal Rates For Regularization Of Statistical Inverse Learning Problems], [Li et al., 2023, Optimal Rates for Regularized Conditional Mean Embedding Learning].

**Experimental Designs Or Analyses:**

This is a theoretical paper.

**Methods And Evaluation Criteria:**

This is a theoretical paper.

**Other Comments Or Suggestions:**

10R: "which states sufficiently wide neural network" -> "which states that sufficiently wide neural networks"

35R: "via an online fashion" -> "in an online fashion"

43R: "In specific" -> "Specifically"

51R: "resulting entirely different learning dynamics" -> "resulting in entirely different learning dynamics"

87L: "Why SGD can overcome" -> "Why can SGD overcome"

77R, 104R: "early stop" -> "early stopping"

138L: For consistency, $\rho_X$ should be $\rho_\mathcal{X}$.

146L: "denote ... be the operator" -> "denote by ... the operator"

114R: "By the Mercer's theorem" -> "By Mercer's theorem"

118R, 125R, 142R, 232L: For consistency, $L_2$ should be $L^2$.

195R: Seems a space is missing between the definition of $m$ and the next sentence?

287L: "optimaly" -> "optimally"

284R: space needed after "s,"

559: The subscript for $\langle\mathbf{x},\mathbf{x}\rangle$ should not be $L^2$, as $\mathbf{x}$ are not functions.

577: In (9), the second inequality is an equality.

636, (16): $f_t$ is only introduced later in (18), so it doesn't look good to use it here. If $f_t$ is replaced by general $f$ here, then the outer expectation should not be there, since $f$ would not be random and there is no other source of randomness.

686: $f^{(1)}_t$ should be $f^{b(1)}_t$.

768, 770: The operator does not live in $\mathcal{H}$, so you cannot take the $\mathcal{H}$-norm of the operators, you should take the operator norm.

**Other Strengths And Weaknesses:**

I think the writing could be very much improved - see Section "Other Comments Or Suggestions". The English is not such a big problem, as it cannot be expected that non-native speakers write in perfect English, but there are some inconsistencies with mathematical notations too.

All the results are given in expectation, and I think in general high-probability bounds are stronger, and personally preferred. This is a minor point.

**Questions For Authors:**

164R: I thought the multiplicities were $N(d,k)=\frac{(2h+d-2)(h+d-3)!}{h!(d-2)!}$? See [Müller, 1998, Analysis of Spherical Symmetries in Euclidean Spaces, p.28, Exercise 6]. Why are your multiplicities different?

The authors repeatedly refer to SGD as an "online algorithm", but this is very different to the usual usage of the word "online learning", whereby the data is fed to the algorithm sequentially, and one is concerned with the regret. I have personally never seen the term "online algorithm" being used for variants of gradient descent. If the authors and the other reviewers agree, I would strongly suggest to remove the word "online" to avoid confusion. The authors also repeatedly use the term "offline" for learning with explicit regularisation, or anything that is not based on variants of gradient descent, but I would consider their SGD set-up also as "offline", as we only have one static dataset.

567, (7): Are you sure that $\sum^\infty_{i=1}\langle\phi_i,T\phi_i\rangle_{L^2}=\sum^\infty_{i=1}\mathbb{E}[\langle\phi_i,K_\mathbf{x}\rangle^2_{L^2}]$? I get that $\sum^\infty_{i=1}\langle\phi_i,T\phi_i\rangle_{L^2}=\sum^\infty_{i=1}\mathbb{E}[K(\mathbf{x},\mathbf{x}')\phi_i(\mathbf{x})\phi_i(\mathbf{x}')]$ and $\sum^\infty_{i=1}\mathbb{E}[\langle\phi_i,K_\mathbf{x}\rangle^2_{L^2}]=\sum^\infty_{i=1}\mathbb{E}[K(\mathbf{x},\mathbf{x}')\phi_i(\mathbf{x}')K(\mathbf{x},\mathbf{x}'')\phi_i(\mathbf{x}'')]$, which are not the same. However, from $\sum^\infty_{i=1}\langle\phi_i,T\phi_i\rangle_{L^2}=\sum^\infty_{i=1}\mathbb{E}[K(\mathbf{x},\mathbf{x}')\phi_i(\mathbf{x})\phi_i(\mathbf{x}')]$, you can proceed with the reproducing property and the Cauchy-Schwarz inequality to get the desired bound.

589, (10): The second equality should be an inequality arising from the Cauchy-Schwarz inequality. They are not the same.

667, (23): I'm sorry if I'm missing something, but how does the positive definiteness of $\boldsymbol{\Sigma}$ here imply (23)?

671, (24): On the first line, shouldn't it be $\mathbb{E}[\langle f_i-f^*_\rho,\boldsymbol{\Sigma}(f_j-f^*_\rho)]\rangle_\mathcal{H}=\mathbb{E}[\langle f^v_i+f^b_i,\boldsymbol{\Sigma}(f^b_j+f^v_j)\rangle_\mathcal{H}]$ instead of $\mathbb{E}[\langle f_i-f^*_\rho,\boldsymbol{\Sigma}(f_j-f^*_\rho)]\rangle_\mathcal{H}=\mathbb{E}[\langle f^v_i+f^b_j,\boldsymbol{\Sigma}(f^b_i+f^v_j)\rangle_\mathcal{H}]$? How is it that $i$ and $j$ switch?

712, 733, 744, 748, 753, etc.: In the first term on the right hand side on the first line, what is random inside the expectation here? It seems to me that nothing is random, so you shouldn't be writing expectations right?

719: I don't understand (31). On the left-hand side we have subscript $t$, but on the right-hand side we have $i$ and $j$. Where do $i$ and $j$ come from? Does this bound hold for all $t$, $i$ and $j$? The preceding proofs suggest that all $i$ and $j$ should be replaced by $t$, see comment above for 671.

761: How do you obtain (37)? I guess you use $\eta_t=\frac{\eta_0}{2^{l-1}}$ and $m=\lceil\frac{n}{\log_2(n)}\rceil$ somehow? I'm sorry that I couldn't immediately see how this leads to (37). Is this obvious?

424: Where was $k^*$ defined? This seems to be the only place in the main body where $k^*$ is used, and it is not introduced. It is used quite heavily in the appendix but it is not introduced there either.

790, (42): Why is $\mathbb{E}[\Xi_i]=0$? On 630, $f^*_\rho$ was defined to be $\text{argmin}_{f\in[\mathcal{H}]^s}\mathcal{E}(f)$, not $\mathbb{E}[y\mid\cdot]$, so I don't think you can assume $\mathbb{E}[\Xi_i]=0$. Am I missing something? This has repercussions, e.g. in (43), where the cross-terms should not disappear.

813: Again, what is $k^*$?

**Relation To Broader Scientific Literature:**

The results would be interesting in the community of learning theory in RKHSs, the attention on which have been recently revived with connections to neural networks via NTKs and the study of benign overfitting. The study of properties of SGDs is much less explored than the closed form solutions, and if it can be shown that SGDs have provable advantages over closed form solutions, as this paper claims, that would be interesting.

**Theoretical Claims:**

See "Questions" section for details.

---

> ### Author Rebuttal · Authors · 2025-03-31
>
> Thank you very much for taking the time to carefully review our paper and for providing detailed and highly valuable feedback. Below, we respond to your questions and concerns one by one.
> ***
> # Response to Questions:
> >*1. 164R: $N(d,k)$.*
>
> The confusion stems from a typo in Line 142L, where we mistakenly wrote $\mathbb{R}^{d}$ instead of $\mathbb{R}^{d+1}$. In our paper, $\mathbb{S}^d$ actually denotes the unit sphere in $\mathbb{R}^{d+1}$. We have corrected this and confirm that the difference only introduces a constant factor, which does not affect the validity of our rates.
>
> >*2. Online algorithm.*
>
> We refer to SGD as an "online algorithm" to emphasize that it can process data in an i.i.d. streaming fashion, in order to distinguish it from "offline algorithm", which operates on a fixed dataset. We have now reduced the use of "online", only when we mention "offline". We have also added a clarification of "offline" to ensure the meanings of both terms are clear.
>
> >*3. 567, (7): $\sum _{i=1}^{\infty}\langle\phi _i,T\phi _i\rangle _{L^2} =\sum _{i=1}^{\infty} \mathbb{E}\left[\left\langle\phi _i,K _{\mathbf{x}}\right\rangle _{L^2}^2\right]$?*
>
> This is a typo, but the conclusion $\mathrm{tr} (T )=\mathbb{E}{K(X,X)}$ still holds. A revised proof is provided below:
> \begin{aligned}
> \sum _{i=1}^{\infty}\langle\phi _i,T \phi _i\rangle _{L^2}=\sum _{i=1}^{\infty} \mathbb{E} \left( \lambda _i^{1/2}\phi _i(X)\right)^2\overset{(a)}{=}\sum _{i=1}^{\infty} \mathbb{E} \left\langle K _X, \lambda _i^{1/2} \phi _i \right\rangle _{\mathcal{H}}^2\overset{(b)}{=}\mathbb{E}\left\langle K _X, K _X \right\rangle _{\mathcal{H}}，
> \end{aligned}
> where $(a)$ follows from the reproducing property, $(b)$ follows from Parseval's identity.
>
> >*4. 589, (10): Typo in Equa.*
>
>  We have corrected this.
>
> >*5. 667, (23): Why positive definiteness of $\boldsymbol{\Sigma}$ imply (23)?*
>
> >*6. 671, (24): $i$ and $j$?*
>
> >*7. 719, (31): $i$ and $j$?*
>
> We address Questions 5, 6 and 7 together. Thank you for pointing out the typo, that we mistakenly used subscript $i$ and $j$ instead of the iteration $t$. All $i$ and $j$ should be replaced by $t$ in Lemma B.1, Lemma B.2 and (76). We have corrected this, and it does not affect the validity, as only the decomposition at the $t$-th iteration are used. The key steps are provided below:
> By the positive definiteness of $\boldsymbol{\Sigma}$, $\left\langle f _{t}^{b} - f _{t}^{v}, \boldsymbol{\Sigma}\left(f _{t}^{b} - f _t^v\right)\right\rangle _\mathcal{H}\ge0$. This implies $\left\langle f _t^b,\boldsymbol{\Sigma}f _t^v\right\rangle _\mathcal{H}+\left\langle f _t^v , \boldsymbol{\Sigma}f _t^b\right\rangle _\mathcal{H}\le \left\langle f _t^b , \boldsymbol{\Sigma}f _t^b\right\rangle _\mathcal{H}+\left\langle f _t^v , \boldsymbol{\Sigma}f _t^v\right\rangle _\mathcal{H}$.
> Based on this, we can obtain the desired result in Lemma B.1. The proofs of Lemma B.2 and (76) follow the same reasoning.
>
> >*8. 712, 733, 744, 748, 753, etc.: $\mathbb{E}$?*
>
> We have removed the expectation where no randomness is involved.
>
> >*9. 761: (37)?*
>
> Below is an explanation of (37).
> Due to $\eta _0 \le \frac{1}{\lambda _1}$ and $\eta _i\le \eta _0$, it follows that $\mathbf{0} \preceq  \boldsymbol{I} -\eta _i\boldsymbol{\Sigma}\preceq \boldsymbol{I}$.
> For all $i$, $\boldsymbol{I} - \eta _i\boldsymbol{\Sigma}$ and $\boldsymbol{\Sigma}$ are positive semidefinite, self-adjoint and mutually commuting operators.
> Denote $\mathcal{M} _n= \prod _{i=1}^{n} (\mathbf{I} - \eta _i \boldsymbol{\Sigma})$, then $\mathcal{M} _n \boldsymbol{\Sigma} \mathcal{M} _n\preceq\cdots\preceq\mathcal{M} _m \boldsymbol{\Sigma}\mathcal{M} _m.$
>
> >*10. 424, 813: What is $k^{\ast}$?*
>
> Thank you for pointing out the inconsistent use of $k^{\ast}$. In Line 424, $k ^{\ast} = \max _{i\in\mathbb{N} _+} \lbrace k:\eta _0\lambda _k\ge \frac{1}{n}\rbrace $, which serves as an effective dimension (see Lemma D.4). While in Appendix B, $k^{\ast}$ can be any positive integer in order to present a more general result. The specific value is clarified in Appendix C and D. We have revised the paper to clearly indicate whether $k^{\ast}$ refers to.
>
> >*11. 790, (42): $\mathbb{E}[ \Xi _{t} ]=0$?*
>
> We apologize for the confusion caused by the improper definition of $f _{\rho}^{\ast}$ in Line 630.
> Although under the source condition given $s$, $f _{\rho}^*$ in line 630 is intended to refer to $\mathbb{E} [ y| \mathbf{x} ]$, this expression used was inappropriate. We have revised the paper to ensure consistent use of $f _{\rho}^*=\mathbb{E} [ y| \mathbf{x} ]$ throughout. Under this definition, $\mathbb E[\Xi _t ]=0$ is a mild assumption.
> ***
> # Response to Suggestions:
> We appreciate your constructive writing suggestions. All points have been addressed, and we have further refined the manuscript.
> ***
> # Response to Related Works:
> Thank you for sharing the references. They have been included in the revised version. Additionally, we have expanded our discussion of the KRR literature.

---

> > ### Comment · Reviewer_sxnu · 2025-04-03
> >
> > Dear authors,
> >
> > Thank you for your rebuttal. Although the specific points I raised were addressed, I think the manuscript should go through a thorough revision, and I maintain my evaluation of the paper. It seems the other reviewers are much more positive though, and I would not mind at all if it was accepted either!
> >
> > Best,
> > reviewer

---

### Official Review · Reviewer_bPCa · 2025-03-13

**Overall Recommendation:** 4

**Summary:**

This paper examines the problem of using SGD to train a kernel regressor. In particular, the paper considers the dot product kernel with input data that is uniformly distributed on the sphere. Assuming that the targets $y = f_*(x) + \varpepsilon$ for $f_*$ with certain smoothness properties, the paper shows that different forms of SGD can achieve the minimax optimality rate.

The paper empirically verifies this in a couple of situations.

**Claims And Evidence:**

The paper is primarily theoretical, with some experiments reinforcing their theoretical claims. The theoretical statements are detailed and rigorously written and have corresponding proofs. The results from the paper are convincing.

The importance of the upper bounds is also well justified.

**Essential References Not Discussed:**

In essence, the paper does not consider SGD until convergence. But an early stopped version of SGD. It would be beneficial for the paper to discuss related works on early stopped SGD (or even GD). I am more familiar with the GD literature so I list some here [A,B,C,D].

I think this is also an import paper on dot product kernel regression that is missing [E]

[A] Madhu S. Advani, Andrew M. Saxe, and Haim Sompolinsky. High-dimensional dynamics of generalization
error in neural networks. Neural Networks, 132:428–446, 2020.

[B] Alnur Ali, J. Zico Kolter, and Ryan J. Tibshirani. A Continuous-Time View of Early Stopping for Least
Squares Regression. In International Conference on Artificial Intelligence and Statistics, pages 1370–1378,
2019.

[C] Garvesh Raskutti, Martin J Wainwright, and Bin Yu. Early Stopping and Non-parametric Regression: An
Optimal Data-dependent Stopping Rule. Journal of Machine Learning Research, 15(11):335–366, 2014.

[D] Rishi Sonthalia, Jackie Loh, Elizaveta Rebrova. On regularization via early stopping for least squares regression. arXiv preprint arXiv:2406.04425. 2024.

[E] Xiao L, Hu H, Misiakiewicz T, Lu YM, Pennington J. Precise learning curves and higher-order scaling limits for dot product kernel regression. arXiv preprint arXiv:2205.14846. 2022 May 30:3.

**Experimental Designs Or Analyses:**

N/A

**Methods And Evaluation Criteria:**

The experiments are appropriate to support the theory.

**Other Comments Or Suggestions:**

**Typos**

The embedding operator Line 126 (Right) should map from $\mathcal{H} \to L^2$.

In Assumption 4.3 it should be clarified for a given $s$ and not for all $s$.

In Theorem 5.1 $f_\rho$ should be explicity defined. It would also be helpful to write 4.7 and 5.1 in the same lanague. Specifically, so that the conditions on $p$ and $\gamma$ are easier to compare. Currently they look similar, but I think there is a difference of $\pm 1$.

**Other Strengths And Weaknesses:**

**Strengths**

The results from the paper are quite strong. They show that optimal minimax rates can be achieve for a variety of well specified and mis-specified problems. This is quite a strong result and I think is sufficient for the paper to be published.

**Questions For Authors:**

Do you what happens if instead of stopping early the algorithms are run until convergence? Do they converge to the KKR solutions?

**Relation To Broader Scientific Literature:**

The paper mostly situated itself very well with respect to broader literature. They provide comprehensive cases against which to compare their results. Specifically, the lower bounds from prior works, as well as the fact that KRR does not meet these lower bounds. This helps strengthen their results, showing that the method they consider achieves the optimal rates.

**Theoretical Claims:**

I read the proof for property one and the beginning of the proof for Theorem 5.1. They look correct to me.

---

> ### Author Rebuttal · Authors · 2025-03-31
>
> We sincerely thank you for your careful review and positive feedback on our work. Your comments have provided concrete guidance for improving the paper and have been a great source of motivation for us. Below, we address your questions and concerns in detail.
> ***
> #  Author's Response to Essential References:
> Thank you for sharing these important related works with us, which have helped us to improve the completeness of our discussion.
> > *1. "Import paper on dot product kernel regression":*
>
> **Our Response:** [E] is indeed a key reference in the high-dimensional dot-product kernel regression. We have now added them in the revision of the paper. It studies the setting where $n\asymp d^{\gamma}$, provides a precise closed form formula for learning curves of dot-product KRR, and identifies the multiple descent phenomenon of KRR. The techniques presented in this paper are also highly beneficial for analyzing optimal convergence rates in high-dimensional scenarios.
>
> >*2. "Related works on early stopped GD":*
>
> **Our Response:** Thank you for providing references on early stopped GD, which enables readers to gain a more comprehensive understanding of the related techniques. We have incorporated a more detailed comparison with early stopped GD in the revision of the paper. In the following, we discuss the contributions and connections to our current work of the papers you shared.
>
> Specifically, references [A–D] examine the generalization performance of early stopped GD in kernel regression, collectively highlighting the close connection between early stopping and ridge regularization. They analyze early stopped GD under scaling conditions such as $n\asymp m$ or $n\asymp d$, where $n$ denotes the sample size, $d$ the input dimension, and $m$ the number of model parameters, emphasizing optimal stopping rules and generalization performance. Our paper complements these studies by considering single pass SGD within the NTK framework (infinite width, $m=\infty$), and explores the optimal learning curves under various ratios between sample size $n$ and input dimension $d$.
>
> >*3. "Related works on early stopped SGD":*
>
> **Our Response:** The single pass SGD we consider can be viewed as a form of early stopping, where each observation is used only once, and overfitting is avoided by making only a single pass through the data. Following your suggestion, we will expand the discussion with related works.
> ***
> # Author's Response to Comments or Suggestions:
> Thank you for your suggestions and for pointing out the typos. We have made the following revisions accordingly.
>
> > *1. "The embedding operator in Line 126 (Right)":*
>
>  **Our Response:**  We have made the revision $T^{\ast}:  \mathcal{H}\rightarrow L^2 $.
>
> > *2. "Clarify of $s$ in Assumption 4.3":*
>
> **Our Response:** We have now clarified in Assumption 4.3 that the source condition is specified for a given $s$.
>
> > *3. "Explicit Definition  of  $f_{\rho}^{\ast}$ in Theorem 5.1":*
>
> **Our Response:** We now consistently use the notation $f_{\rho}^{\ast}\left ( \mathbf{x}  \right )=\mathbb{E}_{(\mathbf{x},y)\sim \rho} \left [ y| \mathbf{x}\right ]$. In addition, we have clarified the definition of the joint distribution $\rho$ of $(\mathbf{x},y)$  in theorems.
>
> > *4. "Write 4.7 and 5.1 in the same language":*
>
> **Our Response:** Although the rates in Theorems 4.7 and 5.1 match for given $s$ and $\gamma$, we agree that the use of $p$ could cause confusion at the boundary cases. We have revised the statements accordingly to ensure consistent language throughout.
> ***
> # Author's Response to Question 1:
>
> >*1. "Do you what happens if instead of stopping early the algorithms are run until convergence? Do they converge to the KKR solutions?"*
>
> **Our Response:** I think what you may be referring to is a scenario different from the single pass SGD we consider: specifically, when a fixed dataset is given and the algorithm performs infinitely many iterations over this fixed sample set, the resulting function tends to $f_{\infty }(\mathbf{x} )=K(\mathbf{x},\mathbf{X})K(\mathbf{X},\mathbf{X})^{-1}\mathbf{y}$, which corresponds to the kernel interpolation solution.
> If we consider single-pass SGD with an infinite stream of i.i.d. samples, the procedure falls under the asymptotic setting, and the excess risk of the solution will converge at the asymptotic convergence rate.

---

> > ### Comment · Reviewer_bPCa · 2025-04-01
> >
> > Thank you for the clarification. I maintain my view that this paper has important results and should be accepted.

---

### Decision · Program_Chairs · 2025-05-01

**Decision:**

Accept (poster)

**Comment:**

This paper studies the generalization performance of kernel regression trained by single-pass SGD and compares the method to ridge regression. The analysis is conducted under the standard source condition assumption on the target. Two scaling are considered the 'high-dimensional' setting where number of sample (n) is proportional to (d) number of parameters and $n = d^{\eta}$.  The results show that exponentially decaying step sizes allow SGD to achieve minimax optimal rates and a constant stepsize with iterate averaging is proven to match the minimax lower bounds in high-dimensions.

The results are interesting as shared by the reviewers and extends a long history of research in this area. I suggest that the authors take into consideration the concerns of the reviewers particularly including references and cleaning up the proofs in the appendix.